# Tracing back primed resistance in cancer via sister cells

Jun Dai [1,7], Shuyu Zheng [1,7], Matías M. Falco [1], Jie Bao[1], Johanna Eriksson [1], Sanna Pikkusaari [1], Sofia Forstén[1], Jing Jiang[1], Wenyu Wang[1], Luping Gao[1], Fernando Perez-Villatoro [1,2], Olli Dufva [3], Khalid Saeed[3], Yinyin Wang[1], Ali Amiryousefi [1], Anniina Färkkilä [1,2,4,5], Satu Mustjoki [3], Liisa Kauppi [1], Jing Tang [1,8] ✉ & Anna Vähärautio [1,6,8] ✉

Exploring non-genetic evolution of cell states during cancer treatments has become attainable by recent advances in lineage-tracing methods. However, transcriptional changes that drive cells into resistant fates may be subtle, necessitating high resolution analysis. Here, we present ReSisTrace that uses shared transcriptomic features of sister cells to predict the states priming treatment resistance. Applying ReSisTrace in ovarian cancer cells perturbed with olaparib, carboplatin or natural killer (NK) cells reveals pre-resistant phenotypes defined by proteostatic and mRNA surveillance features, reflecting traits enriched in the upcoming subclonal selection. Furthermore, we show that DNA repair deficiency renders cells susceptible to both DNA damaging agents and NK killing in a context-dependent manner. Finally, we leverage the obtained pre-resistance profiles to predict and validate small molecules driving cells to sensitive states prior to treatment. In summary, ReSisTrace resolves pre-existing transcriptional features of treatment vulnerability, facilitating both molecular patient stratification and discovery of synergistic pre-sensitizing therapies.

Recent studies have highlighted the role of non-genetic heterogeneity driving cancer treatment resistance[1–4]. Mechanistically, such heterogeneity accumulates continuously via inherent stochasticity in gene expression, and is further strengthened upon each cell division due to uneven partitioning of biomolecules and organelles[5]. As a result, even isogenic cancer cells exhibit variable drug responses, associated with transient changes in their chromatin or expression states before the treatment[1–4]. Multiple methodologies have been developed to explore these pre-existing resistant states, involving physical separation of cell populations[6] or predefined markers[7]. More recently, labelling lineages by random barcodes that are readable by single-cell RNA-sequencing

(scRNA-seq) has enabled a fate-coupled analysis of full transcriptomic profiles. They have been applied to study induced reprogramming[8], hematopoietic differentiation[9] and lately also cancer resistance. For example, ClonMapper allows physical separation based on barcode hybridisation with scRNA-seq readout[10]. However, ClonMapper analyses fate-coupled transcriptomes on very large progenies, and is thus optimal in finding only the structures with clear and stable transcriptional differences. The Watermelon system further couples lineage labels to proliferation rate but was able to determine specific transcriptomics patterns associated with resistant fates only during, not prior to the drug treatment[11]. Furthermore, the published studies do

[1]Research Program in Systems Oncology, Research Programs Unit, Faculty of Medicine, University of Helsinki, Helsinki, Finland. [2]iCAN Digital Precision Cancer Medicine, Helsinki, Finland. [3]Research Program in Translational Immunology, Research Programs Unit, Faculty of Medicine, University of Helsinki, Helsinki, Finland. [4]Institute for Molecular Medicine Finland, Helsinki Institute of Life Sciences, University of Helsinki, Helsinki, Finland. [5]Department of Obstetrics and Gynecology, and Clinical Trial Unit, Comprehensive Cancer Centre, Helsinki University Hospital, Helsinki, Finland. [6]Foundation for the Finnish Cancer Institute, Helsinki, Finland. [7]These authors contributed equally: Jun Dai, Shuyu Zheng. [8]These authors jointly supervised this work: Jing Tang, Anna Vähärautio. ✉e-mail: jing.tang@helsinki.fi; anna.vaharautio@helsinki.fi

not strictly control the number of cell divisions, therefore leading to large transcriptional divergence within the progenies, nor do they estimate the level of this divergence for distinct transcripts and thus are unable to assess the accuracy of fate-associated expression patterns across the transcriptome.

We addressed these limitations by developing ReSisTrace, which leverages the transcriptional similarity of sister cells to enable a high-resolution prediction of primed resistance. ReSisTrace also allows exploring asymmetric transcriptomic features between sister cells, which accumulate phenotypic heterogeneity upon each generation of cancer cells. We further analysed primed resistance in the subclonal context to address the interplay between genetic and non-genetic traits. We applied ReSisTrace against chemotherapy, targeted therapy, and immune cytotoxicity, all in the context of high-grade serous ovarian cancer (HGSOC). The improved resolution of ReSisTrace enables the detection and targeting of distinct transcriptomic patterns that precede treatment resistance, allowing us to discover cell states with increased vulnerability to each applied treatment, as well as small molecules that drive cells to these states.

## Results

### ReSisTrace reveals primed resistance via sister cell inference

To reveal the cell states that are primed to treatment resistance, uniquely labelled cells are synchronised and allowed to divide once, after which sample is split so that half of the cells are analysed by scRNA-seq, while the other half undergoes anti-cancer treatment (Fig. 1a). Surviving cells were allowed to recover, and then analysed by scRNA-seq to identify resistant lineages and their gene expression profiles. We determined the transcriptomes of these resistant lineages at the pre-treatment stage based on sister-cell coupling, and annotated them as pre-resistant cells. For simplicity, cells with labels missing from post-treatment samples were annotated as pre-sensitive whilst they inherently also contain false negative pre-resistant cells due to inevitable cell loss, and cells that failed to divide after release from thymidine block.

We applied ReSisTrace in a HGSOC cell line Kuramochi[12] to address primed resistance against carboplatin chemotherapy, the PARP inhibitor olaparib, and anti-tumor immunity represented by natural killer (NK) cells. Carboplatin is part of standard-of-care for HGSOC patients, while PARP inhibitors are used as maintenance therapy especially for patients with homologous recombination deficiency (HRD)[13,14]. Kuramochi harbours a heterozygous *BRCA2* mutation and is functionally HRD[15,16], although showing lower sensitivity to PARP inhibitors than homozygous *BRCA1/2* mutant cell lines[16]. NK cells are one of the key players in anti-cancer immunity[17], yet - unlike CD8 + T cells - do not need to be autologous. We also included a non-treatment control condition that mimicked the cell cycle synchronisation, splitting, re-plating, and growth conditions of the drug treatment experiments, allowing us to assess the confounding effect of experiment-specific growth fitness within primed drug resistance signals.

Cells were labelled uniquely by using lentiviral constructs with random barcodes of 20 bases, and then sub-sampled from a large pool, analogous to the concepts used in unique molecular identifiers (UMIs)[18]. The ReSisTrace construct was incorporated to the Perturb-seq[19] vector backbone (Fig. 1a), achieving high detection efficiency both before and after the treatments, in approximately 90% of cells (Supplementary Fig. 1a). On average, 87% of the barcodes were unique in each pre-treatment sample (Supplementary Fig. 1b). Cell synchronisation with thymidine block increased the proportion of S phase cells from 29% in control cells to 61%, whereas 8 h after release from thymidine block the proportion of S phase cells was only 19% (Supplementary Fig. 1c, d, Supplementary Methods). As expected, the majority of colonies before splitting the pre-treatment samples consisted of two cells (Supplementary Fig. 1e, f), whereas after splitting, most lineages were represented by only one cell (Supplementary Fig. 1g). To assess

the accuracy of pre-sensitive cell labels while taking into account experimental uncertainties, we performed computational modelling and simulations where we incorporated experimentally measured parameters for uneven cell doubling before the sample was split, random distribution of sister cells during the split, as well as various forms of cell loss during the experimental procedure and analysis (See Supplementary Note for further details). The simulation showed that on average more than 80% of pre-sensitive cells captured by ReSis-Trace were truly positive (Supplementary Fig. 1h, Supplementary Note and Supplementary Data 1) at the killing rate selected for the experiment (Supplementary Fig. 1i). This suggests that the true pre-resistant cells can be captured with four to five-fold enrichment in the observed pre-resistant population compared to the observed pre-sensitive population, despite experimental uncertainties. Furthermore, we performed all the assays in two replicates, achieving highly consistent transcriptomic changes between the pre-resistant and pre-sensitive populations (Supplementary Fig. 1j). Notably, we observed consistent transcriptomic changes also in the non-treatment control condition between pre-surviving and pre-extinct cells, suggesting that the experimental procedures themselves, such as those introduced by the preceding release from the thymidine block, pose a non-random selection pressure which should be addressed.

To validate the use of sister cells as proxies, we confirmed that cells with shared labels, i.e. putative sister cells, have significantly more similar transcriptomes than random pairs of cells (Fig. 1b–d; average Euclidean distances in random cell pairs are 46% higher than those of sister cell pairs). We further compared the genes that showed significantly higher similarity within sister cell pairs, termed sister-concordant genes, against the other genes, termed sister-discordant genes (Fig. 1e). The sister-discordant genes were expressed at lower levels, along with increased drop-out noise ($P < 2.2 \times 10^{-16}$, two-tailed $t$-test; mean expression of the sister-discordant genes being 3.4% of that of the sister-concordant genes) (Supplementary Fig. 2a). Sister-discordant genes also showed on average 49% higher relative degradation rates ($P = 4.2 \times 10^{-10}$, two-tailed $t$-test) and on average 65% higher splicing rates relative to transcription ($P = 0.000038$, two-tailed $t$-test) (Fig. 1f and Supplementary Fig. 2b, c), suggesting that transcripts to be degraded show higher expressional variation. While sister-concordant genes were not enriched in any pathway, the sister-discordant genes were highly enriched in Inositol phosphate metabolism and Phosphatidylinositol signalling system (Fig. 1g). These pathways provide substrates for Phosphatidylinositol 3-kinase/AKT/mTOR signalling, which has been shown to drive asymmetric cell division in divergent contexts including cancer[20–22].

### Subclonal enrichment contributes to pre-resistance signatures

We next explored the structure of cell populations in the transcriptomic space. A UMAP embedding showed no distinct separation between the pre-resistant and pre-sensitive cells (Fig. 2a and Supplementary Fig. 3a). As expected, all the post-treatment samples showed larger clone sizes (Supplementary Fig. 3b) ($P < 2.2 \times 10^{-16}$, two-tailed Kolmogorov-Smirnov test) and smaller lineage diversity compared to the pre-treatment samples (Supplementary Fig. 3c) ($P = 3.80 \times 10^{-5}$, paired two-tailed $t$-test, $n = 8$). This was true also for the control condition, further highlighting the need to assess the pre-resistance signals with the experimental fitness signal. To assess the putative role of genetic heterogeneity, we inferred copy number variations (CNVs) from the scRNA-seq data, and identified five subclones that closely matched the unsupervised Leiden clustering in the UMAP (Fig. 2b, Supplementary Fig. 4a, and Supplementary Fig. 4b; adjusted Rand index 0.83). The subclonal identities of lineages were stable during the treatments (Supplementary Fig. 4c), resulting in similar subclonal proportions between pre-resistant and post-treatment populations (Fig. 2c). These results suggested that subclonal selection is already evident in the pre-resistant cells, as would be expected based on their

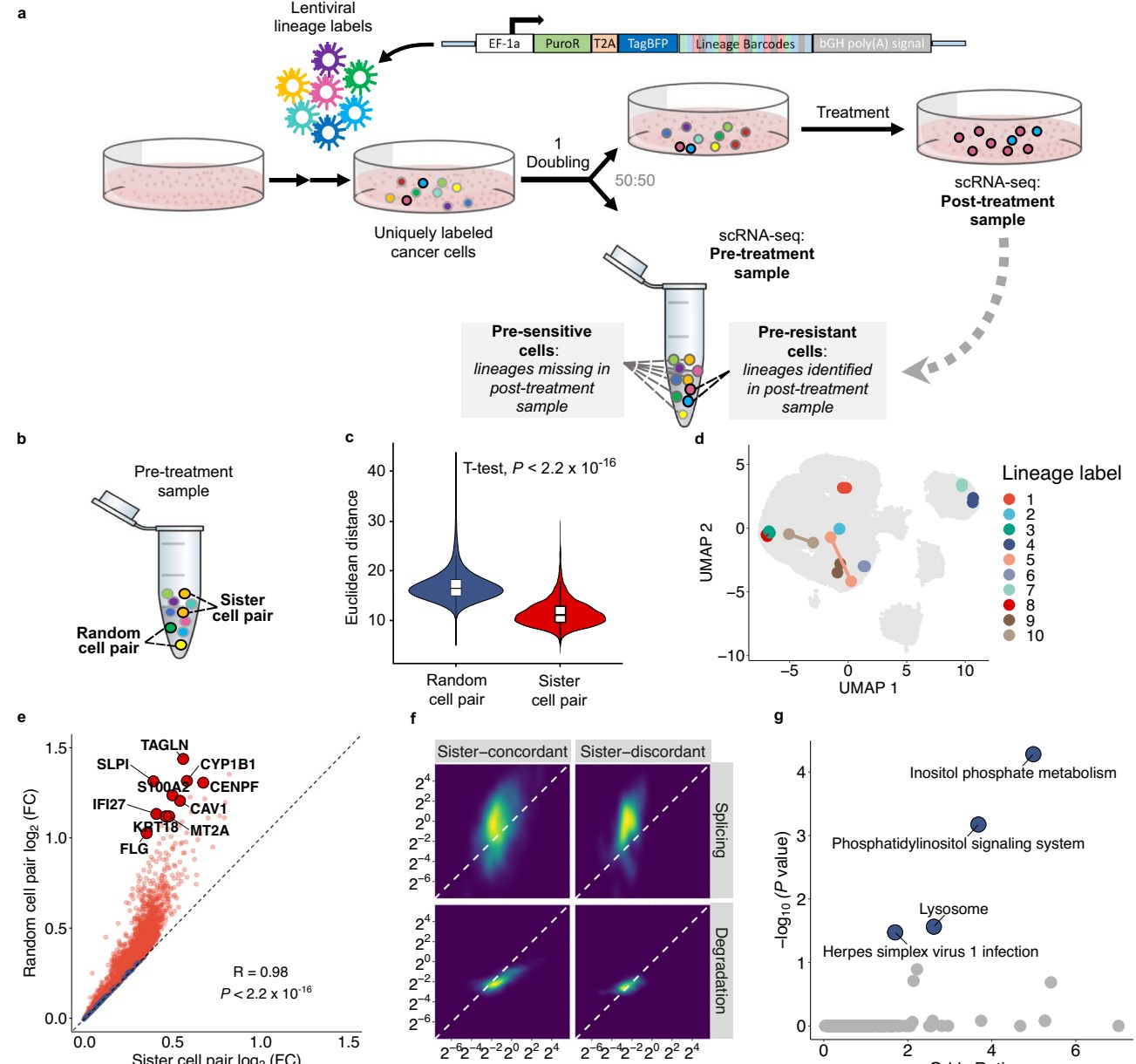

**Fig. 1 | ReSisTrace leverages sister cell similarity to discover cellular states primed for resistance. a** Schematic representation of the ReSisTrace approach. Uniquely labelled cells are synchronised and allowed to divide once. Subsequently, the sample is divided in two for scRNA-seq and treatment. Next, barcodes of recovered cells in the post-treatment sample are determined by scRNA-seq and used to identify pre-resistant cells in the pre-treatment sample. **b** Schematic of sister cell pairs and random cell pairs in a pre-treatment sample. **c** Euclidean distance of sister cell pairs in comparison to random cell pairs ($P < 2.2 \times 10^{-16}$, two-tailed $t$-test, random cell pairs $n = 6.85 \times 10^8$; sister cell pairs $n = 3477$. The boxplots show the 25th percentile, median, and 75th percentile, with the whiskers indicating

the 1.5 × interquartile range. **d** A subset of randomly selected 10 sister cell pairs shown in the Uniform manifold approximation and projection (UMAP). **e** Gene expression fold changes in sister cell pairs ($x$ axis) versus random cell pairs ($y$ axis). The Pearson's correlation and its two-tailed test $P$ value are shown. Sister-concordant genes are displayed as red, and sister-discordant genes as blue. **f** Inferred splicing and degradation rates of expression matched sister-concordant and sister-discordant gene sets, with 919 and 107 genes, respectively. **g** Pathway enrichment of 1332 sister-discordant genes. $P$ values were determined using Fisher's exact test and adjusted with the Benjamini-Hochberg method. Source data are provided as a Source Data file.

shared genetic origin with post-treatment clones. Specifically, we found that subclone A was significantly enriched in the pre-resistant cells compared to pre-sensitive cells at all the treatment conditions, especially for the carboplatin and olaparib treatments, when compared to the control samples ($P = 8.882 \times 10^{-11}$ and $P < 2.2 \times 10^{-16}$, respectively, two-proportion $Z$-test) (Fig. 2c).

With the identity of pre-resistant and pre-sensitive cells, as well as the identity of subclones, we determined rank-based transcriptomic signatures for pre-resistance and subclones (Supplementary Data 2).

We found that the signature of subclone A was positively associated with the pre-resistance signatures for all treatment conditions while negatively associated with the pre-survival signature of the control condition (Fig. 2d; associations estimated by the connectivity scores defined in[23]). Overall, the patterns were consistent with the enrichment patterns of the subclones in the pre-resistant populations (Fig. 2c, d). Gene ontology (GO) enrichment analysis revealed that mRNA surveillance associated terms, nonsense mediated mRNA decay and viral transcription, were enriched in the primed resistance signatures

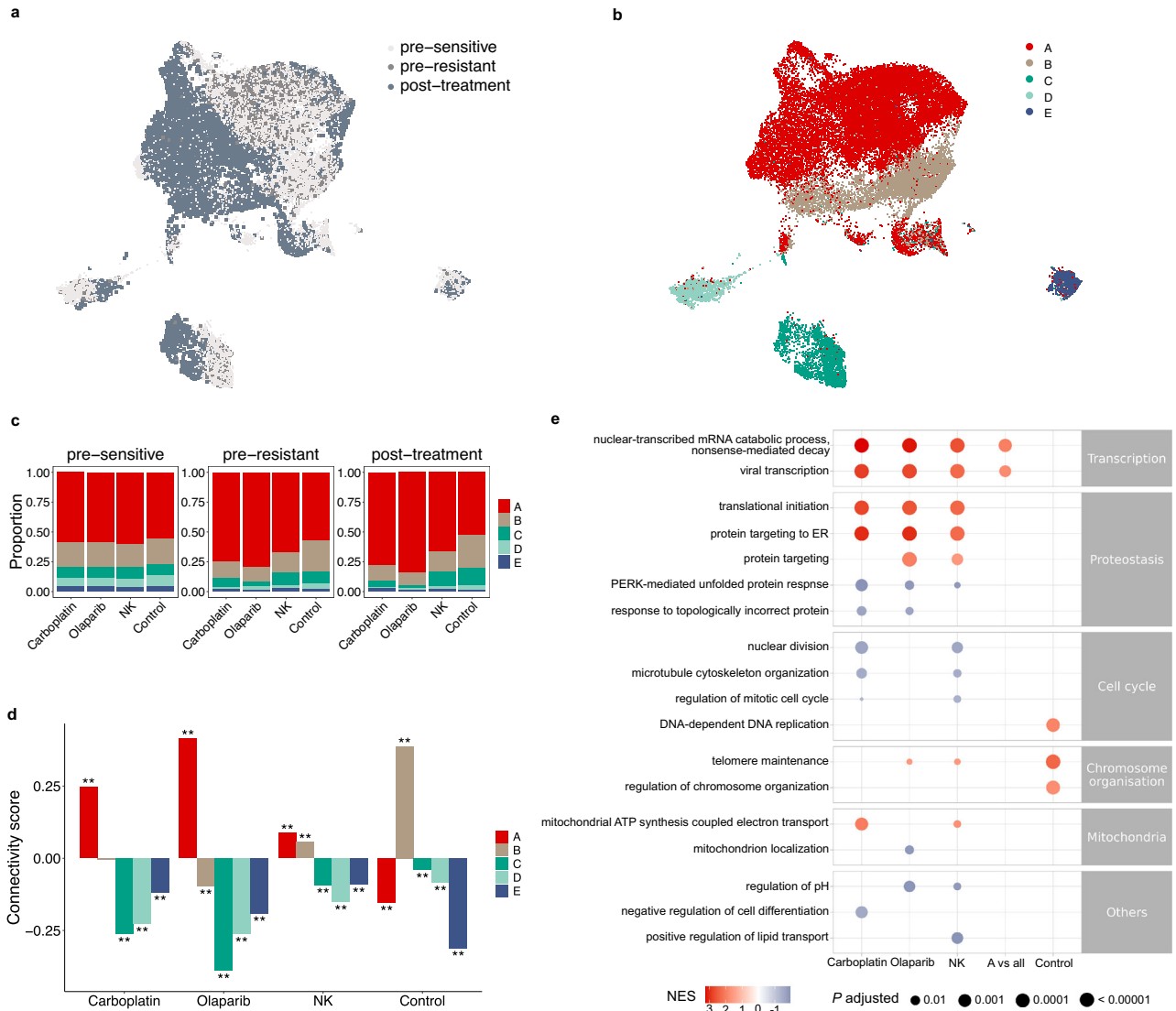

**Fig. 2 | Interplay between genetic and transcriptomic features in primed resistance.** UMAP projection of single cells coloured by (**a**) the treatment groups (pre-sensitive *n* = 29,540 cells, pre-resistant *n* = 3102 cells, and post-treatment *n* = 7802 cells), or (**b**) the inferred CNV subclones. (A, *n* = 24,903 cells; B, *n* = 7904 cells; C, *n* = 3824 cells; D, *n* = 2482 cells; E, *n* = 1331 cells) (**c**) Subclonal proportions in pre-sensitive, pre-resistant, and post-treatment populations for different treatments. **d** connectivity scores between the pre-resistance signatures and the

subclone specific signatures. Unadjusted *P* values were determined by a two-tailed permutation test. **\*\****P* = 0.001998. (Carboplatin *n* = 8419 genes; Olaparib *n* = 8406 genes; NK *n* = 8419 genes; Control *n* = 8410 genes). **e** Representative gene ontologies enriched in the pre-resistance signatures for each condition and the signature of subclone A. GSEA *P* values were adjusted with the Benjamini-Hochberg method. Source data are provided as a Source Data file.

against all three treatment modalities, as well as in subclone A, but not in the pre-survival signal of the control condition. Changes in proteostasis were associated with primed resistance against olaparib and carboplatin (Fig. 2e, Supplementary Data 3). Response to topologically incorrect protein was associated with primed sensitivity towards olaparib or carboplatin, in line with DNA damaging therapies potentiating endoplasmic reticulum (ER) stress to lethal levels[24]. We also found that mitochondrial expression patterns were associated with primed sensitivity towards NK, consistent with recent studies[25]. DNA replication had a positive association only with the pre-survival signature in the control condition yet not with any of the treatment conditions. This is in line with proliferation increasing the chances of sister cell survival after replating in the control condition, but decreasing survival in the face of anti-proliferative therapies, or even NK attack[26]. Interestingly, the cell cycle associations of the two DNA-damaging therapies diverged for mitosis. Mitosis was enriched in primed sensitivity against either NK cells or carboplatin that crosslinks DNA regardless of cell

cycle phase[27]. In contrast, olaparib damages DNA by stalling replication forks, where the immediate damage itself needs to occur in the S phase[28]. In line with this, mitosis was not associated with primed olaparib sensitivity. Together, our results show that subtle pre-resistant features can reflect the subclonal phenotypes that are subsequently enriched during each treatment.

## Context dependent association between HRD status and susceptibility to NK killing

HRD is the key tumour phenotype that predicts clinical response to PARP inhibitors and carboplatin in HGSOC. As the capability to repair DNA can vary even within genetically homogeneous cell populations[29], we hypothesised that cells with HRD-like states could be primed for sensitivity to olaparib and carboplatin. To test the hypothesis, we first determined an HRD-associated transcriptional signature (Supplementary Data 4) from an isogenic cell line pair model, constructed by comparing the gene expression of parental *BRCA1* mutant COV362

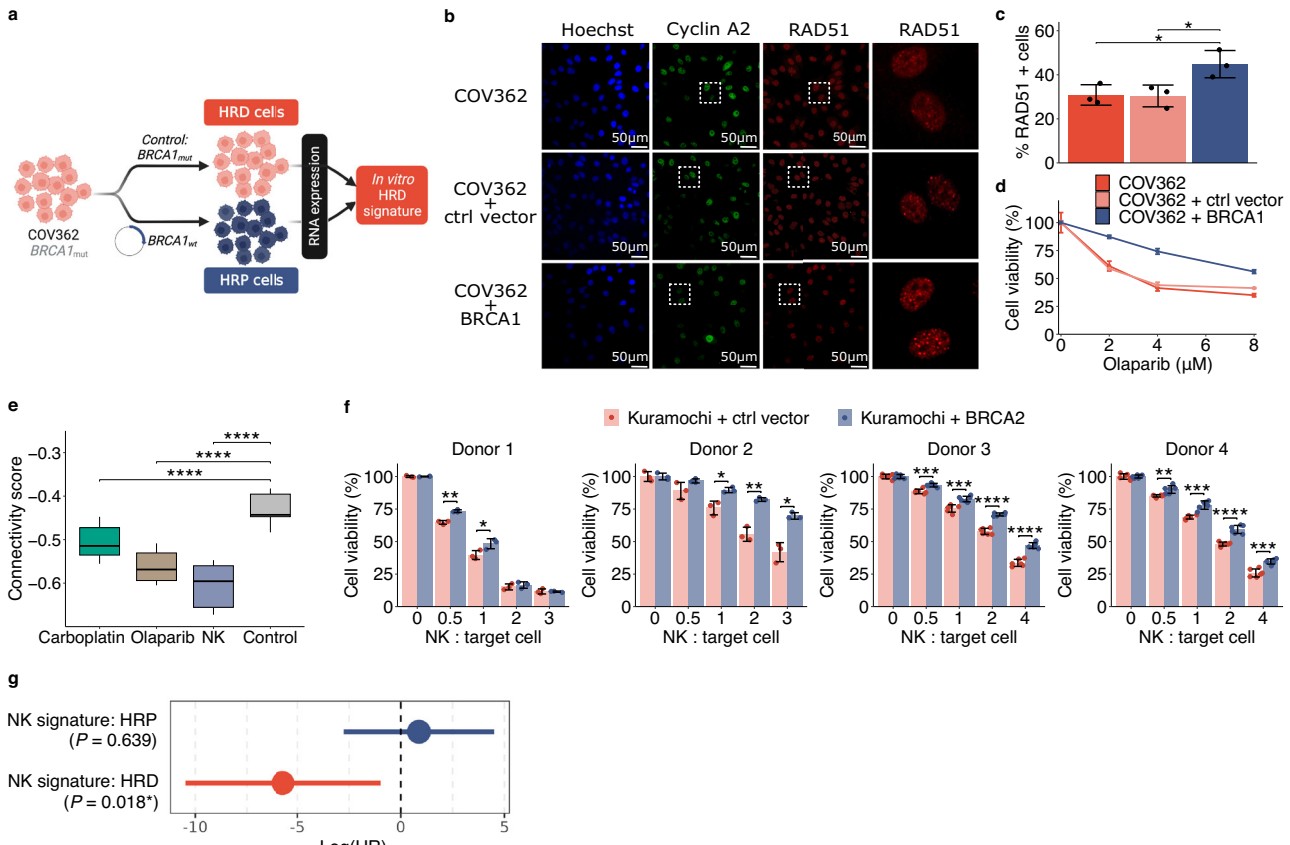

**Fig. 3 | Context-dependent vulnerability of HRD ovarian cancer to innate immunity. a** Schematic representation of the method to determine an HRD signature, derived from scRNA-seq analysis of parental COV362 cells against *BRCA1*-restored cells. **b–d** Functional HR test of RAD51 foci within cyclin A2 positive cells with representative images (**b**) and quantification from three replicates (**c**), as well as olaparib sensitivity (**d**) of the parental, control plasmid or BRCA1 restored COV362 cells. *$0.01 < P < 0.05$, two-tailed *t*-test (RAD51+ cells: COV362 vs. COV362 + BRCA1: $P = 0.039$; COV362 + ctrl vector vs. COV362 + BRCA1: $P = 0.037$). Error bars are mean +/− s.d. of three replicates. **e** Association between the pre-resistance gene signatures and the HRD signature displayed as connectivity scores. Quantiles were determined by varying the adjusted *P* value threshold for filtering the genes in the HRD signature ($5 \times 10^{-16} < P < 0.05$, $n = 20$ gene sets). ****$P < 1 \times 10^{-4}$, two-tailed *t*-test (Control vs. Carboplatin: $P = 5.18 \times 10^{-8}$; Control vs.

Olaparib: $P = 8.02 \times 10^{-15}$; Control vs. NK: $P = 5.44 \times 10^{-15}$). The boxplots show the 25th percentile, median, and 75th percentile, with the whiskers indicating the $1.5 \times$ interquartile range. **f** Measurements of NK cell-mediated killing of Kuramochi control and *BRCA2*-restored cells using NK cells from four donors. Error bars are mean +/− s.d. of at least three replicates. *$0.01 < P < 0.05$; **$0.001 < P < 0.01$; *** $1 \times 10^{-4} < P < 0.001$; ****$P < 1 \times 10^{-4}$, two-tailed *t*-test (exact *P* values are listed in Source Data file). **g** Hazard ratios of overall survival in logarithmic scale for the interaction between NK transcriptional signature and patients' HRD/HRP status. *P* values are determined by the Cox regression models (HRP $n = 117$; HRD $n = 137$). Error bars are mean +/− 95% CI, with NK signature:HRP (mean in log(HR): 0.87 +/− 3.63); NK signature:HRD (mean in log(HR): 5.72 +/− 4.73). Source data are provided as a Source Data file.

cells to the *BRCA1*-restored cells (Fig. 3a). We confirmed that the restored cells displayed a homologous recombination proficient (HRP) phenotype with increased RAD51 foci after DNA damage, resulting in reduced olaparib sensitivity (Fig. 3b–d and Supplementary Fig. 5a). Kaplan-Meier survival analysis confirmed that HRD signature was associated with improved overall survival (OS) in ovarian serous cystadenocarcinoma patients in The Cancer Genome Atlas (TCGA) cohort (hazard ratio = 1.51, $P = 0.009$, log-rank test) (Supplementary Fig. 5b).

We then compared the HRD signature to the pre-resistance signatures captured by ReSisTrace. We confirmed that the pre-resistance signatures in olaparib and carboplatin conditions were negatively associated to the HRD signature, as expected based on their mechanisms of action that target HRD (Fig. 3e). More interestingly, the pre-resistance signature for the NK condition showed even more pronounced negative connectivity scores to the signature (Fig. 3e). This unexpected finding prompted us to explore a causal connection between HRD and sensitivity to NK treatment in ovarian cancer cells. Indeed, we found that the *BRCA2* restored Kuramochi cells (Supplementary Fig. 5c, e) were consistently more resistant to NK when

compared to cells transduced with control plasmid (Fig. 3f). Furthermore, the other two isogenic BRCA1 restored cell lines (COV362 and UWB1.289 (Supplementary Fig. 5d, f, g)) also showed increased resistance against the NK cells from the same donor that was used in the ReSisTrace experiment (Donor 1 from Supplementary Fig. 5h, i). However, the responses to NK cells from other donors did not recapitulate this, suggesting donor-specificity of the NK pre-resistance signature (Supplementary Fig. 5h, i). In the non-autologous context, variations in the HLA-C type can influence the interaction between NK cells and target cells[30], but the differences of HLA-C haplotype matches could not explain the inconsistency between Kuramochi and other cell lines (Supplementary Data 5). To investigate the potential of the NK treatment at a population level, we stratified TCGA ovarian cancer cohort[31] into HRD and HRP groups by a genomic classifier called ovaHRDscar[32], and compared their deconvoluted[33] gene expression profiles. We found that the NK pre-sensitivity signature was positively associated with the clinical HRD signature (connectivity score: 0.38 +/− 0.03). Furthermore, we used known NK cell markers (Supplementary Data 6) to construct an NK signature that indicates presence of NK cells in clinical specimens. Indeed, the NK cell presence was associated with

improved survival only in the HRD patients ($P = 0.018$, Cox proportional hazard model), but not in the HRP patients (Fig. 3g). These results suggest that HRD tumours might be more responsive to innate immune cytotoxicity in the autologous context.

## Targeting treatment-specific primed resistance with small molecules

We hypothesised that small molecules that shift cancer cells toward pre-sensitive states could sensitise them to the corresponding treatments. We searched the L1000 database[34] for such drugs that induce gene expression changes opposite to the pre-resistance signatures (Fig. 4a, upper panel). Top drugs were ranked by the negative connectivity scores, which were further prioritised by a consensus $P$ value threshold derived from multiple variants of the pre-resistance signatures (Fig. 4b and Supplementary Data 7). In addition to the treatment-specific pre-resistance signatures with and without $P$ value thresholding, we used additional variants derived from the contrasts of pre-resistance and pre-survival signatures, either unfiltered or filtered by bootstrapping and/or permutation test $P$ values, and selected drugs that were predicted from all the six signature variants. Drugs with non- or lowly expressed targets in Kuramochi cells were excluded, resulting in ten drugs for experimental validation (Supplementary Data 8).

We performed a two-step functional validation for the predicted drugs. As transcriptional responses to many drug classes are cell-type specific[35], we first assessed whether the drugs indeed induced pre-sensitive signatures also in the context of interest, i.e. Kuramochi cells, and then measured their ability to sensitise these cells to either carboplatin, olaparib or NK cells, when used prior to the main treatment (Fig. 4a, lower panels). For the first step, we generated drug-induced transcriptome profiles in Kuramochi cells through scRNA-seq, with two replicates of each treatment and all samples processed simultaneously to minimise batch effects. Apart from sitagliptin that did not induce consistent changes in gene expression, the drug-induced replicates showed aligned gene expression changes, with average correlation coefficient of 0.78 (Supplementary Fig. 6). The UMAP projection shows that three proteasome inhibitors (bortezomib, delanzomib, and ixazomib) induced transcriptome profiles that were clearly clustered together and distinguished from the DMSO control (Fig. 4c, d, Supplementary Fig. 7). In addition, the expression profiles resulting from treatment with pevonedistat, a neddylation inhibitor[36], when applied in two distinct concentrations, did not overlay with DMSO treated cells' profiles on the UMAP projection. We found that the predicted drugs indeed induced changes opposite to the corresponding pre-resistance signatures, with significant negative connectivity scores (Fig. 4e, Supplementary Fig. 8). For example, clofarabine, one of the top predicted pre-sensitisers for NK cells induced transcriptomic changes with highly negative connectivity to NK pre-resistance. Pevonedistat, predicted to pre-sensitise cells against all three treatments, induced changes with a modest yet significant negative connectivity for their pre-resistance signatures. In contrast, pevonedistat showed a positive connectivity score to the pre-survival signal in the control condition, which was not observed by any other drug.

As the second validation step, we assessed whether the predicted drugs indeed sensitised cells to the treatment in a synergistic manner. We pre-treated the Kuramochi cells with the predicted drugs, changed the media to remove the drugs, and then perturbed the cells with carboplatin, olaparib, or NK cells. The predictions were highly accurate: of the 12 predicted drugs, 11 showed synergy according to multiple reference models (Bliss, HSA, Loewe, or ZIP)[37,38], and eight showed synergy consistently by all of the reference models (Fig. 4f and Supplementary Fig. 9). Importantly, these drugs were confirmed to induce pre-sensitive-like transcriptomic changes in Kuramochi cells, thus validating that the pre-resistance signatures identified by ReSisTrace indeed have functional significance. In particular, pevonedistat,

ixazomib citrate, and GW843682X were validated as the top synergistic drugs for carboplatin, olaparib, and NK cells, respectively (Fig. 4e, f). Furthermore, we confirmed that pevonedistat - that induced an expression profile similar to pre-sensitive cells in all the treatments yet not in the control condition - strongly pre-sensitised Kuramochi cells to all the three treatment modalities already at a low concentration (0.33 μM), despite having a minor inhibition on cell viability as a monotherapy (Supplementary Fig. 9a). Pevonedistat inhibits neddylation, a ubiquitin-like post-translational modification affecting a wide range of cellular functions, such as mRNA splicing and surveillance, DNA replication, and proteostasis[36]. Pevonedistat has shown synergy with multiple treatment modalities in preclinical models, and is currently undergoing several clinical trials as a combination therapy in both haematological and solid cancers[39,40]. Ixazomib citrate, together with other three proteasome inhibitors (delanzomib, carfilzomib, and bortezomib) synergized with olaparib, in line with the increased response to topologically incorrect protein in olaparib pre-sensitive cells (Fig. 2e). In contrast, the NF-κB inhibitor pyrrolidine-dithiocarbamate was less synergistic with olaparib, whereas sitagliptin used to treat type II diabetes had no effect, in line with its failure to induce robust expression changes. For the NK treatment, two drugs that cause mitotic arrest were identified, including the PLK1/3 inhibitor GW843682X and the tubulin polymerisation inhibitor nocodazole, corroborating both the NK pre-sensitive-like expression changes they induced (Fig. 4d, Supplementary Fig. 8) and gene ontology results (Fig. 2e). In addition, the purine nucleoside antimetabolite clofarabine was synergistic, suggesting that the cell cycle phase of cancer cells modulates their vulnerability to NK cells. Taken together, the drug signature analysis based on ReSisTrace enables a systematic and accurate prediction of sensitising small molecules that reverse the pre-resistant states to induce synergistic interactions with the target treatment.

## Discussion

Fates of cancer cells with recently shared ancestry are closely intertwined, showing aligned drug responses[41] and preferential site of metastasis[42,43]. Cells of the same progeny, at least until a couple of generations apart, also present similar gene expression profiles[9,44]. However, transcriptional patterns of pre-treatment cells do not fully capture their future fate[9], and cells with extreme fates, such as high-dose drug persistence, may not display evident gene expression features prior treatment[11]. Compared to previous lineage tracing methods that allow more than one cell division after labelling, ReSisTrace has the advantage in identifying subtle and transient fate-coupled differences in gene expression patterns that are increasingly obscured by stochastic noise upon each cell division. Live cell tracking has revealed that divergence in protein expression significantly increases already from sister cells, i.e. two cell progenies, to cousins or four cell progenies[6]. However, a larger progeny size[11], and even clonal isolation[10] of pre-resistant populations are beneficial when characterising primed states of extremely rare persisters in order to maximise the odds of detecting each pre-resistant lineage. Thus, the selection of optimal lineage tracing method depends on the context, and is a trade-off between sensitivity to detect and analyse the rare progenitors of extreme perisister lineages, and sensitivity to identify more transient expression patterns. Furthermore, as cell cycle synchronisation not only enriches sister cells but also poses additional stress to the cells being assayed, it should be carefully considered especially in more sensitive cellular contexts, such as patient-derived organoids, or primary cells.

Non-genetic heterogeneity should ideally be assayed in the context of genetic heterogeneity, as clinical tumours are never completely isogenic due their inherent genomic instability. We have previously shown that subclones can be traced through the treatment in clinical tumours to assess enrichment of pre-existing cellular states[45]. To

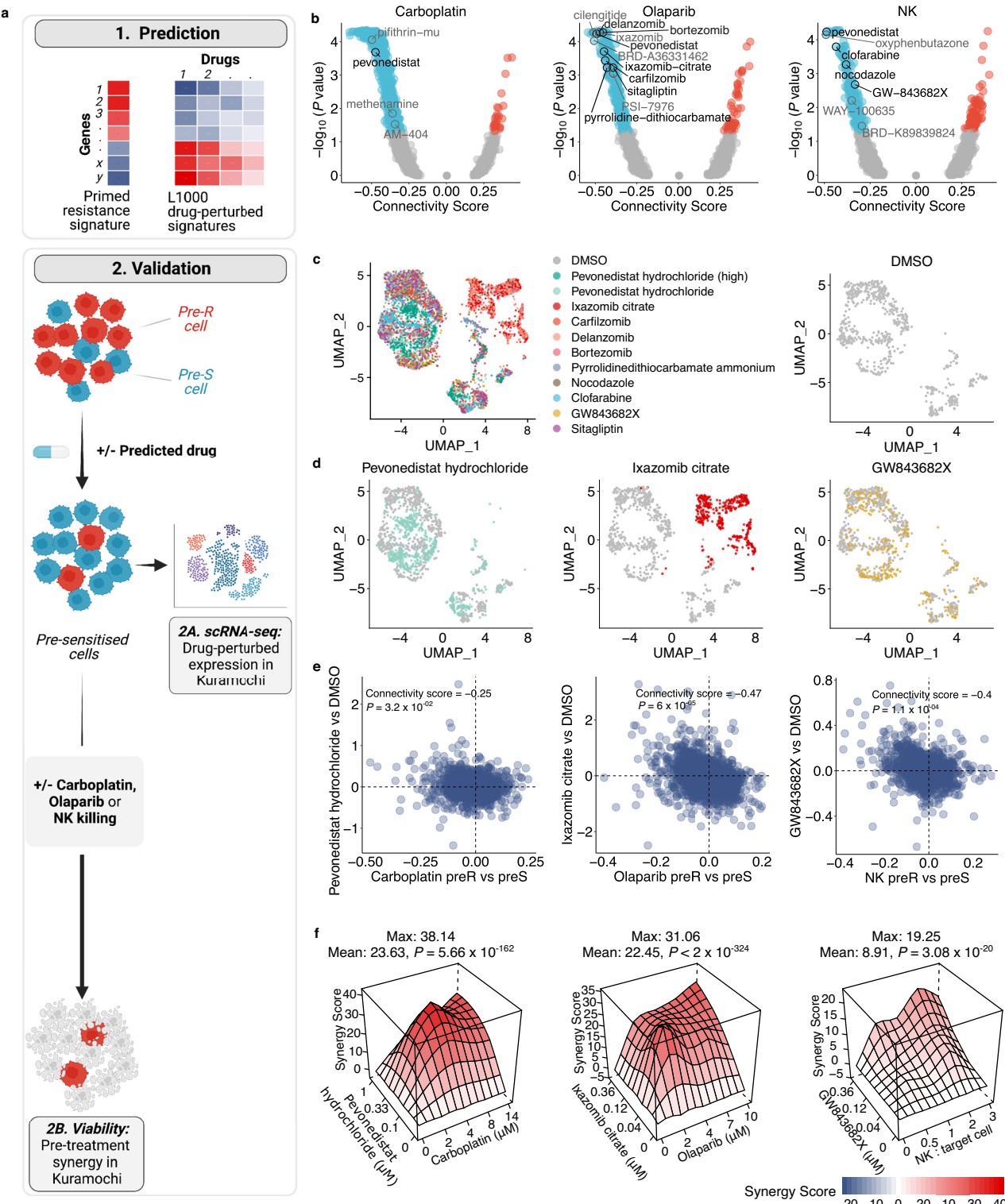

**Fig. 4 | Predicting drugs targeting primed resistance. a** Schematic representation of the approach to predict and validate drugs that leverage the acquired pre-resistance profiles by sequential treatment strategies. **b** Connectivity scores between the L1000 drug-induced signatures and acquired pre-resistance signatures filtered by unadjusted $P < 0.05$. $P$ values were determined by a two-tailed permutation test. **c** Single-cell gene expression of drug perturbed Kuramochi cells projected on a UMAP, with top predicted drugs shown as separate UMAP

projections along with DMSO induced profiles (**d**), and with induced fold changes compared to those of respective pre-resistance profiles (**e**). Unadjusted $P$ values were determined by a two-tailed permutation test. **f** Synergy landscapes for the top drugs for each treatment. Mean and maximal synergy scores are shown, with their two-tailed $P$ values determined by bootstrapping test. Source data are provided as a Source Data file.

improve clinical relevance of ReSisTrace in HGSOC that is an extremely copy number unstable cancer[46], we analysed primed resistance within the subclonal context via scRNA-seq inferred copy number variations. We found that subclonal structures drove global transcriptional clustering, and discovered treatment-specific enrichment of subclones. These enrichment patterns explained certain transcriptomic features of the pre-resistant populations, such as those representing changes in transcriptional quality control. However, the majority of the biological processes enriched in treatment-specific pre-resistance patterns - such as those related to proteostasis or cell cycle - were not detected by subclonal analysis. This suggests that the increased resolution of sister cell lineage tracing enables the discovery of deeper, non-genetic features preceding resistance that cannot be identified by tracing cellular progenies at the lower, subclonal resolution. It is important to consider that subclonal enrichment contributes to differential expression of not only CNV affected drivers but also the co-amplified passenger genes[47], thus providing an additional level of noise to the pre-resistance signal and supporting the analysis of pre-resistant states in their subclonal context. Thus, ReSisTrace revealed the interplay between genetic and phenotypic heterogeneity in primed resistance, an aspect that becomes increasingly relevant for patient-derived models such as organoids or xenografts.

DNA damage is known to induce NK-mediated cell killing[48] and aneuploidy activates NK clearance of non-transformed senescent cells via NF-κB signaling[49]. However, the effect of HRD on NK cytotoxicity has not been reported, although activation of intracellular innate immune signalling has been shown to induce HRD[50]. In Kuramochi cells, we identified a connection between subtle intra-population differences in HRD state and sensitivity to NK killing, which was validated by showing increased resistance upon BRCA2 restoration against NK cells from four donors. However, other NK donor combinations did not show aligned responses in additional BRCA1 restored isogenic cell line pairs, and such discrepancy was not explained by differences in HLA-C haplotype matches. It is also known that some cell lines are more sensitive to TRAIL, TNF or Fas ligand mediated, and some more to granzyme or perforin mediated killing[51,52]. Thus, one putative reason for the inconsistency between Kuramochi and other cell lines could be that the effect of HRD is more relevant in some of these NK-induced apoptotic mechanisms than others. This suggests that primed resistance against innate immune cells can be very context-specific. Interestingly though, the presence of NK derived transcripts associated positively with ovarian cancer patient survival only in an HRD dependent manner, suggesting that in the native, autologous context, HRD patients could be more susceptible to NK based anti-tumor therapies than patients with HRP tumours. Taken together, the putative connection between innate immunity and HRD provides an intriguing extension to the body of evidence showing that HRD tumours are more susceptible to adaptive immunity, with improved T cell-mediated immunosurveillance[53].

Drug resistance driven by non-genetic heterogeneity is mechanistically complex. Functional genetic screens can reveal causal factors but are largely limited to mechanisms that rely on individual genes. Longitudinal transcriptomic comparisons can capture complex drug-induced patterns but do not allow detecting the evolutionary selection and enrichment of pre-existing resistant states during the treatment. On the contrary, ReSisTrace allows characterization of resistant populations already prior to the treatment, thus revealing also the contrasting vulnerable states that could be induced via pre-treatments. We applied a systematic approach of using drug-perturbed gene expression signatures[34] to predict pre-sensitising drugs, and validated that context-specific induction of pre-sensitive states was a pre-requisite for effective pre-treatments. We thus leveraged the plasticity of cell states by identifying sequential treatments, providing an alternative paradigm to simultaneous combinatorial therapy predictions that aim to kill the maximal proportion of malignant cells. Importantly,

sequential treatments can mitigate many problems of simultaneous combinatorial therapies - such as elevated toxicities and adverse drug-drug interactions[54,55] - but have thus far been largely neglected in drug prediction approaches. In the clinical trial setting, we envision that the predicted drugs could be used as short-term sensitisers prior to main cytotoxic therapies that need to be started without delay for patients with advanced disease, such as those diagnosed with HGSOC. In summary, sister-resolution lineage tracing can reveal complex cell states that prime treatment resistance in cancer, and allows opposing these states to overcome drug resistance already before it emerges.

## Methods

### Cell culture
COV362 cells (Merck, 07071910) were grown in DMEM (Corning, 15-013-CV) with 10% fetal bovine serum (FBS) (Gibco, 10270106), 1% penicillin–streptomycin (PS) (Gibco, 15140122), and 1% GlutaMAX (Gibco, 35050038). Kuramochi (JCRB Cell Bank, JCRB0098) cells were grown in RPMI1640 (Corning, 10-040-CV) with 10% FBS, and 1% PS. UWB1.289 (ATCC, CRL-2945) and UWB1.289 + BRCA1 (ATCC, CRL-2946) cells were cultured according to the handling procedure from ATCC.

### Small molecules
Pevonedistat hydrochloride (HY-10484), Ixazomib citrate (HY-10452), Delanzomib (HY-10454), Bortezomib (HY-10227), Carfilzomib (HY-10455), Pyrrolidinedithiocarbamate ammonium (HY-18738), Sitagliptin (HY-13749), GW843682X (HY-11003), Clofarabine (HY-A0005), Nocodazole (HY-13520) were all purchased from MedChemExpress.

### Expansion of NK cells
NK cells were expanded by using K562-mbIL21-41BBL feeder cells as described previously[56]. Briefly, peripheral blood mononuclear cells (PBMCs) were isolated via gradient centrifugation with Ficoll-Paque from buffy coats of healthy donors. Five million PBMCs were suspended together with ten million feeder cells irradiated with 100 Gy, into 40 ml of R10 media supplemented with 10 ng/ml recombinant human IL-2 (R&D Systems, 202-IL-050). Cells were passaged twice per week and feeder cells were added in a 1:1 ratio after 7 days to maintain stimulation. After two weeks of culture, NK cells were purified with the NK Cell Isolation Kit (Miltenyi) and live-frozen. For the experiments, NK cells from four donors were used: donor 1 NK cells for ReSisTrace experiment, all donors' NK cells for BRCA1/2 restored cell line pairs, and NK cells from donor 2 to test predicted pre-sensitising drugs. All NK cells were cultured for 3 days between thawing and the experiments.

### Synthesis of lineage barcodes library
The pBA439_UMI20 plasmid (Supplementary Data 9) was constructed by ssDNA synthesis and NEBuilder® assembly tool. The lineage barcodes part ssDNA_UMI20 (5′-CTGGGGCACAAGCTTAATTAAGAATTC ANNNNTGNNNNACNNNNGANNNNGTNNNNCTAGGGCCTAGAGGGC CCGTTTAAAC-3′) was ordered from Integrated DNA Technologies. pBA439 was a gift from Jonathan Weissman (Addgene, plasmid # 85967)[19]. pBA439 vector was digested with EcoRI-HF (NEB, R3101S) and AvrII (NEB, R0174S), then the larger linear fragment pBA439_linear was recycled and purified by columns (Macherey-Nagel, 740609.50). The ssDNA_UMI20 was assembled with pBA439_linear to construct a pBA439_UMI20 library by the NEBuilder HiFi DNA assembly (NEB, E2621L). For amplification, the pBA439_UMI20 library was electroporated into 25 μl of Lucigen Endura ElectroCompetent Cells (Lucigen, 60242-2) using prechilled 1 mm electroporation cuvettes (BioRad, 1652089) in a BioRad GenePulser I machine set to 10 μF, 600 Ω, and 1800 V. Within seconds after the pulse, 1 ml of 37 °C Recovery Medium (Lucigen, 80026-1) was added and bacteria were grown in round-bottom tubes for 1 h at 37 °C while shaking at 180 r.p.m. Then, 1 ml of

the bacterial culture was plated on a 25 × 25 cm bioassay plate containing LB medium with 100 µg/ml ampicillin. Plates were incubated at 32 °C for 22 h, then LB medium was added and cells were scraped off the plate. Bacterial cells were pelleted by 15 min of centrifugation at 5000 × g at 4 °C, and plasmid DNA was extracted with a Maxiprep endotoxin-free kit (Macherey-Nagel, 740424.10).

### Lentivirus production and cell transduction

Lentiviral particles were produced by transfecting 293FT cells (Thermo Fisher, R70007) with the pBA439_UMI20 library, along with packaging constructs psPAX2 (Addgene, plasmid # 12260) and pMD2.G (Addgene, plasmid # 12259), gifts from Didier Trono. Constructs were titred by serial dilution on the 293FT cells. The lentivirus was collected 48 h and 72 h after transfection and aliquoted to store at −80 °C, following filtering through a low-protein binding 0.45-µm filter.

### Experimental BRCA +/− models

To clone the BRCA1/2 plasmids, lentiMPH v2 (Addgene, plasmid #89308) was modified by replacing Hygro selection marker with GFP to generate lentiMPH-GFP. *BRCA1* was amplified from pDEST-mCherry-LacR-BRCA1 (Addgene, plasmid #71115), and *BRCA2* was amplified from pCIN BRCA2 WT (Addgene, plasmid #16245). PCR amplified products (templates and primers presented in Supplementary Data 10) were assembled by the NEBuilder HiFi DNA assembly (NEB, E2621L) to generate lenti-EMPTY-GFP, lenti-BRCA1-GFP, and lenti-BRCA2-GFP. Lentivirus were made as described earlier to transduce either COV362 or Kuramochi cells to restore the *BRCA1* and *BRCA2* functions. lentiMPH v2 was a gift from Feng Zhang, pDEST-mCherry-LacR-BRCA1 was a gift from Daniel Durocher, pCIN BRCA2 WT was a gift from Mien-Chie Hung[57–59].

The COV362 model was validated with a colony formation assay, where COV362 control and *BRCA1* restored cells (10,000 cells per well) were exposed to 0, 2.5 or 5 µM olaparib. Colonies were stained with crystal violet after 12 days of incubation.

### ReSisTrace experimental workflow

Lentiviral p24 titer testing was performed by Biomedicum Virus Core (HelVi-BVC) using Alliance HIV-1 P24 ANTIGEN ELISA Kit (PerkinElmer, NEK050B001KT). For transduction, 320 µl of pBA439_UMI20 lentivirus library (3.25 × 10⁶ pg/ml) were used to transduce 6 million Kuramochi cells to achieve MOI between 0.2 and 0.25, and 2 days after transduction 2 µg/ml puromycin were added for a seven day selection to remove unbarcoded cells. All barcoded cells were equally divided into 6 vials (6 million cells per vial), 4 of which were cryopreserved and 2 were used for the subsequent experiment. Cells were synchronised with 2 mM thymidine (Merck, T1895) for 42 h, then cells were washed twice with PBS to remove thymidine, 16,000 cells/well were seeded on 6-well plates. After 48 h, 4 wells were used for cell counting (27,000–31,000 cells/well), and 8 wells were used for the ReSisTrace experiment. For each of the 8 samples, half of the cells were loaded for scRNA-seq as pre-treatment samples, and the remaining half were seeded back for either normal culture (13 days) or treated with carboplatin (Selleckchem, S1215, 1.2 µM, 3 days), olaparib (Selleckchem, S1060, 1.2 µM, 7 days), and NK cells (26,000 cells, 1 day). The drug concentrations or the number of NK cells were chosen to kill 70–80% of cancer cells compared to untreated cultures (Supplementary Fig. 1i). After the treatment, cells were recovered for 10 (5 cell cycles), 6 (3 cell cycles) and 7 days (3–4 cell cycles) respectively, after which indicated number of cells (Supplementary Data 11) from each well were loaded for scRNA-seq as post-treatment samples. The samples were sequenced very deeply, with approximately 0.86 billion reads per sample, to achieve high-resolution data with minimal dropout of both labels and transcripts in this proof-of-concept experiment. We recovered around 105k cells in total before QC and around 78k cells after Seurat QC. From these cells, we further removed 8640 cells without

lineage label(s) and 28,640 after treatment cells whose lineage labels were missing from the corresponding before treatment specimen (column AT-New-cell in Supplementary Data 11), resulting in 40,444 cells (3102 + 29,540 from BT and +7802 cells from AT samples) that were used for the analysis (detailed in Supplementary Data 11).

### Computational modelling and simulation of lineage tracing

For the actual ReSisTrace experiments, the protocol involves multiple steps starting from labelling the initial pool of cells to harvesting viable cells for scRNA-seq. Additional uncertainties may be introduced from e.g. unequal doubling, or cell drop-out due to quality control (Supplementary Note, Supplementary Data 12). Therefore, the lineage tracing process was simulated by considering the following parameters that may affect the experiment outcome, including (1) SampleSize - total number of cells at the start of experiment; (2) KillRateTotal - killing rate of the treatment; (3) LineagePropAfterDoubling - proportions of singletons, twins, and quadruplets after doubling; (4) NCellBT - number of cells with lineage labels in the pre-treatment sample which passed the quality control; (5) NCellAT - number of cells with lineage labels in the post-treatment sample which passed the quality control; (6) NCellPreSeq - number of cells after recovering from treatment; (7) NCellSeq - number of cells loaded to scRNA-seq.

To reflect the actual experimental setting up, prior knowledge was utilised to constrain these parameters, including (1) SampleSize to be estimated as 16,000; (2) KillRateTotal to be estimated between 70% and 80% of cell growth inhibition; (3) LineagePropAfterDoubling to be estimated as [42.46%, 53.04%, 4.5%] for the proportions of singletons, twins, and quadruplets separately, based on the observations shown in Supplementary Fig. 1e; (4) NCellBT, NCellAT, NCellPreSeq, and NCellSeq were estimated based on the results of scRNA-seq data as shown in Supplementary Data 1.

The simulation was initiated with these estimated parameters, creating a scenario where lineage labels were randomly assigned to cells and then traced into the final scRNA-seq data in both pre-treatment and post-treatment samples. Pre-sensitive predictive rate was determined, defined as the proportion of true pre-sensitive cells in the predicted pre-sensitive cells. In contrast, the pre-resistant predictive rate, defined as the proportion of true pre-resistant cells in the predicted pre-resistant cells, is always 100%. Therefore, A higher pre-sensitive predictive rate suggested a higher quality of the experiment data for determining the gene expression signature of primed resistance. To obtain a theoretical upper bound of the pre-sensitive predictive rate, an ideal scenario with perfect doubling and zero drop-out during the scRNA-seq was also simulated. Standard deviations were determined using 100 iterations (Supplementary Fig. 1h).

### Drug-induced transcriptome profiles by scRNA-seq

Kuramochi cells were seeded in 6-well plates at around 50% confluence overnight. Predicted small molecules (see concentrations from Supplementary Data 13) or DMSO as control were added for 24 h in duplicates. Then cells were simultaneously washed and detached from plates. To further minimise the batch effect, each sample was individually multiplexed with 3′ CellPlex Kit Set A (10x Genomics, PN-1000261). After labelling and washing, every 12 samples were pooled and then loaded for scRNA-seq experiments according to 10x Genomics instructions.

### Single-cell RNA-sequencing library preparation

On the 10x Genomics platform, for samples from ReSisTrace experiments, we used the Chromium Next GEM Single Cell 3′ Kit v3.1 (PN-1000268), Chromium Next GEM Chip G Single Cell kit (PN-1000120), and Dual Index Kit TT Set A (PN-1000215) were used. For samples from isogenic BRCAness +/− COV362 cells, cells were multiplexed with 3′ CellPlex Kit Set A (PN-1000261), then 7000 cells were loaded for

scRNA-seq. For library preparation, the 3' Feature Barcode Kit (PN-1000262) and the Dual Index Kit NN Set A (PN-1000243) were additionally used. The resulting cDNA libraries were quantified by both Agilent BioAnalyzer (Agilent, 5067-4626) and q-PCR (Roche, KK4835), and sequenced by Illumina NovaSeq 6000.

## NK cells mediated killing for BRCA +/− models

On day 1, 8000 cancer cells per well were seeded on 96-well plates, and NK cells were cultured with 10 ng/ml IL2. On day 2, NK cells were added into cancer cells in indicated ratios (Fig. 3f and Supplemntary Fig. 5h, i) to cancer cell numbers. On day 3, after removing the medium and the NK cells that were in suspension, the viability of adherent cancer cells was measured by the CellTiter-Glo® Luminescent Assay (Promega, G9242).

## Cell viability assay

1500 cells/well were seeded in 96-well plates and then allowed to adhere to the plates. Indicated concentrations of olaparib (Fig. 3d and Supplementary Fig. 5g) were added for 4 to 6 days. Cell viability was determined using CellTiter-Glo® Luminescent Assay (Promega, G9242).

## Homologous recombination repair functional assay

HGSOC cells were seeded on 96-well plates in 60% confluence/10,000 cells per well in triplicates and allowed to attach to the bottom for 12–24 h. The cells were irradiated with 5 Gy of ionising radiation to induce DNA damage, followed by 8 h recovery time before fixation in 2% PFA. Fixed cells were permeabilized in 0.2% Triton X-100 in PBS++ (PBS with 1 mM CaCl2 and 0.5 mM MgCl2) for 20 min, followed by 30-minute blocking in staining buffer (0.5% BSA, 0.15% glycine, 0.1% Triton X-100 in PBS++). Next, the cells were incubated with primary antibodies against RAD51 (ab133534, Abcam, diluted 1:1000) and cyclin A2 (GTX634420, GeneTex, diluted 1:500) overnight at 4 °C, followed by incubation with secondary antibodies (goat anti-mouse IgG-Alexa Fluor 488, A11029; or goat anti-mouse IgG-Alexa Fluor 568, A11004; and goat anti-rabbit IgG-Alexa Fluor 647, A21245; LifeTechnologies, diluted 1:1000) at RT for 1 h. Nuclei were counterstained with 2 µg/ml Hoechst. Images were acquired with Opera Phenix High Content Screening System (PerkinElmer) at 40x magnification. Images were analysed with ImageJ software using custom macros. At least 100 cyclin A2 positive nuclei per each well were analysed. Nuclei with ≥5 RAD51 foci were considered as RAD51 positive. Percentage of cyclin A2 and RAD51 positive nuclei out of all cyclin A2 positive nuclei was calculated.

## Drug combination screening and synergy scoring

1500 (for olaparib or carboplatin combination) or 4000 (for NK cells combination) Kuramochi cells/well were seeded in 96-well plates and then allowed to adhere to the plates. Indicated concentrations of small molecules (Supplementary Fig. 9a) were added for 24 h, then cells were refreshed with normal medium with indicated concentrations of either olaparib or carboplatin added for 5 days or with different ratios of NK cells for 24 h. At the end point, cell viability was determined using CellTiter-Glo® Luminescent Assay (Promega, G9242).

Dose-response matrices were analysed with SynergyFinder Plus (synergyfinder.org) (R package v3.4.5)[60]. Four reference models were utilised to access the synergy, including HSA (Highest single agency), Bliss (Bliss independence), Loewe (Loewe additivity), and ZIP (Zero interaction potency). As the reference models rely on different mathematical assumptions of synergy, we reported the results for all of them. The degree of interactions between two drugs derived from the HSA model was visualised in a synergy landscape over the dose matrices (Fig. 4f and Supplementary Fig. 9). Significance of average synergy score was evaluated by bootstrapping test. Default parameters were used for the analysis.

## Lineage label mapping and processing

The lineage barcodes sequence (5'-CTGGGGCACAAGCTTAATTAA-GAATTCANNNNTGNNNNACNNNNGANNNNGTNNNNCTAGGGCCTA-GAGGGCCCGTTTAAAC-3') was added to the GDCh38.d1.vd1 reference genome with GENCODE v25 annotation, as an individual gene named *pBA439_UMI_20* according to the parental lentiviral library. The Cell Ranger software (v5.0.1)[61] was used to perform read alignment and UMI quantification using the modified reference. Default parameters were used for the analysis.

Lineage label sequences for each cell were extracted from the "possorted_genome_bam.bam" file output by cellranger count command, by considering the sequences mapped to gene *pBA439_UMI_20* and the sub-pattern "CANNNNTGNNNNACNNNNGANNNNGTNNN NCT". To mitigate the effect of sequencing errors, the directional network-based method from UMI-tools (v1.0.1)[62] was applied on the lineage label sequences (cluster_method = "directional"). The sequences that differ at a single base from the representative sequence were corrected. In addition, we removed the non-unique lineage label sequences expressed by more than four cells in the pre-treatment samples. We assigned the cells from each pre-treatment sample into two groups: (1) pre-resistant cells with lineage labels matching those of the corresponding post-treatment samples; (2) pre-sensitive cells assigned to lineages that were not detected after the treatment.

## Preprocessing of the scRNA-seq data

We used the Seurat (v4.0.4)[63] to perform the data quality control, normalization, top variable gene selection, scaling, dimensionality reduction, and differentially expressed gene (DEG) analysis. For each ReSisTrace sample, the genes expressed in less than three cells were removed. Based on the distribution of the UMI counts (Supplementary Fig. 10), number of genes, and percentage of mitochondrial transcripts, we filtered out the cells using the thresholds indicated in Supplementary Data 14.

We used the "NormalizeData" function (default parameter setting) to normalize gene counts, and the "FindVariableFeatures" function (default parameter setting) to find the top 2000 variable genes. The "FindMarkers" function (test.use = "wilcox", logfc.threshold = 0, min.pct = 0) was used to determine the differentially expressed genes by Wilcoxon rank sum test between different groups. To minimize experimental noise from cases where both sisters were sampled in the scRNA-seq analysis before the treatment, we performed the differential analysis only on the cells lacking sisters in the same pre-treatment samples.

## Sister cell similarity

We merged the pre-treatment data from all the experiments. Cells with matching labels were defined as sister cells, gaining 2939 lineages with sister cells. Lowly expressed genes whose total normalised expressions are less than 1.25 (10% quantile) were removed. The UMAP coordinates for visualisation of randomly selected sister cells were determined with the first 20 principal components of the top 2000 variable genes. For extracting the sister concordant genes, we considered only the lineages with two sister cells, and removed the sister cells with the Euclidean distance of their transcriptomes larger than 14.9 (90% quantile). The genes with $\log_2$ fold changes ($\log_2$FCs) significantly lower in the sister cell pairs compared to random cell pairs were defined as the sister-concordant genes (two-tailed *t*-test with an FDR-adjusted *P* value threshold of 0.05).

## RNA velocity analysis

We selected expression-matched sets of sister-concordant and sister-discordant genes. We generated a binned histogram for the sum of expression in the pooled samples, and selected genes that belong to the bin with the most balanced sister-concordant and sister-discordant genes (the second-lowest bin).

We used the velocyto (v0.17.17)[64] to align and quantify the spliced and unspliced RNA for each gene. Default parameters were used. The dynamical model from scVelo (v0.2.3)[65] was used to fit the gene-specific rates of RNA transcription, splicing, and degradation ("scvelo.pp.filter_and_normalize" function: min_shared_counts = 20; "scvelo.pp.moments function": n_neighbors = 30; "scvelo.tl.velocity function": mode = "dynamical").

### Shannon diversity
Shannon diversity of lineage labels for each sample was determined by the "diversity" function from vegan R package (v 2.6.2)[66]. Default parameters were used for analysis.

### Pathway analysis
KEGG pathways for the expression matched sister-concordant and discordant genes were determined by Enrichr (https://maayanlab.cloud/Enrichr/)[67,68]. Gene set enrichment analysis (GSEA) function from the clusterProfiler package (3.18.1)[69] was used for the ordered fold changes (logarithmic scaled) extracted from the pre-resistance differential expression analysis to inspect the enriched gene ontology biological process terms. Default parameters were used for the analysis.

### Subclone analysis
CNV profiles for the individual cancer cells for each sample were obtained with InferCNV (1.7.1)[70] (cutoff = 0.1, analysis_mode = subclusters, leiden_resolution = 0.5, tumor_subcluster_partition_method = leiden) using a common set of stromal cells from previously published data[45] as references. Briefly, InferCNV infers changes in the number of copies by averaging relative expression levels over a sliding window across large genomic regions. To enhance the accuracy of CNV estimates, denoising filters are applied based on the standard deviation of residual normal expression values. The Leiden algorithm was used for determining the underlying subclonal populations based on the CNV profiles. At the subclone level, a Hidden Markov Model followed by a Bayesian network is utilized to compute the posterior probability of a CNV region belonging to a specific amplification or deletion state. CNV regions with higher mean posterior probabilities for the normal state are removed as likely false positive predictions. Exceptionally for the Olaparib1 sample, the algorithm was not able to recognise the smallest subclone (subclone E), due to the small sample size. Therefore, the inferCNV analysis was jointly run with the other Olaparib sample for the subclonal label determination.

### Pre-resistant gene expression signatures
After identifying the pre-resistant and pre-sensitive cells in the pre-treatment samples, fold changes of gene expressions were determined. The $log_2FC$ between pre-resistant and pre-sensitive cells for each gene was averaged across two replicates.

### HRD signatures and NK signature
Two independent HRD signatures were derived. The clinically-derived HRD signature was extracted from The Cancer Genome Atlas Cohort (TCGA) cohort where HGSOC patients were stratified into HRD/HRP groups based on a genomic scar analysis using ovaHRDscar[32]. For transcriptomic comparison between these groups, we used the limma R package (3.46.0)[71] on the bulk expression data[31] that had been deconvoluted to retain only cancer cell derived expression using PRISM (0.9)[33]. Default parameters were used for analysis. The experimental HRD signature was extracted from the BRCA1 +/− cell line pair by DEG analysis from the scRNA-seq data as described earlier. For the HRD signatures, we determined $log_2FCs$ of HRD to HRP using sister-concordant genes only, thresholded with indicated $P$ values, We used "connectivityScore" function (method = "gwc", gwc.method = 'spearman', nperm = 300) from

PharmacoGx R package (v3.0.2)[23] to determine the connectivity scores between the HRD signatures and pre-resistance signatures.

The NK transcriptional signature was obtained from the coexpressed NK cell markers in our previous analysis of 22 clinical HGSOC tumours[45]. To estimate NK cell enrichment within the immune compartment of the TCGA cohort, we determined Mann-Whitney $U$ statistic-like scores from the deconvoluted gene expressions using the UCell R package (1.3.1)[72]. Default parameters were used for analysis. A Cox proportional hazard regression model on the overall survival was built using the survival R package (3.3.1)[73], with the formula: coxph (Surv (time, status) ~ nk_signature:HRD_status), wherein 'nk_signature' refers to the UCell score described above.

### Predicting drugs to target primed resistance
Six variants of pre-resistant gene expression signatures were generated for each treatment with thresholds of $P$ values based on different adjustment methods, including: (1) Original $log_2FC$, (2) Original $log_2FC$ filtered by the Wilcoxon rank sum test $P$ value < 0.05, (3) $log_2FC$ contrast between treatment and non-treatment control, (4) $log_2FC$ contrast between treatment and non-treatment control filtered by bootstrapping test $P$ value < 0.05, 5) $log_2FC$ contrast between treatment and non-treatment control filtered by permutation test $P$ value < 0.05, and 6) $log_2FC$ contrast between treatment and non-treatment control filtered by both bootstrapping test $P$ value < 0.05 and permutation test $P$ value < 0.05.

For the contrast signatures, the $log_2FC$ between pre-resistant and pre-sensitive cells was calculated by for each gene was averaged across two replicates and then subtracted by the $log_2FC$ of pre-surviving versus pre-extinct cells from the non-treatment control samples:

$$\frac{log_2\left(\frac{R_1}{S_1}\right) + log_2\left(\frac{R_2}{S_2}\right)}{2} - \frac{log_2\left(\frac{R_1'}{S_1'}\right) + log_2\left(\frac{R_2'}{S_2'}\right)}{2} \quad (1)$$

where $R_1$ and $R_2$ are the gene expression values in treatment pre-R group in repeats 1 and 2, respectively; $S_1$ and $S_2$ are the gene expression values in treatment pre-S group in repeats 1 and 2, respectively; $R_1'$ and $R_2'$ are the gene expression values in control pre-R group in repeats 1 and 2, respectively; $S_1'$ and $S_2'$ are the gene expression values in control pre-S group in repeats 1 and 2, respectively. Only the sister-concordant genes were included in the rank-based signature.

The similarity of pre-resistant signatures and drug-induced consensus gene expression signatures were evaluated by the connectivity scores defined in the Connectivity Map (CMAP) project[74]. The analysis was performed by using the "connectivityScore" function (method = "fgsea", nperm = 1000) from PharmacoGx R package (1.3.1). The drugs with negative connectivity scores consistently with all of the six variants of pre-resistant signatures ($P$ value < 0.05) were selected as the top candidates (Supplementary Data 7). We filtered out drugs whose targets were not expressed or expressed at below 10% of Kuramochi cells, including pifithrin-mu, AM-404, and methenamine for carboplatin; PSI-7976 and cilengitide for olaparib; oxyphenbutazone and WAY-100635 for NK cells (Supplementary Data 8).

### Statistics & reproducibility
All data showed the mean +/− standard deviation (s.d.) of at least three biological replicates with the n indicated in each experiment, unless specifically indicated in figure legends. The statistical analyses were indicated in the legends of each figure, with $p < 0.05$ indicating a statistically significant difference. The statistical analysis was performed in R (>3.0).

### Reporting summary
Further information on research design is available in the Nature Portfolio Reporting Summary linked to this article.

## Data availability

All the data needed to evaluate the conclusions in the paper are available in the manuscript or the supplementary materials. All of the ReSisTrace and drug-perturbed scRNA-seq raw data have been deposited in the Gene Expression Omnibus (GEO) under accession code GSE223003. The data is publicly available. Source data are provided with this paper.

## Code availability

The source code for data analysis are available at GitHub (https://github.com/TangSoftwareLab/ReSisTrace) and Zenodo (https://doi.org/10.5281/zenodo.10418352).

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

## Acknowledgements

We thank FIMM High Content Imaging and Analysis unit for imaging services. BSL-2 laboratory for recombinant virus work was provided by Biomedicum Virus Core (HelVi-BVC) supported by Helsinki Institute of Life Science (HiLIFE) and the Faculty of Medicine, University of Helsinki, and Biocenter Finland. The flow cytometry analysis was performed at the HiLife Flow Cytometry Unit, University of Helsinki. Single-cell transcriptomics was performed at FIMM Single-Cell Analytics unit supported by HiLIFE and Biocenter Finland. The results are in part based upon data generated by the TCGA Research Network: https://www.cancer.gov/tcga. Cartoons in Figs. 3a and 4a were created with BioRender.com. This work was supported by The Sigrid Jusélius Foundation (A.V., J.T., A.F.), The Cancer Foundation Finland (A.V., S.Z., F.P.), Foundation for the Finnish Cancer Institute (K. Albin Johansson Cancer Research Fellowship for AV), The Academy of Finland projects 289059 (A.V.), 319243 (A.V.), 317680 (J.T.), 320131 (J.T.), 351165 and 351196 (J.T., A.V.), ERA PerMed JTC2020 PARIS/Academy of Finland project 344697 (A.V.), European Research Council No. 716063 (J.T.), HORIZON-MSCA-2021-PF project 101067835 (M.F.), University of Helsinki Research Foundation (J.D., S.Z., F.P.), K. Albin Johanssons stiftelse (S.Z.), Ida Montinin Säätiö (S.Z., F.P.), Biomedicum Helsinki Foundation (S.Z., F.P.), Cancer Society of Finland (A.F.), Academy of Finland (grant number 350396 to A.F.), European Union (ERC, SPACE 101076096). Open access funded by Helsinki University Library.

## Author contributions

Conceptualization and supervision: J.T. and A.V. Study design, data interpretation, and preparation of the manuscript: A.V., J.T., J.D., S.Z., M.F., and J.E. Execution of experiments: J.D., J.E., J.J., L.G., S.F., S.P. and J.B.; NK cell preparation: O.D., and K.S. Computational and statistical analysis: S.Z., J.T., M.F., A.A., Y.W., W.W. and F.P. Resources: A.V., J.T., S.M., L.K. and A.F. All authors wrote the paper.

## Competing interests

The authors declare no competing interests.
