## [Peer Review File · Nature Communications]

Tracing back primed resistance in cancer via sister cellsREVIEWER COMMENTS

Reviewer #1 (Remarks to the Author): expertise in single cell RNA-seq methods

The authors describe a labeling strategy that allows to track sister cells after cell division. This system is utilized to assess treatment surviving cell sister cells, termed 'pre-resistant' cells and pre-sensitive cells, which are the ones that did not survive after treatment. Furthermore, CNV profiles are inferred from single-cell data, gene-ontology analysis performed, lineage signature association with HRD state is determined and associated with clinical outcomes and finally, small molecules for pre-sensitive phenotype induction are determined using previously characterized pre-sensitive and pre-resistant profiles and the synergistic effect with the actual treatment is shown.

While the experimental design is interesting and worth an applause, unfortunately the manuscript is poorly written, results do not present novel insights, and analysis appears to be done by cherry-picking data to fit the story that authors try to sell. The main message of this work is that ReSisTrace can identify the unique transcriptional signatures, which can help designing/identifying the synergistic drugs to sensitize the cells. While the value of this is arguable, the manuscript often uses misleading and confusing statements (see comments below). It seems that ReSisTrace can only identify very generic drugs having a very broad effect on cellular functions (i.e. not being specific to particular biological process). Such drug candidates are easy to predict even without doing sophisticated studies; in principle almost any drug that affects basic cellular functions (e.g. ER function, mitochondria function, DNA replication, etc) will sensitize the cells. In other words, the findings of this work are obvious and lack novel insights. Therefore, with all due respect I cannot recommend this work for publication at this stage.

Criticism and comments:

Have you confirmed in any possible way whether or not all the cells that were used in the study were actually arrested at S phase due to cell cycle arrest with thymidine? How do you rule out a possibility that a fraction of cells might be unresponsive to thymidine or show delayed response?

The results of control conditions in majority of experiments follow treatment conditions results almost identically. Why? It seems these results indicate the weak effect overall.

When performing gene ontology analysis and association with HRD, control conditions are not included. This is needed to prove that the conclusions and statements of the manuscript are indeed specific to a certain condition, and do not apply to other conditions i.e. control.

Why out of NK cell treatment on BRCA-rescued cells only one donor NK cells had any effect? Furthermore, in discussion authors argue that the effect cannot be explained by HLA-C haplotype matches. The part about HRD and BRCA rescuing lacks experimental support.

It seems that the authors utilize UMAPs to derive conclusions that cannot be derived from UMAP coordinates alone. The observed shifts in most cases hint towards batch effect as opposed to biological response.

Line 88: Authors state that they performed additional experiments "to deconvolute the confounding effect of experiment-specific growth fitness from primed drug resistance." The exact conditions and details of this experiments are missing.

Line 95: by saying "before splitting" do you also mean "before drug treatment"?

Line 96-97: Authors are stating: "As expected, the majority of colonies before splitting the

samples consisted of two cells (Fig. S1c), whereas after splitting, most lineages were represented by only one cell (Fig. S1d). “

Results presented in Figure S1c, show that large part of lineages (~40%) before splitting are represented by only one clone. It is thus unclear how do authors differentiate whether lineages after harvesting are derived from two-cells and not from single-clones? Moreover, cells that formed clumps and clusters before splitting, even if they represent small fraction, after splitting will contribute significantly to the single-clone subpopulation because each said cluster would contain >4 cells before splitting.

Line 98-99: I could not understand how the schematics presented in Figure S1e, represents “Mathematical modelling and computer simulations” . Also, it is unclear how the presented schematics support the statement: “pre-sensitive cells captured by ReSisTrace were truly positive”

Line 100-102: I find following statements unsupported by the data: This suggests that the true pre-resistant cells can be captured with four to five-fold enrichment in the observed pre-resistant population compared to the observed pre-sensitive population, despite experimental uncertainties such as cell loss. Authors have not taken into account that before after splitting most of the clones will be derived from the pre-existing single-clones and from the cell clusters. Clones derived from 2-cells will constitute lower fraction and thus much higher enrichments might be needed to capture true pre-resistant cells.

Line 104: what do you consider “robust transcriptomic changes”?

Lines 108-121. Please be precise and quantitative. For example: Line 109: it would be more accurate to say that observed differences in your data sets a small but significant. E.g. “We observe small statistically significant differences”. Line 112: indicate how many times lower levels? e.g. “xxx-times lower levels”. Line 114: indicate how many fold higher? “xxx-times higher relative degradation rates” Line 115, “higher by what degree? Line 118 “high enriched”? By what degree? etc etc.

Lines 123-125. A UMAP embedding showed no distinct separation between the pre-resistant and pre-sensitive cells, while the post-treatment cells were shifted from the pre-treatment cells (Fig. 2a and Fig. S3a).

In Fig2a, where all treatments are combined such shift in UMAP coordinates is indeed evident, even though, critically, UMAP coordinates should not be used as a measure of similarity in any scenario. Moreover, in Figure S3a, where UMAPs for all conditions including control are shown separately, such “shift” in UMAP coordinates post-treatment is evident not only for all treatments, but to a very similar extent for the control conditions, indicating that the observed shift might be related to batch effect (e.g. most likely related to prolonged culture, and not biological treatment effect, as all samples post-treatment, including the control, show the shift in exact same direction and approximately the same coordinates). It is important to emphasize that UMAP coordinates should NOT be used to make conclusions in any scenario as they can be easily altered by varying parameters used for UMAP construction and are very sensitive to batch effect, which is most likely what is seen in all post-treatment samples. There is a high chance that if any kind of batch correction algorithm would be used the observed “differences” will disappear. And, if the post-treatment samples would form a completely separate cluster in UMAP space and no such cluster would be seen in control, in that case, a conclusion on treatment effect would hold.

Lines 126-128, related to Figure S3c.

Authors state that, as expected, lineage diversity is higher in pre-treatment samples. In figure S3C, Shannon diversity indexes for different treatment samples show reduced diversity post-treatment for all treatment conditions, however, the same trend to a very similar extent applies for both replicates of control. Moreover, the control replicates have difference in Shannon diversity between replicates in both pre-treatment and post-treatment conditions, i.e. pre-treatment control 1 has index of approximately 5.4 and control 2 has

index of approximately 5.75, and post-treatment control 1 drops to approximately 5, while control 2 drops to approximately 5.4, which is almost the same value as for control 1 pre-treatment. Similar difference between replicates is observed for all treatment conditions. How certain are the authors that the observed drop in diversity is indeed related to treatment, if control (no treatment) conditions show the same results? Why the pre-treatment diversity varies between replicates to the extent relative to the treatment effect? The control should have no difference pre-treatment and post-treatment, because there was no actual treatment for the control.

Line 130: It is not presented convincingly that CNV clones inferred from scRNA-Seq indeed matched UMAP clusters. Needs to be presented in a more clear way.

Lines 133-137. Authors state: These results suggested that subclonal selection is already evident in the pre-resistant cells. Specifically, we found that subclone A was significantly enriched in the pre-resistant cells compared to pre-sensitive cells at all the treatment conditions, especially for the carboplatin and olaparib treatments ($P = 8.882 \times 10E-11$ and $P < 2.2 \times 10E-16$, respectively, two- proportion Z-test) (Fig. 2c)

First, the sample sizes are vastly different between conditions: pre-sensitive $n = 19,540$, pre-resistant $n = 3,102$, and post-treatment $n = 7,802$ and looking at the barplots, the proportion difference could arise due to difference in sample size. Second, by definition the “pre-resistant” cells are the sister cells of the ones found in “post-treatment” samples, as stated in lines 74-45. The “pre-resistant” and “post-treatment” samples have identical composition – by definition there is no possible way the “pre-resistant” samples could have lineages not found in the “post-treatment” samples as the “pre-resistant” cells are the ones already found in “post-treatment” samples as per annotation strategy, thus, the identical composition before treatment (pre-resistant) and after treatment suggest that the treatment did not have any effect on the basis of CNV subclones, i.e. the treatment could have eradicated at least one clone and thus would be linked to subclonal selection, as not all clones survived the treatment. However, that is not the case, thus, inferring CNVs does not add any value to the manuscript.

Line 140: To my understanding the “connectivity score” does not represent “transcriptional signature”. This should to be clarified.

Line 142-143: “Subclone E showed a treatment-specific pattern, consistent with the enrichment patterns of the subclones in the pre-resistant populations“. There is no strong experimental evidence to make such claim.

Lines 155-157. Together, our results show that subtle pre-resistant features can reflect the subclonal phenotypes that are subsequently enriched during each treatment.

These are inadequate conclusions drawn based on gene ontology analysis on pre-resistant cells from different treatments. Considering that gene ontology terms include terms not necessarily applicable to the experimental context (i.e. terms “ossification”, “animal organ regeneration” in Figure 2e), the authors should include the same analysis performed on “pre-sensitive” and control samples. This way, the authors would verify that these “pre-resistant features” are indeed found and specific for pre-resistant samples. This analysis on control samples or ‘pre-sensitive’ cells is lacking.

Line 167-169. The statement is misleading. The survival analysis shows only very weak improvement.

Lines 171-172. We confirmed that the pre-sensitivity signatures in olaparib and carboplatin conditions were positively associated to the HRD signature

Figure 3c. the connectivity score median for carboplatin, Olaparib and NK treatment are approximately 0.25, 0.35 and 0.45 respectively. Is this value considered high? That is not clear for the reader. What is a base for “connectivity score” to be considered “pre-sensitivity signature”.? Please add the connectivity score for control cells. Also, as “pre-sensitive” signature association with HRD is portrayed in this figure, it would be beneficial to include

“pre-resistant” signature connectivity score with HRD signature – if the conclusions drawn from this analysis are true, only ‘pre-sensitive’ cells would have association with HRD.

Lines 195-196. We searched the L1000 database for such drugs that induce gene expression changes opposite to the pre-resistance signatures.

Why did the authors search for signatures opposite to pre-resistance signatures, instead of searching directly for pre-sensitive signatures they have determined previously?

Some comments related to Figures:

The value of Figure S1f is questionable, what does it teach us? Why X axis presents only very narrow window, it should display values from 0 to 100.

FigS1. X-axis. What do numbers 1 and 2 indicate?

You should indicate more clearly what concentrations of drugs, and count of NK was used to generate Fig S1.

Figure 1d. Please elaborate why transcriptional activity of sister-concordant and -disconcordant is similar? What does it teach us?

Figure 4c, lines 206-211. The UMAP projection shows that three proteasome inhibitors (bortezomib, delanzomib, and ixazomib) induced transcriptome profiles that were clearly clustered together and distinguished from the DMSO control (Fig. 4c, Fig. S6). In addition, pevonedistat, a neddylation inhibitor (REF 36), when applied in two distinct concentrations, clearly shifted the transcriptional profile of Kuramochi cells to a unique state.

DMSO treated cells are colored in grey, which is not at all visible in the UMAP with all conditions. Furthermore, if in the panels below “Pevonedistat hydrochloride”, “Ixazomib citrate” and “GW843682X” DMSO cells are portrayed in grey, condition “ixazomib citrate” is the only one clearly separating from DMSO, as Pevonedistat is intertwined with grey cells and such miniscule shift can be associated with batch effect, while the third treatment co-localizes with DMSO grey cells in the UMAP space almost ideally.

Material and Methods:

For each package used in the study please indicate all the parameters.

As a supplementary material please provide a full sequence in fasta format file, for the final Lentiviral Lineage Label construct.

Line 378. Please indicate what ratio of cells to lentivirus was used for transfection.

Line 383. Please indicate for how long cells were synchronized with thymidine.

Line 384. It is not clear how do authors wash away intracellular thymidine, and if not, then how to they ensure that intracellular thymidine does not affect further diversification of cells (e.g. cells that start to divide are those that were able to pump out the thymidine).

Line 385. Please indicate how many cells per well were obtained after 48h of incubation.

Line 385. Please indicate how many cycles cells underwent during 10, 6 and 7 days of incubation. It is not clearly explained what was the rationale of choosing 10, 6 and 7 days?

Line 440. What are the ratios of NK to cancer cells?

Lines 446, 465, 466. What are concentrations?

Line 452: What do you consider “standard procedures”?

Reviewer #2 (Remarks to the Author): expertise in ovarian cancer biology and treatment response

This manuscript is beautifully written with detailed methodology included.

The manuscript details the need for new high-resolution prediction of cells at a transcriptomic level which, prior to exposure to drug/other therapy (carboplatin, olaparib or NK cell killing), are already primed to be display phenotypic resistance. Further study of these features included different genetic contexts, such as seen clinically (e.g., HRD states

stemming from BRCA1 or 2 mutations). Further extension of these studies identified potential sensitizing agents to be used prior to therapeutic drug/other exposure as proof of principle, to revert identified transcriptional profiles to more sensitive ones. Lastly, there was a link found between sensitivity to innate immunity and HRD status, which adds to the well accepted understanding that HRD confers enhanced response to adaptive immunity.

The in vitro models use both ovarian endometrioid but mainly a high-grade serous cell line (BRCA2+). The need for cell cycle synchronization is stressed. Labeled sister cells are characterized to be representative clones of the untreated state.

Cells are exposed in vitro one time with drugs for 5 days, or NK cells for 24h. Doses were given which effect 70-80% killing of cells. The scRNA-seq transcriptomic profile of viable cells at that point determined definition of drug resistance. We know that repeated exposure can result in further killing of cells, however in this model, their immediate goal was achieved with this experimental design. However, drug resistance likely not only exists apriori, but develops upon repeated exposure, or exposure to different drugs along the continuum.

PARP inhibitors have shown most efficacy in HRD states (mainly as maintenance, not really treatment—see line 82), however has shown some efficacy as well in non-HRD patient populations. Suggest modifying this sentence.

Regarding the link between innate immunity and HRD stated to be new, I refer the authors to the following article (and others) for possible discussion: Pharmacologic induction of innate immune signaling directly drives homologous recombination deficiency. Lena J. McLaughlin, Lora Stojanovic, Aksinija A. Kogan, +8, and Feyruz V. Rassool. 117 (30) 17785-17795; <https://doi.org/10.1073/pnas.2003499117>, PNAS 2020. This area is important, and I believe underlies what is seen clinically, in that different patients have differing levels of innate immunity, leading to more or less control of their tumor growth/ quiescence. Unfortunately, in epithelial ovarian cancer, the levels of innate immunity are thought to be low in most cases.

Lastly, what do the authors see is the clinical translation of the in vitro sensitizing data presented? Inclusion of sequential sensitizing agents? If so, the drugs themselves optimally should be chosen to not add significant toxicity to the therapeutic combinations which follow. In addition, the duration of sensitizing therapy needs to be short as most ovarian cancer patients have advanced disease and need to get started on treatment as soon as possible after diagnosis.

Overall, this manuscript has high impact in the identification of pre-sensitive and resistant clones of cells which exist apriori, prior to primary treatment. The data raise the possibility that treatment of these cells with pre-sensitizing agents may help with response to primary therapy. The hope, would be that perhaps on follow-up, this approach may reduce the development of resistant cells, and demonstrate further downstream advantages.

Setsuko K. Chambers

Reviewer #3 (Remarks to the Author): expertise in computational transcriptomics and evolution

I have been involved to provide expert advice on the computational evolution aspects of this

work, which presents the idea of studying differential expression across sister cells. The notion of sister cell is linked to the type of experimental design (reSisTrace) proposed by the authors, which use barcodes to tag cells, cell-cycle synchronisation to phase division clocks, and harvest cells after 48 hours (~1 division) upon splitting pre-exposure and post-exposure cells. Then the authors use this design to scan for particular inhibitors and determine the resistance mechanisms linked to aneuploidy and HRD phenotypes in cell lines.

The paper is largely experimental and contains little computational parts to be evaluated, some of which deserving clarifications. Overall, I anticipate that the general merits of this work should be better assessed based on the relevance of the biological conclusions and the presented experimental design. The computational aspects of this work are:

- Mathematical modelling for reSisTrace: the authors claim to use a model to design reSisTrace, but they do not give any detail about the computations carried out in this “model”. I understand that they have some parameters mentioned in the Methods section (and Figure S1), but literally no description is given (a formula!) about how these parameters are used. The term simulation is also used, but without any immediate reference to what a simulator is in this case. I suspect that this mathematical model is a simple combinatorial equation that the authors use to determine some parameters of the assay. However, this is all left to my imagination, which should not be the case. This part should be clarified and properly framed, otherwise it is not possible to understand how much it contributes to the overall work — is this a real mathematical model?
- CNV inference: the authors use InferCNV to infer copy numbers from their data. This part is taken-as-it-is because I understand that it is not the main point of the paper to discuss the quality of that inference. I think that overall the results from InferCNV - which I do know well - should be reliable also because the subclones are very large in the input datasets. The authors might just perform some post-inference analysis (e.g, bootstrap) to determine the confidence in the InferCNV estimates.
- Connectivity scores: this metric is taken from another paper (ref [30]) and is therefore not discussed here.

Overall, a general decision on this manuscript should focus on how sister-resolution lineage tracing seems a novel approach to reveal complex cell states associated with treatment resistance in cancer.

Response to reviews for the manuscript “Tracing back primed resistance in cancer via sister cells”

We thank the reviewers for their insightful comments and suggestions. We have revised the manuscript according to the comments, which hopefully has made it more clear and compelling.

The major changes in the revised manuscript are:

- Added cell cycle analysis of thymidine block (Supplementary Fig. 1c-d) according to suggestion from Reviewer #1.
- Added a Supplementary Note and expanded the main text to improve the description of the performed simulations to address the comments raised by both Reviewer #1 and Reviewer #3.
- Added Leiden clustering analysis (Supplementary Fig. 4b) to address the criticism related to subclones and main transcriptional structures from Reviewer #1.
- Added data of HLA-C haplotypes of the utilised NK cells and ovarian cancer cell lines (Supplementary Data 5), and expanded the discussion on the context-specificity of NK killing to address comments raised by Reviewer #1.
- Added a separate plot for the DMSO control induced expression pattern to Fig. 4c according to the suggestion from Reviewer #1.
- Clarified the text related to the control condition specimens, and its use in the pre-resistance signatures, as well as the use of connectivity score as an association metric to address the comments raised by Reviewer #1.
- Expanded the discussion of the clinical use of the drugs predicted to overcome primed resistance to address the suggestion by Reviewer #2.

Detailed point-by-point responses to the comments (responses in red):

Reviewer #1 (Remarks to the Author): expertise in single cell RNA-seq methods

The authors describe a labeling strategy that allows to track sister cells after cell division. This system is utilized to assess treatment surviving cell sister cells, termed ‘pre-resistant’ cells and pre-sensitive cells, which are the ones that did not survive after treatment. Furthermore, CNV profiles are inferred from single-cell data, gene-ontology analysis performed, lineage signature association with HRD state is determined and associated with clinical outcomes and finally, small molecules for pre-sensitive phenotype induction are determined using previously characterized pre-sensitive and pre-resistant profiles and the synergistic effect with the actual treatment is shown.

While the experimental design is interesting and worth an applause, unfortunately the manuscript is poorly written, results do not present novel insights, and analysis appears to be done by cherry-picking data to fit the story that authors try to sell. The main message of this work is that ReSisTrace can identify the unique transcriptional signatures, which can help designing/identifying the synergistic drugs to sensitize the cells. While the value of this is arguable, the manuscript often uses misleading and confusing statements (see comments below). It seems that ReSisTrace can only identify very generic drugs having a very broad effect on cellular functions (i.e. not being specific to particular biological process). Such drug candidates are easy to predict even without doing sophisticated studies; in principle almost any drug that affects basic cellular functions (e.g. ER function, mitochondria function, DNA replication, etc) will sensitize the cells. In other words, the findings of this work are obvious and lack novel insights. Therefore, with all due respect I cannot recommend this work for publication at this stage.

We thank the reviewer for this critical perspective on our work, yet disagree that the findings are obvious and that all the sensitising drugs could have been predicted without analysing primed resistance. While protease inhibitors have been used in combination with platinum drugs even in the clinical trial setting (Ramirez et al. *Gynecol Oncol.* 108: 68-71 (2008)), their synergy with olaparib for ovarian cancer were not reported previously. To assess existing drug combinations in a systematic way, we retrieved the DrugComb database (<https://drugcomb.org>), which is currently the largest resource of drug combination data (Zheng et al, *Nucleic Acids Res.* 49:W174-W184 (2021)), and found that no data exist for the combination of olaparib and bortezomib/carfilzomib for ovarian cancer. We also found that the average synergy score for bortezomib combination (i.e. a combination that involves bortezomib and another drug) is close to zero, suggesting that the synergy of bortezomib is not ubiquitous (**Table R1** below, reproducible by the source code attached). We performed the same analysis for carfilzomib and carboplatin, and the results were similar. In contrast, the drug synergy scores in our results were much higher (**Fig. 4f, Supplementary Fig. 7b-c**), suggesting that our findings are indeed novel and significant. Furthermore, the neddylation inhibitor pevonedistat is not a common finding of drug predictions, and no data exists in DrugComb for pevonedistat in ovarian cancer. Thus, we would not have tested pevonedistat as a pre-treatment without it being predicted to drive cells to the opposite of the identified pre-resistant states.

Table R1. Drug synergy scores in ovarian cancer retrieved from DrugComb. Multiple synergy scores were reported as they are based on different mathematical models. NA stands for no data.

	ZIP	HSA	LOEWE	BLISS
Bortezomib + Olaparib	NA	NA	NA	NA
Bortezomib + Carboplatin	0.57	1.66	-7.46	1.17

Bortezomib + others	0.50	0.86	-10.32	0.38
Carfilzomib + Olaparib	NA	NA	NA	NA
Carfilzomib + others	NA	NA	NA	NA
Pevonedistat + Olaparib	NA	NA	NA	NA
Pevonedistat + Carboplatin	NA	NA	NA	NA
Pevonedistat + NK	NA	NA	NA	NA

More importantly, distinct from previous findings that are largely serendipitous and spontaneous, our method provides a novel systematic approach to rationalise the drug combination discovery, by interrogating the single cell transcriptomic signatures of drug resistance subpopulations of cancer cells. Regarding basic cellular functions, it is also important to note that while many of the predicted pre-treatment drugs did match the functions identified in the Gene Ontology (GO) analysis of the pre-resistance signatures, these sensitising drugs were predicted purely on the drug perturbed expression profiles from the L1000 database, not the GO or pathway analysis. Compared to the previous lineage tracing methodologies, such as the Watermelon system, our method is unique in its ability to successfully identify the primed drug resistant populations to establish their transcriptomic signatures. We further present a computational strategy to predict drug combinations to revert these transcriptomic states which we experimentally validate. To highlight the novelty both in methodological as well as biological discovery aspects, we have modified the text in the 'Discussion section' (new text in bold):

*"We applied a **systematic approach** of using drug-perturbed gene expression signatures³⁵ to predict pre-sensitising drugs, and validated that context-specific induction of pre-sensitive states was a prerequisite for effective pre-treatments."*

Criticism and comments:

Have you confirmed in any possible way whether or not all the cells that were used in the study were actually arrested at S phase due to cell cycle arrest with thymidine? How do you rule out a possibility that a fraction of cells might be unresponsive to thymidine or show delayed response?

We thank the reviewer for this suggestion. We have now performed the suggested experiment and added the thymidine block cell cycle analysis as new **Supplementary Fig. 1c-d**. It shows that 61% of cells are in S phase, 28% in G1 phase and 9% in G2/M phase after thymidine block, while without thymidine block, 29% are in S phase, 49% in G1 phase and 19% in G2/M phase.

Importantly, the protocol rather aims to maximise the proportion of sister cells have as many shared labels between pre- and post-treatment samples but is not dependent on total thymidine block, as also supported by computational modelling and simulations (see also the response to Reviewer #1 comment starting: Line 100-102). In our detailed analysis, the proportion of lineages with four cells in the pre-treatment samples was on average 0.015%, implicating that putative leakage from thymidine block -that would lead to lower resolution due to increased proportion of large progenies - is on an acceptable level. Furthermore, to mitigate the effect of these rare large progenies on the resolution, we have filtered out lineages that are represented with above four cells in the pre-treatment sample. Thus we can conclude that i) while thymidine block does not arrest all cells to S phase, it does enrich S phase cells on a sufficient level to analyse primed resistance.

To address the comment on delayed response, as well as a later question on possible remaining intracellular thymidine, we further measured cell cycle proportions 4 or 8 h after the release from the block. Here, cells restarted progressing through the cell cycle, where 23% and 41% of cells were in G2/M at 4 and 8 hours after the release, respectively. Thus, we can further conclude that ii) as the release from thymidine block leads to cell cycle progression into G2/M phase at 4 and 8 h following the release, while in the unreleased specimen cell cycle proportions stay the same, the cells do not show a delayed response to thymidine.

To present the new thymidine block results, we have added the following sentence to the 'Results' section:

"Cell synchronisation with thymidine block increased the proportion of S phase cells from 29% in control cells to 61%, whereas 8 hours after release from thymidine block the proportion of S phase cells was only 19% (Supplementary Fig. 1c-d)."

And modified the figure and legend of Supplementary Fig. 1c-d as follows:

(c) Cell cycle phase proportions after 42 hours thymidine block, and 4 or 8 hours following release or control without release, and **(d)** respective histograms from flow cytometry.

The results of control conditions in majority of experiments follow treatment conditions results almost identically. Why? It seems these results indicate the weak effect overall.

We thank the reviewer for this notion and agree that the control condition consistently shows similar changes to those seen in the treatments. We acknowledge this in our manuscript repeatedly, stating that the control experiment presents consistent selection pressure. This highlights the need to remove this experimental fitness effect from further analyses as we have done for all analyses of the pre-resistance signatures (presented in Figures 2 to 4).

When performing gene ontology analysis and association with HRD, control conditions are not included. This is needed to prove that the conclusions and statements of the manuscript are indeed specific to a certain condition, and do not apply to other conditions i.e. control.

We thank the reviewer for this important notion. We have in fact removed the effect of the control condition from pre-resistance signatures against carboplatin, olaparib, or NK cells, and described this in the 'Methods' section but unfortunately failed to communicate this clearly in the main text. We have now added a sentence indicating this (new text in bold):

*“With the identity of pre-resistant and pre-sensitive cells, as well as the identity of subclones, we determined rank-based transcriptomic signatures for pre-resistance and subclones (Supplementary Data 2). **To remove the effect of experimental fitness from further analyses, pre-resistance signatures were adjusted to the control condition by subtracting the control pre-survival signal from the pre-resistance signals.**”*

Why out of NK cell treatment on BRCA-rescued cells only one donor NK cells had any effect? Furthermore, in discussion authors argue that the effect cannot be explained by HLA-C haplotype matches. The part about HRD and BRCA rescuing lacks experimental support.

First, while we found that the effect of BRCA rescuing was context dependent, it is not true that only one donor had any effect. In the ReSisTrace experiment performed in Kuramochi cells, we used NK cells from Donor 1, and thus our prediction of HRD cells being vulnerable to NK cells is based on this specific context. Indeed, NK cells from this donor were consistently less effective in killing BRCA1/2 restored cells in all the three isogenic cell line pairs. We further assessed the effect of HRD to NK killing in distinct NK donor and ovarian cancer cell line combinations, wherein BRCA2 restored Kuramochi cells were significantly more resistant against all three donors' NK cells, yet the BRCA1 restored COV362 and UWB1.289 cells were not. To more clearly acknowledge the context-dependency, we have now changed the respective 'Results' section subheader to:

“Context dependent association between HRD status and susceptibility to NK killing”

Regarding the haplotype match statement, we acknowledge that we only referred to it in the 'Discussion' section, without showing the actual results in the 'Results' section. We have now added a new **Supplementary Data 5** to show the haplotype matches for the combinations between NK donors and ovarian cancer cell lines, and added the following sentence to describe it in the 'Results' section:

"In the non-autologous context, variations in the HLA-C type can influence the interaction between NK cells and target cells³¹, but the differences of HLA-C haplotype matches could not explain the inconsistency between Kuramochi and other cell lines (Supplementary Data 5)."

Supplementary Data 5: HLA-C haplotypes of the utilised NK cells and ovarian cancer cell lines

Sample	Donor 1	Donor 2	Donor 3	Donor 4	Kuramochi	COV362	UWB1.289
HLA-C haplotypes	C1/C1	C1/C2	C1/C1	C2/C2	C1/C1	C1/C1	C1/C1

To further speculate the reasons for context-dependent effects, we have expanded the text addressing this in the 'Discussion' section (new text in bold):

*"However, other NK donor combinations did not show aligned responses in additional BRCA1 restored isogenic cell line pairs, and such discrepancy was not explained by differences in HLA-C haplotype matches. **It is also known that some cell lines are more sensitive to TRAIL, TNF or Fas ligand mediated, and some more to granzyme or perforin mediated killing^{51,52}. Thus, one putative reason for the inconsistency between Kuramochi and other cell lines could be that the effect of HRD is more relevant in some of these NK-induced apoptotic mechanisms than others.**"*

Finally, if the last part of this comment is referring to the validity of the BRCA restored models themselves, we show RAD51 foci in cyclin A2 positive cells after ionising radiation as a functional readout of homologous recombination repair capability in the three BRCA1/2 restored cell lines we have used, two of which we created for this paper (BRCA2 restored Kuramochi, BRCA1 restored COV362). The assay is the gold standard for HRD functional testing (Cruz et al. Ann Oncol. 29:1203-1210 (2018); Tumati et al. Clin Cancer Res. 24:4482-4493 (2018); Meijer et al. Oncogene. 41:3498-3506 (2022)).

It seems that the authors utilize UMAPs to derive conclusions that cannot be derived from UMAP coordinates alone. The observed shifts in most cases hint towards batch effect as opposed to biological response.

We agree that two-dimensional UMAP projection coordinates alone should not be used to quantify differences between multidimensional data points such as single cell expression

profiles. However, UMAP projection does preserve overall data structure better than projections produced with other dimensionality reduction methods (Becht et al. Nat Biotechnol 37: 38–44 (2019)) and is hence commonly used in the field of single cell transcriptomics to visualise overall transcriptomic structures within studied data sets. This is also the basis for our use of UMAPs in the manuscript: visualising the overall transcriptomic structure of the data. We do not derive any quantitative results from UMAP coordinates, nor do we base any conclusions solely on the UMAP projections. For example, we assessed sister cell similarity using euclidean distance, and used the UMAP projection only as a visual example.

We address the separate points related to UMAPs below as follows: i) the batch effects when answering the concerns related to Lines 123-125, ii) conclusions of UMAPs, subclones and transcriptional clustering on the comment on Line 130, and iii) drug induced profiles in UMAPs on comment to Figure 4c, lines 206-211. In each of these points, we have further modified the main text to clarify the interpretations made based on the UMAP projections and other available data.

Line 88: Authors state that they performed additional experiments “to deconvolute the confounding effect of experiment-specific growth fitness from primed drug resistance.” The exact conditions and details of this experiments are missing.

This statement refers to the control condition, details of which we specify in the Methods section. For the control condition, cells were seeded back for an unperturbed cell culture for 13 days, as for olaparib the cells were treated for 7 days and allowed to recover for 6 days, and for carboplatin treated for 3 days and allowed to recover for 10 days. To make the terminology more clear, we rephrased the sentence as follows (new text in bold):

*“We also included a non-treatment control **condition** that mimicked the splitting, re-plating, and growth conditions of the drug treatment experiments, allowing us to **remove** the confounding effect of experiment-specific growth fitness from primed drug resistance **signals**.”*

Line 95: by saying “before splitting” do you also mean “before drug treatment”?

Yes, the splitting of the sample is performed after one cell division in the pre-treatment sample. To make this more clear, we have now modified the text as follows (new text in bold):

*“To reveal the cell states that are primed to treatment resistance, uniquely labelled cells are synchronised and allowed to divide once, after which **sample is split so that** half of the cells are analysed by scRNA-seq, while the other half undergoes anti-cancer treatment (Fig. 1a).”*

*“As expected, the majority of colonies before splitting the **pre-treatment** samples consisted of two cells (Supplementary Fig. 1e-f), whereas after splitting, most lineages were represented by only one cell (Supplementary Fig. 1g).”*

Line 96-97: Authors are stating: “As expected, the majority of colonies before splitting the samples consisted of two cells (Fig. S1c), whereas after splitting, most lineages were represented by only one cell (Fig. S1d). “

Results presented in Figure S1c, show that large part of lineages (~40%) before splitting are represented by only one clone. It is thus unclear how do authors differentiate whether lineages after harvesting are derived from two-cells and not from single-clones? Moreover, cells that formed clumps and clusters before splitting, even if they represent small fraction, after splitting will contribute significantly to the single-clone subpopulation because each said cluster would contain >4 cells before splitting.

First, to acknowledge that non-divided cells, jointly with cell losses, will contribute to false negative pre-resistant cells, we have modified the text as follows (new text in bold):

“For simplicity, cells with labels missing from post-treatment samples were annotated as ‘pre-sensitive’ whilst they inherently also contain false negative pre-resistant cells due to inevitable cell loss, and cells that failed to divide after release from thymidine block.”

Secondly, while it is true that large clones contribute more to total cell numbers than to the number of lineages, these clones are indeed so rare (4 cells: 0.015% of colonies; no colonies in images with above 4 cells) that they only constitute a negligible proportion of the total cell numbers in samples before splitting. This observed lack of large colonies also argues against having clumps and clusters of cells before splitting. To further mitigate the effects of large progenies we have i) excluded progenies with >4 cells in pre-treatment sample from all data analyses, and ii) extracted pre-resistance signatures only from pre-resistant/pre-sensitive lineages that are represented by one cell in pre-treatment sample, as described in the ‘Methods’ section.

Furthermore, the measured uneven doubling of cells prior splitting the samples was taken into account in the computational simulation in the parameter ‘LineagePropAfterDoubling’ as described in the new Supplementary Note.

Line 98-99: I could not understand how the schematics presented in Figure S1e, represents “Mathematical modelling and computer simulations” . Also, it is unclear how the presented schematics support the statement: “pre-sensitive cells captured by ReSisTrace were truly positive”

Figure S1e (in previous version; moved to Supplementary Note in the revised manuscript) showed the steps for the computer simulations that mimic the lineage tracing experiments. We agree that the description of the simulation was too limited and have now added a Supplementary Note, and expanded the text as follows (new text in bold):

“To assess the accuracy of pre-sensitive cell labels while taking into account experimental uncertainties, we performed computational modelling and simulations where we incorporated experimentally measured parameters for uneven cell

doubling before the sample was split, random distribution of sister cells during the split, as well as various forms of cell loss during the experimental procedure and analysis (See Supplementary Note for further details). The simulation showed that on average more than 80% of pre-sensitive cells captured by ReSisTrace were truly positive (Supplementary Fig. 1h, Supplementary Note and Supplementary Data 1) at the killing rate selected for the experiment (Supplementary Fig. 1i). ”

Line 100-102: I find following statements unsupported by the data: This suggests that the true pre-resistant cells can be captured with four to five-fold enrichment in the observed pre-resistant population compared to the observed pre-sensitive population, despite experimental uncertainties such as cell loss. Authors have not taken into account that before after splitting most of the clones will be derived from the pre-existing single-clones and from the cell clusters. Clones derived from 2-cells will constitute lower fraction and thus much higher enrichments might be needed to capture true pre-resistant cells.

We thank the reviewer for bringing this important notion to our attention. First, the estimate of four to five fold enrichment of capturing pre-resistant cells in the observed pre-resistant population is based on the simulation where on average more than 80% of pre-sensitive cells are true positive, i.e. less than 20% are false negative pre-resistant cells, compared to 100% true positive in the observed pre-resistant population, giving the stated enrichment.

Secondly, we implemented a parameter called “LineagePropAfterDoubling” in the simulation to mimic the uneven division of cells in the doubling stage. This parameter was set to mimic the experimental observation that 53.04% cells were doubled, 4.5% cells were quadrupled, and 42.46% of the cells remained as singletons. To estimate the upper bound of the pre-sensitive predictive rate, “LineagePropAfterDoubling” was set as “doubled = 100%, quadrupled = 0% and singleton = 0%” to mimic the perfect doubling condition. We have now added a Supplementary Note to provide more details about the parameter settings in the simulation, and modified the main text as described in the previous answer.

Line 104: what do you consider “robust transcriptomic changes”?

We refer to the highly consistent transcriptomic changes between the two replicates, analogous to the treatment conditions mentioned in the previous sentence. To make this more clear, we have now changed the text (new text in bold):

“Notably, we observed consistent transcriptomic changes also in the non-treatment control condition between pre-surviving and pre-extinct cells, suggesting that the experimental procedures themselves pose a non-random selection pressure which should be corrected.”

Lines 108-121. Please be precise and quantitative. For example: Line 109: it would be more accurate to say that observed differences in your data sets a small but significant.

E.g. “We observe small statistically significant differences”. Line 112: indicate how many times lower levels? e.g. “xxx-times lower levels”. Line 114: indicate how many fold higher? “xxx-times higher relative degradation rates” Line 115, “higher by what degree? Line 118 “high enriched”? By what degree? etc etc.

We agree with the reviewer for the need to be precise and quantitative, and have now modified the text accordingly as follows (new text in bold):

*“To validate the use of sister cells as proxies, we confirmed that cells with shared labels, i.e. putative sister cells, have significantly more similar transcriptomes than random pairs of cells (Fig. 1b-d; **average Euclidean distances in random cell pairs are 46% higher than those of sister cell pairs**). We further compared the genes that showed significantly higher similarity within sister cell pairs, termed ‘sister-concordant genes’, against the other genes, termed ‘sister-discordant genes’ (Fig. 1e). The sister-discordant genes were expressed at lower levels, along with increased drop-out noise ($P < 2.2 \times 10^{-16}$, two-tailed t-test; **mean expression of the sister-discordant genes being 3.4% of that of the sister-concordant genes**) (Supplementary Fig. 2a). Sister-discordant genes also showed **on average 49% higher relative degradation rates** ($P = 4.2 \times 10^{-10}$, two-tailed t-test) and **on average 65% higher splicing rates relative to transcription** ($P = 0.000038$, two-tailed t-test) (Fig. 1f and Supplementary Fig. 2b-c), suggesting that transcripts to be degraded show higher expressional variation.”*

Lines 123-125. A UMAP embedding showed no distinct separation between the pre-resistant and pre-sensitive cells, while the post-treatment cells were shifted from the pre-treatment cells (Fig. 2a and Fig. S3a).

In Fig2a, where all treatments are combined such shift in UMAP coordinates is indeed evident, even though, critically, UMAP coordinates should not be used as a measure of similarity in any scenario. Moreover, in Figure S3a, where UMAPs for all conditions including control are shown separately, such “shift” in UMAP coordinates post-treatment is evident not only for all treatments, but to a very similar extent for the control conditions, indicating that the observed shift might be related to batch effect (e.g. most likely related to prolonged culture, and not biological treatment effect, as all samples post-treatment, including the control, show the shift in exact same direction and approximately the same coordinates). It is important to emphasize that UMAP coordinates should NOT be used to make conclusions in any scenario as they can be easily altered by varying parameters used for UMAP construction and are very sensitive to batch effect, which is most likely what is seen in all post-treatment samples. There is a high chance that if any kind of batch correction algorithm would be used the observed “differences” will disappear. And, if the post-treatment samples would form a completely separate cluster in UMAP space and no such cluster would be seen in control, in that case, a conclusion on treatment effect would hold.

We thank the reviewer for this notion and agree that the shift in the post-treatment samples, including the control condition, can be due to a batch effect that most likely results from prolonged culture as suggested by the reviewer. However, as we use post-

treatment samples only to derive the identities of resistant lineages and subclones, potential or even likely batch effects in them do not affect the conclusions presented in the manuscript. Thus, commenting on the potential transcriptomic shifts in post-treatment samples only adds confusion and we have removed it from the text, as follows:

~~"A UMAP embedding showed no distinct separation between the pre-resistant and pre-sensitive cells, while the post-treatment cells were shifted from the pre-treatment cells"~~

Lines 126-128, related to Figure S3c.

Authors state that, as expected, lineage diversity is higher in pre-treatment samples. In figure S3C, Shannon diversity indexes for different treatment samples show reduced diversity post-treatment for all treatment conditions, however, the same trend to a very similar extent applies for both replicates of control. Moreover, the control replicates have difference in Shannon diversity between replicates in both pre-treatment and post-treatment conditions, i.e. pre-treatment control 1 has index of approximately 5.4 and control 2 has index of approximately 5.75, and post-treatment control 1 drops to approximately 5, while control 2 drops to approximately 5.4, which is almost the same value as for control 1 pre-treatment. Similar difference between replicates is observed for all treatment conditions. How certain are the authors that the observed drop in diversity is indeed related to treatment, if control (no treatment) conditions show the same results? Why the pre-treatment diversity varies between replicates to the extent relative to the treatment effect? The control should have no difference pre-treatment and post-treatment, because there was no actual treatment for the control.

We thank the reviewer for this notion and agree that the control condition also in this analysis shows similar changes to those seen in the treatments. This shows that selection pressure is present also in the control experiment, highlighting the need to remove these experimental fitness effects from further analyses as we have done for the following results, presented in Figures 2 to 4.

To acknowledge this better in the text, we have now modified the text as follow (new text in bold):

*"As expected, all the post-treatment samples showed larger clone sizes (Supplementary Fig. 3b) ($P < 2.2 \times 10^{-16}$, two-tailed Kolmogorov-Smirnov test) and smaller lineage diversity compared to the pre-treatment samples (Supplementary Fig. 3c) ($P = 3.80 \times 10^{-5}$, paired t -test, $n = 8$). **This was true also for the control condition, further highlighting the need to adjust the pre-resistance signals with the experimental fitness signal.**"*

Line 130: It is not presented convincingly that CNV clones inferred from scRNA-Seq indeed matched UMAP clusters. Needs to be presented in a more clear way.

To address the similarity of transcriptomic clustering and inferred subclones, we have now used unsupervised Leiden clustering for the transcriptomic data and compared the resulting clusters to the inferred CNV subclones. The adjusted Rand index between these clusters was 0.83, exemplifying high correspondence. We have now added a new

Supplementary Fig. 4b to display Leiden clustering on the UMAP and modified the text as follows (new text in bold):

*“To assess the putative role of genetic heterogeneity, we inferred copy number variations (CNVs) from the scRNA-seq data, and identified five subclones that closely matched the **unsupervised Leiden** clustering in the UMAP (Fig. 2b, **Supplementary Fig. 4a**, and **Supplementary Fig. 4b**; **adjusted Rand index 0.83**).”*

Updated figure and the new legend of **Supplementary Fig. 4b**:

(b) Unsupervised Leiden clustering of transcriptomic profiles projected on the UMAP; adjusted Rand index between transcriptomic clusters and inferred subclones is 0.83.

Lines 133-137. Authors state: These results suggested that subclonal selection is already evident in the pre-resistant cells. Specifically, we found that subclone A was significantly enriched in the pre-resistant cells compared to pre-sensitive cells at all the treatment

conditions, especially for the carboplatin and olaparib treatments ($P = 8.882 \times 10^{-11}$ and $P < 2.2 \times 10^{-16}$, respectively, two- proportion Z-test) (Fig. 2c)

First, the sample sizes are vastly different between conditions: pre-sensitive $n = 19,540$, pre-resistant $n = 3,102$, and post-treatment $n = 7,802$ and looking at the barplots, the proportion difference could arise due to difference in sample size. Second, by definition the “pre-resistant” cells are the sister cells of the ones found in “post-treatment” samples, as stated in lines 74-45. The “pre-resistant” and “post-treatment” samples have identical composition – by definition there is no possible way the “pre-resistant” samples could have lineages not found in the “post-treatment” samples as the “pre-resistant” cells are the ones already found in “post-treatment” samples as per annotation strategy, thus, the identical composition before treatment (pre-resistant) and after treatment suggest that the treatment did not have any effect on the basis of CNV subclones, i.e. the treatment could have eradicated at least one clone and thus would be linked to subclonal selection, as not all clones survived the treatment. However, that is not the case, thus, inferring CNVs does not add any value to the manuscript.

We thank the reviewer for this question. First, the difference in sample size has limited effect on the subclonal proportion difference. We performed downsampling 1000 times on each treatment group with an equal size of 500 cells. The subclonal proportions were plotted in Fig. R1a with error bars. The subclonal proportion pattern is similar to the pattern in Fig. 2c for the manuscript. We further compared the proportion of each CNV subclone in the downsampled pre-sensitive, pre-resistant and post-treatment samples (Fig. R1b). The result still supports the conclusion that “subclone A was significantly enriched in the pre-resistant cells compared to pre-sensitive cells at all the treatment conditions, especially for the carboplatin and olaparib treatments”.

Fig. R1 Subclonal proportions in pre-sensitive, pre-resistant, and post-treatment populations for different treatments. (a) The subclonal proportions from the downsampling data ($n = 500$ for each group). Mean values of subclonal proportions in each group are plotted. The error bar indicates the standard deviation from 1000 iterations. (b) The subclonal proportions from the downsampling data. The error bar indicates the standard deviation from 1000 iterations. **** $P < 1 \times 10^{-4}$, two-tailed t-test, $n = 1000$.

To address the second point: while no subclones were eradicated by the treatments, there are differences in their proportions in pre-resistant populations (as indicated above), reflected also in the post-treatment samples. Hence, we do not agree with the notion that the treatment did not have any effect on the basis of CNV subclones. While variation in

recovery times and rates of re-initiated proliferation could in principle have deviated the subclonal proportions in the post-treatment samples from those seen in the respective pre-resistant populations, we agree that their shared genetic origin means that their subclonal composition should be very similar. To make this clearer, we have modified the text accordingly (new text in bold):

*“These results suggested that subclonal selection is already evident in the pre-resistant cells, **as would be expected based on their shared genetic origin with post-treatment clones.**”*

Line 140: To my understanding the “connectivity score” does not represent “transcriptional signature”. This should to be clarified.

Line 142-143: “Subclone E showed a treatment-specific pattern, consistent with the enrichment patterns of the subclones in the pre-resistant populations“. There is no strong experimental evidence to make such claim.

First, it is exactly correct that the connectivity score does not represent a transcriptional signature. Connectivity score is a measure of association and our phrasing was imprecise as indicated by the reviewer.

Regarding the statement about subclone E, we agree that the statement is ambiguous.

To address both comments, we have modified the text as follows:

*“We found that the signature of subclone **A and E were positively associated** with the pre-resistance signatures (Fig. 2d; **associations estimated by the connectivity scores defined in²³**), while signatures of subclones B, C or D **showed a negative association.**”*

Lines 155-157. Together, our results show that subtle pre-resistant features can reflect the subclonal phenotypes that are subsequently enriched during each treatment.

These are inadequate conclusions drawn based on gene ontology analysis on pre-resistant cells from different treatments. Considering that gene ontology terms include terms not necessarily applicable to the experimental context (i.e. terms “ossification”, “animal organ regeneration” in Figure 2e), the authors should include the same analysis performed on “pre-sensitive” and control samples. This way, the authors would verify that these “pre-resistant features” are indeed found and specific for pre-resistant samples. This analysis on control samples or ‘pre-sensitive’ cells is lacking.

As we have explained above, the pre-resistance signatures applied here have been compiled by subtracting the signal from the control condition (mean log fold changes between pre-survival and pre-extinct cells) from the signal of each treatment condition (mean log fold changes between pre-resistant and pre-sensitive cells of each treatment condition). Hence, in the comparison shown in Fig. 2e, there would not be any signal left in the control condition, as the remaining log fold changes in the control sample will be all

zeros. As the pre-sensitive cells are used as a contrast to pre-resistant cells, each pre-sensitivity signature is simply the opposite signature of the corresponding pre-resistance signature. To avoid confusion, we have now removed the term 'pre-sensitivity signature' from the manuscript, and only use the term 'pre-resistance signature' (see below).

What comes to the gene ontology terms that are not necessarily applicable to the experimental context, it is a core feature of these terms that exist to unify gene attributes across distinct organisms and contexts. These 'non-fitting' annotations typically indirectly refer to more general processes; for example, ossification combines bone morphogenetic protein (BMP), sonic-hedgehog and Notch signalling that are also important in cancer resistance; and "animal organ regeneration" refers to stem cells and plasticity.

Line 167-169. The statement is misleading. The survival analysis shows only very weak improvement.

The Kaplan-Meier analysis shows a survival benefit with a p-value of 0.009, which is highly significant. We have now added the p value and hazard ratio (HR) to the main text:

"Kaplan-Meier survival analysis confirmed that HRD signature was associated with improved overall survival (OS) in ovarian serous cystadenocarcinoma patients in The Cancer Genome Atlas (TCGA) cohort (hazard ratio = 1.51, P = 0.009, log-rank test) (Supplementary Fig. 5b)."

Lines 171-172. We confirmed that the pre-sensitivity signatures in olaparib and carboplatin conditions were positively associated to the HRD signature
Figure 3c. the connectivity score median for carboplatin, Olaparib and NK treatment are approximately 0.25, 0.35 and 0.45 respectively. Is this value considered high? That is not clear for the reader. What is a base for "connectivity score" to be considered "pre-sensitivity signature"? Please add the connectivity score for control cells. Also, as "pre-sensitive" signature association with HRD is portrayed in this figure, it would be beneficial to include "pre-resistant" signature connectivity score with HRD signature – if the conclusions drawn from this analysis are true, only 'pre-sensitive' cells would have association with HRD.

Firstly, we'd like to thank the reviewer for allowing us to make the manuscript more stream-lined. As stated above, 'pre-sensitivity signature' is simply the inverse of pre-resistance signature. As multiple comments from Reviewer #1 clearly demonstrate, introducing this term adds unnecessary confusion to the manuscript. Thus we have used only 'pre-resistance signature' in the updated manuscript to simplify the terminology, and modified the text as follows (new text in bold):

*"We then compared the HRD signature to the **control adjusted pre-resistance** signatures captured by ReSisTrace. We confirmed that the **pre-resistance** signatures in olaparib and carboplatin conditions were **negatively** associated to the HRD signature, as expected based on their mechanisms of action that target HRD (Fig. 3e). More interestingly, the **pre-***

resistance signature for the NK condition showed even more pronounced negative connectivity scores to the signature (Fig. 3e)."

Secondly, with regards to the values of connectivity scores, they can vary within a range from -1 to 1, analogous to other measures of association such as correlation. In the updated manuscript where the HRD signature is compared to the pre-resistance signatures, the values are negative. We consider these values to be intermediate, but prefer not to state subjective views on their strength in the manuscript.

Third, as also stated above, connectivity score is a similarity metric, and a totally separate concept of pre-sensitivity signature. To make this difference and the terminology more clear for the reader, we have now modified the Fig. 3c and its legend as follows:

“(e) Association between the pre-resistance gene signatures and the HRD signature displayed as connectivity scores.”

Lines 195-196. We searched the L1000 database for such drugs that induce gene expression changes opposite to the pre-resistance signatures.

Why did the authors search for signatures opposite to pre-resistance signatures, instead of searching directly for pre-sensitive target signatures they have determined previously?

This question relates to the terminology issue that is addressed in our response above.

Some comments related to Figures:

The value of Figure S1f is questionable, what does it teach us? Why X axis presents only very narrow window, it should display values from 0 to 100.

We thank the reviewer for this important notion. In the actual experiments, we controlled the killing rate of the anti-cancer treatment to be between 70% and 80%, to collect enough resistant cells for sequencing analysis. When performing the computational simulation we set the "KillRateTotal" parameter ranging from 70% to 80% to match the experimental setting. We have now added a Supplementary Note to give more details about the parameter settings in the computational modelling and computer simulation.

FigS1. X-axis. What do numbers 1 and 2 indicate?

We understand this comment to refer to the x-axis of subpanels a and b within Supplementary Fig. 1 and address that here. The numbers refer to repeats within each condition, and to make that clearer, we have modified the legend as follows (new text in bold):

*"Supplementary Fig. 1 Cell doubling, lineage labelling efficiency, and quality assessment of ReSisTrace. (a) Percentage of cells with the lineage barcodes detected in each sample; **numbers 1 and 2 refer to repeats within each condition.** (b) Proportion of sample-shared or sample-unique barcodes in each pre-treatment sample "*

You should indicate more clearly what concentrations of drugs, and count of NK was used to generate Fig S1.

We have now modified the legend as follows (new text in bold):

*"(j) Consistency of gene expression changes for pre-resistant against pre-sensitive cells between the replicates of each treatment condition: **carboplatin (1.2 μ M, 3 days), olaparib (1.2 μ M, 7 days), and NK cells (26,000 NK cells to 16 000 cancer cells, 1 day).** (Carboplatin n = 30,613; Olaparib n = 30,586 ; NK n = 30,613; Control n = 27,478)"*

Figure 1d. Please elaborate why transcriptional activity of sister-concordant and -discordant is similar? What does it teach us?

To avoid expression level-associated biases when comparing splicing and degradation rates, we chose an expression bin (indicated in Supplementary Fig. 2a) where sister-concordant genes and sister-discordant genes are equally represented for this analysis. To make this more clear, we have now modified the figure legend as follows (new text in bold):

*"(f) Inferred splicing and degradation rates of **expression matched** sister-concordant and sister-discordant **gene sets, with 919 and 107 genes, respectively.**"*

Figure 4c, lines 206-211. The UMAP projection shows that three proteasome inhibitors (bortezomib, delanzomib, and ixazomib) induced transcriptome profiles that were clearly clustered together and distinguished from the DMSO control (Fig. 4c, Fig. S6). In addition, pevonedistat, a neddylation inhibitor (REF 36), when applied in two distinct concentrations, clearly shifted the transcriptional profile of Kuramochi cells to a unique state. DMSO treated cells are colored in grey, which is not at all visible in the UMAP with all conditions. Furthermore, if in the panels below “Pevonedistat hydrochloride”, “Ixazomib citrate” and “GW843682X” DMSO cells are portrayed in grey, condition “ixazomib citrate” is the only one clearly separating from DMSO, as Pevonedistat is intertwined with grey cells and such miniscule shift can be associated with batch effect, while the third treatment co-localizes with DMSO grey cells in the UMAP space almost ideally.

We have now added a UMAP showing only the DMSO condition to **Fig. 4c**. Regarding the potential batch effect, there seems to be a misunderstanding in how these samples were processed. We treated and analysed all the drug- or DMSO perturbed samples in duplicates, so that all treatments, including DMSO, and sample collection, were performed simultaneously in one batch, and we further pooled the samples by cell multiplexing method before single-cell RNA-seq sample processing to minimise any batch effects. For pevonedistat, we further applied two different concentrations, both of which resulted in expression profiles that did not overlay with the DMSO control profiles. Hence, the batch effect is an unlikely explanation of the observed difference.

To further clarify our sample processing and the conclusions that can be made based on the UMAP projections, we have modified the main text as follows (new text in bold):

*“For the first step, we generated drug-induced transcriptome profiles in Kuramochi cells through scRNA-seq, **with two replicates of each treatment and all samples processed simultaneously to minimise batch effects**. The UMAP projection shows that three proteasome inhibitors (bortezomib, delanzomib, and ixazomib) induced transcriptome profiles that were clearly clustered together and distinguished from the DMSO control (Fig. 4c-d, Supplementary Fig. 6). In addition, **the expression profiles resulting from treatment with pevonedistat, a neddylation inhibitor³⁷, when applied in two distinct concentrations, did not overlay with DMSO control treated cells’ profiles on the UMAP projection.**”*

And in the ‘Methods’ section as follows (new text in bold):

*“Kuramochi cells were seeded in 6-well plates at around 50% confluence overnight. Predicted small molecules (**see concentrations from Supplementary Data 11**) or **DMSO as control** were added for 24h in duplicates. Then cells were **simultaneously** washed and detached from plates. **To further minimise the batch effect**, each sample was individually multiplexed with 3' CellPlex Kit Set A (10x Genomics, PN-1000261). After labelling and washing, every 12 samples were pooled and then loaded for scRNA-seq experiments according to 10x Genomics instructions.”*

Material and Methods:

For each package used in the study please indicate all the parameters.

The applied parameters, or the use of default parameters, are now added to the 'Methods' section.

As a supplementary material please provide a full sequence in fasta format file, for the final Lentiviral Linearge Lable construct.

This is an important point, and we thank the reviewer for pointing this out and have now added the sequence as Supplementary Data 12.

Line 378. Please indicate what ratio of cells to lentivirus was used for transfection.

We did indicate the multiplicity of infection (MOI) value in the 'Methods' section but to make this clearer, we have now modified the description as follows:

*"Lentiviral p24 titer testing was performed by Biomedicum Virus Core (HeVi-BVC) using Alliance HIV-1 P24 ANTIGEN ELISA Kit (PerkinElmer, NEK050B001KT). For transduction, 320 µl of pBA439_UMI20 lentivirus library (3.25*10⁶ pg/ml) were used to transduce 6 million Kuramochi cells to achieve MOI between 0.2 and 0.25"*

Line 383. Please indicate for how long cells were synchronized with thymidine.

We have added now this information to the 'Methods section (new text in bold):

*"Cells were synchronised with 2mM thymidine **for 42h.**"*

Line 384. It is not clear how do authors wash away intracellular thymidine, and if not, then how to they ensure that intracellular thymidine does not affect further diversification of cells (e.g. cells that start to divide are those that were able to pump out the thymidine).

We have indicated in the Methods section that *"cells were washed twice with PBS to remove thymidine"*. Now with the additional experiments that we have performed and addressed in the answer to the first comment of Reviewer #1, we can also assess how well cells are released from the block. From Supplementary Fig. 1c-d, we can see that cells do continue cell cycle progression after removing thymidine 4h and 8h.

Line 385. Please indicate how many cells per well were obtained after 48h of incubation.

We have now added this information to the 'Methods' section (new text in bold):

*"After 48 hours, 4 wells were used for cell counting (**27,000-31,000 cells/well**)"*

Line 385. Please indicate how many cycles cells underwent during 10, 6 and 7 days of incubation. It is not clearly explained what was the rationale of choosing 10, 6 and 7 days?

We have now added this information to the 'Methods' section (new text in bold):
"After the treatment, cells were recovered for 10 (**5 cell cycles**), 6 (**3 cell cycles**) and 7 days (**3-4 cell cycles**) respectively"

Line 440. What are the ratios of NK to cancer cells?

We have now added this information to the 'Methods' section (new text in bold):
"On day 2, NK cells were added into cancer cells in indicated ratios (**Fig. 3f and Supplementary Fig. 5h-i**) to cancer cell numbers."

Lines 446, 465, 466. What are concentrations?

We have now added this information to the 'Methods' section (new text in bold):
"Indicated concentrations of olaparib (**Fig. 3d and Supplementary Fig. 5g**) were added."
"Indicated concentrations of small molecules (**Supplementary Fig. 7a**) were added for 24h"

Line 452: What do you consider "standard procedures"?

To make the homologous recombination repair functional assay subsection in Methods clearer, we modified the text as follows (new text in bold):

"Fixed cells were **permeabilized in 0,2% Triton X-100 in PBS++ (PBS with 1mM CaCl₂ and 0,5mM MgCl₂) for 20 minutes, followed by 30-minute blocking in staining buffer (0,5% BSA, 0,15% glycine, 0,1% Triton X-100 in PBS++). Next, the cells were incubated with primary antibodies against RAD51 (ab133534, Abcam, diluted 1:1000) and cyclin A2 (GTX634420, GeneTex, diluted 1:500) overnight at 4°C, followed by incubation with secondary antibodies (goat anti-mouse IgG-Alexa Fluor 488, A11029; or goat anti-mouse IgG-Alexa Fluor 568, A11004; and goat anti-rabbit IgG-Alexa Fluor 647, A21245; LifeTechnologies, diluted 1:1000) at RT for 1 hour. Nuclei were counterstained with 2 µg/ml of Hoechst."**

Reviewer #2 (Remarks to the Author): expertise in ovarian cancer biology and treatment response

This manuscript is beautifully written with detailed methodology included.

The manuscript details the need for new high-resolution prediction of cells at a transcriptomic level which, prior to exposure to drug/other therapy (carboplatin, olaparib or NK cell killing), are already primed to be display phenotypic resistance. Further study of these features included different genetic contexts, such as seen clinically (e.g., HRD states stemming from BRCA1 or 2 mutations). Further extension of these studies identified

potential sensitizing agents to be used prior to therapeutic drug/other exposure as proof of principle, to revert identified transcriptional profiles to more sensitive ones. Lastly, there was a link found between sensitivity to innate immunity and HRD status, which adds to the well accepted understanding that HRD confers enhanced response to adaptive immunity.

The in vitro models use both ovarian endometrioid but mainly a high-grade serous cell line (BRCA2+). The need for cell cycle synchronization is stressed. Labeled sister cells are characterized to be representative clones of the untreated state.

Cells are exposed in vitro one time with drugs for 5 days, or NK cells for 24h. Doses were given which effect 70-80% killing of cells. The scRNA-seq transcriptomic profile of viable cells at that point determined definition of drug resistance. We know that repeated exposure can result in further killing of cells, however in this model, their immediate goal was achieved with this experimental design. However, drug resistance likely not only exists apriori, but develops upon repeated exposure, or exposure to different drugs along the continuum.

PARP inhibitors have shown most efficacy in HRD states (mainly as maintenance, not really treatment—see line 82), however has shown some efficacy as well in non-HRD patient populations. Suggest modifying this sentence.

We thank the reviewer for these supportive comments and insightful overview of the manuscript in the research context of resistance in the gynecological cancer setting. The point raised about PARP inhibitors is important and to address this, we have modified the sentence as follows (new text in bold):

*“Carboplatin is part of standard-of-care for HGSOC patients, while PARP inhibitors are used **as maintenance therapy especially for patients with homologous recombination deficiency (HRD).**”*

Regarding the link between innate immunity and HRD stated to be new, **I refer the authors to the following article (and others) for possible discussion:** Pharmacologic induction of innate immune signaling directly drives homologous recombination deficiency. Lena J. McLaughlin, Lora Stojanovic, Aksinija A. Kogan, +8, and Feyruz V. Rassool. 117 (30) 17785-17795; <https://doi.org/10.1073/pnas.2003499117>, PNAS 2020. This area is important, and I believe underlies what is seen clinically, in that different patients have differing levels of innate immunity, leading to more or less control of their tumor growth/quiescence. Unfortunately, in epithelial ovarian cancer, the levels of innate immunity are thought to be low in most cases.

We acknowledge that the connection between innate immunity and HRD is not new, and to address this, we have modified Discussion text as follows (new text in bold):

*“However, the effect of HRD on NK cytotoxicity has not been reported, **although activation of intracellular innate immune signalling has been shown to induce HRD (McLaughlin et al. 2020).**”*

Lastly, what do the authors see is the clinical translation of the in vitro sensitizing data presented? Inclusion of sequential sensitizing agents? If so, the drugs themselves optimally should be chosen to not add significant toxicity to the therapeutic combinations which follow. In addition, the duration of sensitizing therapy needs to be short as most ovarian cancer patients have advanced disease and need to get started on treatment as soon as possible after diagnosis.

This is an important aspect, and we are thankful for this clinical viewpoint to the targeting of primed resistance. To acknowledge this, we have added the following sentence to the Discussion section:

“In the clinical trial setting, we envision that the predicted drugs could be used as short-term sensitizers prior to main cytotoxic therapies that need to be started without delay for patients with advanced disease, such as those diagnosed with HGSOC.”

Overall, this manuscript has high impact in the identification of pre-sensitive and resistant clones of cells which exist apriori, prior to primary treatment. The data raise the possibility that treatment of these cells with pre-sensitizing agents may help with response to primary therapy. The hope, would be that perhaps on follow-up, this approach may reduce the development of resistant cells, and demonstrate further downstream advantages.

Setsuko K. Chambers

Reviewer #3 (Remarks to the Author): expertise in computational transcriptomics and evolution

I have been involved to provide expert advice on the computational evolution aspects of this work, which presents the idea of studying differential expression across sister cells. The notion of sister cell is linked to the type of experimental design (reSisTrace) proposed by the authors, which use barcodes to tag cells, cell-cycle synchronisation to phase division clocks, and harvest cells after 48 hours (~1 division) upon splitting pre-exposure and post-exposure cells. Then the authors use this design to scan for particular inhibitors and determine the resistance mechanisms linked to aneuploidy and HRD phenotypes in cell lines.

The paper is largely experimental and contains little computational parts to be evaluated, some of which deserving clarifications. Overall, I anticipate that the general merits of this work should be better assessed based on the relevance of the biological conclusions and the presented experimental design. The computational aspects of this work are:

- Mathematical modelling for reSisTrace: the authors claim to use a model to design reSisTrace, but they do not give any detail about the computations carried out in this

“model”. I understand that they have some parameters mentioned in the Methods section (and Figure S1), but literally no description is given (a formula!) about how these parameters are used. The term simulation is also used, but without any immediate reference to what a simulator is in this case. I suspect that this mathematical model is a simple combinatorial equation that the authors use to determine some parameters of the assay. However, this is all left to my imagination, which should not be the case. This part should be clarified and properly framed, otherwise it is not possible to understand how much it contributes to the overall work — is this a real mathematical model?

We apologise for the confusion. The actual implementation of the computer simulation was based on the workflow of lineage tracing, involving multiple parameters to mimic the actual experimental factors (previous Figure S1e, now moved to a new Supplementary Note). There is no formal mathematical model as analytical formulas to quantify the outcome of the experiment were not available. We have changed the title of the method section from ‘Mathematical modelling and simulation of lineage tracing’ into ‘Computational modelling and simulation of lineage tracing’. Furthermore, we have added a new Supplementary Note to explain how the computer simulation was done in greater detail. We have also modified the main text to clarify this:

“To assess the accuracy of pre-sensitive cell labels while taking into account experimental uncertainties, we performed computational modelling and simulations where we incorporated experimentally measured parameters for uneven cell doubling before the sample was split, random distribution of sister cells during the split, as well as various forms of cell loss during the experimental procedure and analysis (See Supplementary Note for further details).”

- CNV inference: the authors use InferCNV to infer copy numbers from their data. This part is taken-as-it-is because I understand that it is not the main point of the paper to discuss the quality of that inference. I think that overall the results from InferCNV - which I do know well - should be reliable also because the subclones are very large in the input datasets. The authors might just perform some post-inference analysis (e.g, bootstrap) to determine the confidence in the InferCNV estimates.

We thank the reviewer for raising awareness that readers might not be familiar with inferCNV or doubt its output. For this purpose we have added a brief explanation in the Methods section about how inferCNV works, in addition to the denoising filters and post-inference analysis, using Hidden Markov Models, in order to increase confidence on the CNV estimates. As the Reviewer pointed out, it is also worth mentioning that, due to their size, the detected subclones are reliable, which is our main rationale for using inferCNV in this study and not the interpretation of the particular aberrations on the number of copies. Moreover, **Supplementary Fig. 4c** (previous **Fig. S4b**) shows that the identified subclones are stable for pre- and post-treatment clones, which further increases the confidence on the correct subclone identification.

The text below is now added to the Methods section (new text in bold):

”Briefly, inferCNV infers changes in the number of copies by averaging relative expression levels over a sliding window across large genomic regions. To enhance the accuracy of CNV estimates, denoising filters are applied based on the standard deviation of residual normal expression values. The Leiden algorithm was used for determining the underlying subclonal populations based on the CNV profiles. At the subclone level, a Hidden Markov Model followed by a Bayesian network is utilized to compute the posterior probability of a CNV region belonging to a specific amplification or deletion state. CNV regions with higher mean posterior probabilities for the normal state are removed as likely false positive predictions.”

- Connectivity scores: this metric is taken from another paper (ref [30]) and is therefore not discussed here.

Yes, connectivity score is an association metric that is taken from the cited work (reference ²³ in the revised manuscript), and we have further clarified the text regarding its use.

Overall, a general decision on this manuscript should focus on how sister-resolution lineage tracing seems a novel approach to reveal complex cell states associated with treatment resistance in cancer.

Indeed, our method provides a novel method to interrogate and rationally target pre-existing resistant states of cancer cells in sister-cell resolution. Compared to the previous lineage tracing methodologies, such as the Watermelon system, our method is unique in its ability to successfully identify the primed drug resistant populations to establish their transcriptomic signatures. We further present a computational strategy to predict drug combinations to revert these transcriptomic states which we then experimentally validate. We have highlighted the novelty of the sister-resolution lineage tracing throughout the manuscript, and hopefully with all the modifications we have made this point clearer.

REVIEWER COMMENTS

Reviewer #1 (Remarks to the Author):

Please refer to the enclosed file.

Reviewer #2 (Remarks to the Author):

I have reviewed the responses to my review.

I find the responses are appropriate.

One issue which was not referred to, (most likely because I did not pose this as a question), is my comment about the reality of development of drug resistance, which is based on repeated drug exposures over time. Your model was a one-time drug exposure, however to translate more to patients, a better model is one with repeated drug exposures over time. Have you considered developing such a model?

Reviewer #3 (Remarks to the Author):

the authors have addressed my main concern, which required to state a formal definition of the mathematical model and simulations carried out. This has been done with a new Supplementary Note.

Just to be clear, I do not understand your reply sentence "There is no formal mathematical model as analytical formulas to quantify the outcome of the experiment were not available." What I cared about are the formulas that are now in the figure in Supplementary Note. Those formulas are the formal definition of the mathematical model, so I suggest that:

- the authors put those formula in the text of the Note (what do you multiply, randomly sample etc etc) and not just in the figure
- make the code that has implemented those simulations available as Supplementary Data.

I think the latter is required for whoever needs to improve and customise their own ResisTRACE experiments.

I have no other comments.

Round 2

The reviewer appreciates the responses by authors for clarifying some parts of the manuscript, yet the remaining concerns regarding data analysis require careful attention.

1. Regarding proper data analysis and reproducibility

CRITICAL: Throughout the manuscript, the authors employ several unconventional and arguably questionable analytical parameters. The explanations and terminology used in the text can be confusing at times. If this manuscript is to be accepted, it should be published with all the notebooks and codes used for data analysis and figure generation. This would enable others to reproduce the results and avoid potential misunderstandings.

To illustrate my point, consider Lines 531-533 in the revised manuscript: "The 'FindMarkers' function (`test.use = "wilcox"`, `logfc.threshold = 0`, `min.pct = 0`) was used to determine the differentially expressed genes by Wilcoxon rank sum test between different groups.". The authors will recognize that the choice of parameters, `logfc.threshold = 0`, `min.pct = 0`, is dubious. Under these parameters, outliers (i.e., genes expressed by a very small fraction of cells with a very low count) will show high DGE. However, such an output is not suitable for identifying differentially expressed genes in a given dataset. It's important to note that the field's consensus is to only include genes in DGE calculations that are expressed by a certain fraction of the cells, clusters, etc. The authors should be aware of this as they refine their analysis.

2. Regarding selection pressure and what is actually being selected

Previous reviewer's comment: *The results of control conditions in majority of experiments follow treatment conditions results almost identically. Why? It seems these results indicate the weak effect overall.*

Response: We thank the reviewer for this notion and agree that the control condition consistently shows similar changes to those seen in the treatments. We acknowledge this in our manuscript repeatedly, stating that the control experiment presents consistent selection pressure. This highlights the need to remove this experimental fitness effect from further analyses as we have done for all analyses of the pre-resistance signatures (presented in Figures 2 to 4).

Follow-up comment 1: Although it's possible to repeatedly acknowledge in the text that control conditions closely mirror the treatment, this doesn't resolve any underlying issues if the experiments were not properly conducted or if they introduced unwanted effects or artifacts.

It seems that cells under both control and drug conditions are experiencing a strong selection pressure that dominates the transcriptional signal. In essence, factors from the experimental setup, appear to exert a stronger impact on the cells than the drugs being studied. In their response, the authors explained that they used 2 mM thymidine treatment for 42 hours to enrich the cells in the S phase. After thymidine removal, they subjected the cells to purifying selection. In other words, only some cells could adapt to this new condition and divide, and the presence or absence of

the drug had a minor, if not insignificant, effect on them. Hence, the transcriptional signature captured by the authors is dominated by cell adaptation to thymidine removal, instead of being indicative of pre-sensitivity or pre-resistance, sensitivity or resistance to the drug. The authors might disagree with this interpretation, but the results presented in the manuscript do not appear to disprove these arguments.

Follow-up comment 2: Similarly, the data clearly shows that in the majority of experiments, the control conditions largely mimic the treatment conditions. The authors propose omitting this observation from further analysis, but doing so would obscure the fact that the control conditions almost perfectly reflect the treatment conditions. Given that the control mirrors the treatment pattern even for different kinds of treatments, it could be inferred that a) the different treatments had virtually no significant effect; b) there may be human error involved (e.g., failure to add treatment or inadvertent addition of treatment to control); c) there could be something fundamentally incorrect with the experiments or the reagents used; d) the authors are detecting cell adaptation to thymidine removal. In light of these observations, the relevance of the "Experimental fitness effect" is questionable.

3. Regarding gene ontology analysis:

Previous reviewer's comment: *When performing gene ontology analysis and association with HRD, control conditions are not included. This is needed to prove that the conclusions and statements of the manuscript are indeed specific to a certain condition, and do not apply to other conditions i.e. control.*

We thank the reviewer for this important notion. We have in fact removed the effect of the control condition from pre-resistance signatures against carboplatin, olaparib, or NK cells, and described this in the 'Methods' section but unfortunately failed to communicate this clearly in the main text. We have now added a sentence indicating this (new text in bold):

"With the identity of pre-resistant and pre-sensitive cells, as well as the identity of subclones, we determined rank-based transcriptomic signatures for pre-resistance and subclones (Supplementary Data 2). **To remove the effect of experimental fitness from further analyses, pre-resistance signatures were adjusted to the control condition by subtracting the control pre-survival signal from the pre-resistance signals.**"

Follow-up comment 3: It seems that the authors misunderstood the initial comment. The comment aimed at GO terms being enriched in different conditions, i.e. checking if a certain term is enriched not only in treatment condition but in control (in Figure 2e). This does not require any removal of effect (of what?). Furthermore, it is not clear what exactly was subtracted and from what, (normalized, raw counts?).

Follow-up comment 4: The authors admit they performed data analysis "**...by subtracting the control pre-survival signal from the pre-resistance signals.**"

There is a reasonable concern that authors processed the data incorrectly. Authors must be aware that data manipulations such as "subtraction" shall not be performed in the context of transcriptomic data. This is different for methods such as fluorescence images where subtracting the background signal is a standard procedure, or delta-delta Ct for qPCR. Yet, in transcriptomics field simply subtracting

the expression value of control is inappropriate.

Follow-up comment 5: It is unclear, what is meant by stating i.e. Carboplatin preR-preS log2FC adjusted by non-treatment control (taken from Supplementary data 2)? fold-change of pre-R vs pre-S with subtracted fold-change of control?

4. Regarding batch effects:

Answer with regards to batch effect: We treated and analysed all the drug- or DMSO perturbed samples in duplicates, so that all treatments, including DMSO, and sample collection, were performed simultaneously in one batch, and we further pooled the samples by cell multiplexing method before single-cell RNA-seq sample processing to minimise any batch effects.

Comment: It is good to know that the authors took measures to minimize the batch effect in sample processing, however, there seems to be a misunderstanding of the general batch effect concept. One can harvest as many conditions in the same go, but if the cells were growing in different conditions, for simplicity let's say in different wells, the cells are already **literally** in different batches. Sure, harvesting separately and then processing in a pool will minimize further effects and the multiplexing effort is welcome, however the samples inherently come from different batches. Thus, it would be beneficial to show that the treatment effect persists even after batch effect correction for different treatments. Also, given the response, it is now unclear how the different experiment DMSO controls differ within one another? It would be beneficial, for the proof of the claimed non-existence of batch effect, to show that DMSO-treated cells from all these different experiments overlap fully and perfectly.

5. Regarding threshold and additional figure:

New Comment 1: In Supplementary data 10, for majority of conditions the lower bound for UMI is 20 000, which is extremely high value for the field (standard practice is 1000 UMI). What is the rationale to use such value? Please provide justification for using such unusually large threshold and why cells expressing e.g. 19000 UMIs are discarded? The authors should provide the sequencing depth metrics or UMI count histograms because these numbers raise unnecessary suspicions.

New Comment 2: Additional data representation is required: Please provide UMI distribution histogram from 0 to 200k (or higher). Use Fig 6.1 as a guiding example:

https://biocellgen-public.svi.edu.au/mig_2019_scrnaseq-workshop/quality-control-and-data-visualisation.html

Response to round 2 reviews for the manuscript “Tracing back primed resistance in cancer via sister cells”

We thank the reviewers for their additional comments and suggestions. We have revised the manuscript accordingly to address all issues that were raised.

The major changes in the re-revised manuscript are:

- We have now placed notebooks and codes used for data analysis and figure generation to GitHub to address a request from Reviewer #1.
- We have clarified how the treatment-specific signatures were adjusted with the control condition by adding a more detailed description and the applied formula in the Methods section, to address concerns from Reviewer #1. We have also modified the text to acknowledge the effect of thymidine block in experimental fitness.
- We have added a new Supplementary Fig. 6 showing comparisons of the replicates of drug-induced profiles, allowing to assess potential batch effects to address concerns from Reviewer #1.
- We have added a new Supplementary Fig. 9 to show histograms of UMI counts in each sample to justify the chosen UMI thresholds, to address a request from Reviewer #1.
- We have added formula to the text of the Supplementary Note, and shared the code for implementing the simulations as Supplementary Data 13, in addition to sharing the code in GitHub to address a suggestion of Reviewer #3.
- In addition, we have updated Fig. 4e and Supplementary Fig. 7 (previous Supplementary Fig. 6), and Supplementary Data 10. For Fig. 4e and Supplementary Fig. 7, we noticed a bug in the code used for analysis that resulted in omitting one of the two replicates when calculating average log₂FCs for the drug-induced single cell expression profiles. Importantly, this did not change our major conclusions. For Supplementary Data 10, we further found that we had marked incorrect UMI thresholds for two samples in the table; this does not affect our analysis results. We have now double-checked all of our code and found no further mistakes. The details of these updates, including comparison of previous and updated figures, can be found at the end of this response letter.

Round 2, Rev#1:

The reviewer appreciates the responses by authors for clarifying some parts of the manuscript, yet the remaining concerns regarding data analysis require careful attention.

1. Regarding proper data analysis and reproducibility

CRITICAL: Throughout the manuscript, the authors employ several unconventional and arguably questionable analytical parameters. The explanations and terminology used in the text can be confusing at times. If this manuscript is to be accepted, it should be published with all the notebooks and codes used for data analysis and figure generation. This would enable others to reproduce the results and avoid potential misunderstandings.

To illustrate my point, consider Lines 531-533 in the revised manuscript: "The 'FindMarkers' function (`test.use = "wilcox", logfc.threshold = 0, min.pct = 0`) was used to determine the differentially expressed genes by Wilcoxon rank sum test between different groups.". The authors will recognize that the choice of parameters, `logfc.threshold = 0, min.pct = 0`, is dubious. Under these parameters, outliers (i.e., genes expressed by a very small fraction of cells with a very low count) will show high DGE. However, such an output is not suitable for identifying differentially expressed genes in a given dataset. It's important to note that the field's consensus is to only include genes in DGE calculations that are expressed by a certain fraction of the cells, clusters, etc. The authors should be aware of this as they refine their analysis.

Round 2 response: We have now placed notebooks and codes used for data analysis and figure generation to GitHub (<https://github.com/TangSoftwareLab/ReSisTrace/tree/master>) to enable others to reproduce the results and avoid potential misunderstandings.

Regarding the possible outliers from lowly expressed genes within the top DGEs, the top100 DGEs ordered by log2FC are all expressed above 20% of the cells. We did not filter by percent expressed (`min.pct`) or minimal log fold change (`logfc.threshold`) in our initial DGE analyses, in order to maximise the overlap between the gene sets from distinct conditions and replicates for the follow-up analyses. If we had used for example the default thresholds of the *Seurat* package, we would have gained log2FC values for distinct subsets of genes for each condition and replicate. We acknowledge that this results in inclusion of lowly expressed genes that results in noise, though not in the top DGEs ordered by log2FC as discussed above. Moreover, we did further filter the genes for the analyses shown in Figures 2 to 4 by including only sister conserved genes, which reduced the presentation of lowly expressed genes (see **Fig. R2.1** below). For example, the lowly expressed genes (`min.pct < 0.1` corresponding to the default filtering value in *Seurat*) represent around 9% of the differentially expressed genes if we include the non sister-conserved genes. However, if we restrict this list to only sister conserved genes, as we did in all of our downstream analyses, this percentage drops to ~1%, suggesting that the impact of the lowly expressed outlier genes is very limited.

Figure R2.1: Histograms of sister concordant and discordant gene expression in distinct samples and replicates. The x axis shows percentage of cells that express a particular gene, while y axis shows the number genes in particular expression range in blue for sister-discordant genes and in red for sister-concordant genes. Red vertical dashed line indicates the default threshold (0.1) from the *Seurat* package.

2. Regarding selection pressure and what is actually being selected

Previous reviewer's comment: *The results of control conditions in majority of experiments follow treatment conditions results almost identically. Why? It seems these results indicate the weak effect overall.*

Response: We thank the reviewer for this notion and agree that the control condition consistently shows similar changes to those seen in the treatments. We acknowledge this in our manuscript repeatedly, stating that the control experiment presents consistent selection pressure. This highlights the need to remove this experimental fitness effect from further analyses as we have done for all analyses of the pre-resistance signatures (presented in Figures 2 to 4).

Follow-up comment 1: Although it's possible to repeatedly acknowledge in the text that control conditions closely mirror the treatment, this doesn't resolve any underlying issues if the experiments were not properly conducted or if they introduced unwanted effects or artifacts.

It seems that cells under both control and drug conditions are experiencing a strong selection pressure that dominates the transcriptional signal. In essence, factors from the experimental setup, appear to exert a stronger impact on the cells than the drugs being studied. In their response, the authors explained that they used 2 mM thymidine treatment for 42 hours to enrich the cells in the S phase. After thymidine removal, they subjected the cells to purifying selection. In other words, only some cells could adapt to this new condition and divide, and the presence or absence of the drug had a minor, if not insignificant, effect on them. Hence, the transcriptional signature captured by the authors is dominated by cell adaptation to thymidine removal, instead of being indicative of pre-sensitivity or pre-resistance, sensitivity or resistance to the drug. The authors might disagree with this interpretation, but the results presented in the manuscript do not appear to disprove these arguments.

Follow-up comment 2: Similarly, the data clearly shows that in the majority of experiments, the control conditions largely mimic the treatment conditions. The authors propose omitting this observation from further analysis, but doing so would obscure the fact that the control conditions almost perfectly reflect the treatment conditions. Given that the control mirrors the treatment pattern even for different kinds of treatments, it could be inferred that a) the different treatments had virtually no significant effect; b) there may be human error involved (e.g., failure to add treatment or inadvertent addition of treatment to control); c) there could be something fundamentally incorrect with the experiments or the reagents used; d) the authors are detecting cell adaptation to thymidine removal. In light of these observations, the relevance of the "Experimental fitness effect" is questionable.

Round 2 response: We agree with the reviewer that thymidine block and cellular adaptation to thymidine removal form very likely a part of the selection pressure shared by all the treatments and the control condition, and thus part of the experimental fitness effect. To acknowledge this, we have modified the manuscript as follows (new text in bold):

*"Notably, we observed consistent transcriptomic changes also in the non-treatment control condition between the pre-surviving and pre-extinct cells, suggesting that the experimental procedures themselves, **such as those introduced by the preceding release from the thymidine block**, pose a non-random selection pressure which should be corrected."*

Importantly, we do not omit the fitness effect but instead address it according to standard bioinformatic procedures as described below in the following response. Finally, the functional relevance of the identified primed resistance signatures was validated by the drug pre-treatment experiments (Fig. 4). There, the drugs that induced expression changes that are opposite to those found in the control adjusted pre-resistance signatures, were also successful in pre-sensitising Kuramochi cells to perturbation with carboplatin, olaparib or NK cells, suggesting that the approach to adjust for the experimental fitness indeed was valid.

3. Regarding gene ontology analysis:

Previous reviewer's comment: *When performing gene ontology analysis and association with HRD, control conditions are not included. This is needed to prove that the conclusions and statements of the manuscript are indeed specific to a certain condition, and do not apply to other conditions i.e. control.*

We thank the reviewer for this important notion. We have in fact removed the effect of the control condition from pre-resistance signatures against carboplatin, olaparib, or NK cells, and described this in the 'Methods' section but unfortunately failed to communicate this clearly in the main text. We have now added a sentence indicating this (new text in bold):

"With the identity of pre-resistant and pre-sensitive cells, as well as the identity of subclones, we determined rank-based transcriptomic signatures for pre-resistance and subclones (Supplementary Data 2). **To remove the effect of experimental fitness from further analyses, pre-resistance signatures were adjusted to the control condition by subtracting the control pre-survival signal from the pre-resistance signals.**"

Follow-up comment 3: It seems that the authors misunderstood the initial comment. The comment aimed at GO terms being enriched in different conditions, i.e. checking if a certain term is enriched not only in treatment condition but in control (in Figure 2e). This does not require any removal of effect (of what?). Furthermore, it is not clear what exactly was subtracted and from what, (normalized, raw counts?).

Follow-up comment 4: The authors admit they performed data analysis "**...by subtracting the control pre-survival signal from the pre-resistance signals.**"

There is a reasonable concern that authors processed the data incorrectly. Authors must be aware that data manipulations such as "subtraction" shall not be performed in the context of transcriptomic data. This is different for methods such as fluorescence images where subtracting the background signal is a standard procedure, or delta-delta Ct for qPCR. Yet, in transcriptomics field simply subtracting the expression value of control is inappropriate.

Follow-up comment 5: It is unclear, what is meant by stating i.e. Carboplatin preR-preS log2FC adjusted by non-treatment control (taken from Supplementary data 2)? fold-change of pre-R vs pre-S with subtracted fold-change of control?

Round 2 response: Follow-up comments 3 to 5 are all addressing the same issue and hence we answer them jointly here. Our subtraction approach is analogous to the *makeContrast* method included in *limma* package (Richie et al. 2015: doi: 10.1093/nar/gkv007; Law et al. 2020, doi: 10.12688/f1000research.27893.1), which is a

gold-standard package for RNA-seq analysis.

We are calculating our signatures by ranking the gene expression changes between pre-resistant and pre-sensitive cells as log2 fold changes (log2FC), wherein we subtract the log2FCs in the control condition from the log2FCs within the treatment condition. For each gene, the value being ranked is essentially as follows:

$$\log_2\left(\frac{R}{S}\right) - \log_2\left(\frac{R'}{S'}\right) = \log_2\left(\frac{R/S}{R'/S'}\right) = \log_2\left(\frac{R/R'}{S/S'}\right)$$

where R is the gene expression value in the treatment pre-R group; S is the gene expression value in the treatment pre-S group; R' is the gene expression value in the control pre-R group (i.e. in pre-survival cells); S' is the gene expression value in the control pre-S group (in pre-extinct cells). By normalising the fold changes as such, those genes that show similar gene expression in pre-resistant cells of a treatment condition and pre-survival cells in the control condition when compared to pre-sensitive or pre-extinct cells, respectively, will have a very small absolute value being ranked.

To make our approach more clear, we have modified the Methods section as follows (new text in bold, with the formula added):

“The log2FC between pre-resistant and pre-sensitive cells for each gene was averaged across two replicates and then subtracted by the log2FC of pre-surviving versus pre-extinct cells from the non-treatment control samples:

$$\frac{\log_2\left(\frac{R_1}{S_1}\right) + \log_2\left(\frac{R_2}{S_2}\right)}{2} - \frac{\log_2\left(\frac{R'_1}{S'_1}\right) + \log_2\left(\frac{R'_2}{S'_2}\right)}{2}$$

where R_1 and R_2 are the gene expression values in treatment pre-R group in repeats 1 and 2, respectively; S_1 and S_2 are the gene expression values in treatment pre-S group in repeats 1 and 2, respectively; R'_1 and R'_2 are the gene expression values in control pre-R group in repeats 1 and 2, respectively; S'_1 and S'_2 are the gene expression values in control pre-S group in repeats 1 and 2, respectively. Only the sister-concordant genes were included in the rank-based signature.”

Supplementary Data 2 columns ‘Carboplatin preR-preS log2FC adjusted by non-treatment control’, ‘Olaparib preR-preS log2FC adjusted by non-treatment control’ and ‘NK preR-preS log2FC adjusted by non-treatment control’ indeed show the resulting, control adjusted log2FCs, according to what was suggested by Reviewer #1 in ‘Follow up comment 5’ above.

Importantly, as highlighted also in the previous answer, the functional relevance of the resulting, control adjusted signatures in primed resistance is validated by the drug pre-treatment experiments (Fig 4).

4. Regarding batch effects:

Answer with regards to batch effect: We treated and analysed all the drug- or DMSO perturbed samples in duplicates, so that all treatments, including DMSO, and sample collection, were performed simultaneously in one batch, and we further pooled the

samples by cell multiplexing method before single-cell RNA-seq sample processing to minimise any batch effects.

Comment: It is good to know that the authors took measures to minimize the batch effect in sample processing, however, there seems to be a misunderstanding of the general batch effect concept. One can harvest as many conditions in the same go, but if the cells were growing in different conditions, for simplicity let's say in different wells, the cells are already **literally** in different batches. Sure, harvesting separately and then processing in a pool will minimize further effects and the multiplexing effort is welcome, however the samples inherently come from different batches. Thus, it would be beneficial to show that the treatment effect persists even after batch effect correction for different treatments. Also, given the response, it is now unclear how the different experiment DMSO controls differ within one another? It would be beneficial, for the proof of the claimed non-existence of batch effect, to show that DMSO-treated cells from all these different experiments overlap fully and perfectly.

Round 2 response: First of all, we have performed the drug treatments and the DMSO control treatment in two replicates in a single experimental set-up, i.e. we do not have separate DMSO treatments for each drug treatment separately. We do agree that each well poses a batch effect. To assess the replicates separately, we projected them on a UMAP for qualitative evaluation (new Suppl Fig 6a) and plotted a scatter plot of log₂FCs comparing each of the drug-treated replicate to DMSO replicates to allow a quantitative evaluation of the batch effect (new Suppl Fig 6b). Both comparisons show that the batch effect of separate wells is limited in this experiment.

To address the new data shown in the new Supplementary Figure 6, we modified the text as follows (new text in bold):

*“For the first step, we generated drug-induced transcriptome profiles in Kuramochi cells through scRNA-seq, with two replicates of each treatment and all samples processed simultaneously to minimise batch effects. **Apart from sitagliptin that did not induce consistent changes in gene expression, the drug-induced replicates showed aligned gene expression changes, with average correlation coefficient of 0.78 (Supplementary Fig. 6a, b).** The UMAP projection shows that three proteasome inhibitors (bortezomib, delanzomib, and ixazomib) induced transcriptome profiles that were clearly clustered together and distinguished from the DMSO control (Fig. 4c-d, Supplementary Fig. 7).”*

Supplementary Figure 6. Drug-induced expression profiles shown as UMAP projection (a), and gene expression changes to the DMSO control for corresponding replicates 1 and 2 of each drug treatment (b).

5. Regarding threshold and additional figure:

New Comment 1: In Supplementary data 10, for majority of conditions the lower bound for UMI is 20 000, which is extremely high value for the field (standard practice is 1000 UMI). What is the rationale to use such value? Please provide justification for using such unusually large threshold and why cells expressing e.g. 19000 UMIs are discarded? The authors should provide the sequencing depth metrics or UMI count histograms because these numbers raise unnecessary suspicions.

New Comment 2: Additional data representation is required: Please provide UMI distribution histogram from 0 to 200k (or higher). Use Fig 6.1 as a guiding example:

https://biocellgen-public.svi.edu.au/mig_2019_scrnaseq-workshop/quality-control-and-data-visualisation.html

Round 2 response: Regarding the rationale, we have selected the threshold based on the UMI distribution in our data (histogram shown below and as Supplementary Fig. 9). This is analogous to the method shown in the link cited by the reviewer above, where the tutorial states that: "Wells with few reads/molecules are likely to have been broken or failed to capture a cell, and should thus be removed." In the shown example on pluripotent stem cells, the tutorial proceeds to filter cells with less than 25 000 UMIs, a threshold that is even stricter than the one we have used.

Notably, RNA content and thus UMI count is proportional to the cell size (Padovan-Merhar et al. Cell 2015; doi: 10.1016/j.molcel.2015.03.005). The cells we are studying, high-grade serous ovarian cancer cells, are very large and thus have a large quantity of RNA. This means that they are expected to also have higher UMI counts than for example the PBMC cells standardly used in for example the 'Seurat' package tutorials, provided that the sequencing is sufficiently deep. In this case the cells had around 140 000 reads in average, suggesting that the sequencing depth indeed is sufficient to detect higher UMI counts for ovarian cancer cells when compared to PBMCs.

To address the new Supplementary Fig. 9, we changed the text in the 'Methods' section as follows (new text in bold):

*"Based on the distribution of the UMI counts (**Supplementary Fig. 9**), number of genes, and percentage of mitochondrial transcripts, we filtered out the cells using the thresholds indicated in Supplementary Data 10."*

Supplementary Figure 9. Histogram of UMI counts, showing the selected thresholds, listed in Supplementary Data 10, as red vertical lines.

Reviewer #2 (Remarks to the Author):

I have reviewed the responses to my review.

I find the responses are appropriate.

One issue which was not referred to, (most likely because I did not pose this as a question), is my comment about the reality of development of drug resistance, which is based on repeated drug exposures over time. Your model was a one-time drug exposure, however to translate more to patients, a better model is one with repeated drug exposures over time. Have you considered developing such a model?

Round 2 response: We agree that emergence of resistance in the clinic is a step-wise, gradual process, as for example HGSOc patients typically receive 6 or more cycles of platinum-taxane chemotherapy. In the current, simplified form, our approach is only observing the first of those steps but could be modified further to encompass multiple rounds of treatment, by repeatedly re-treating a part of the post-treatment sample. This would be possible without seriously compromising the detection of resistant lineages as each viable lineage is typically represented by a multitude of cells after recovery.

Reviewer #3 (Remarks to the Author):

the authors have addressed my main concern, which required to state a formal definition of the mathematical model and simulations carried out. This has been done with a new Supplementary Note.

Just to be clear, I do not understand your reply sentence "There is no formal mathematical model as analytical formulas to quantify the outcome of the experiment were not available." What I cared about are the formulas that are now in the figure in Supplementary Note. Those formulas are the formal definition of the mathematical model, so I suggest that:

- the authors put those formula in the text of the Note (what do you multiply, randomly sample etc etc) and not just in the figure
- make the code that has implemented those simulations available as Supplementary Data.

I think the latter is required for whoever needs to improve and customise their own ResisTRACE experiments.

I have no other comments.

Round 2 response: We have now added the formula to the text of the Supplementary Note, and made the code for implementing the simulations available as Supplementary Data 13, in addition to sharing the code in GitHub.

Additional updates:

1. Drug-induced expression profiles

When plotting the log2FCs separately for drug pre-treatments as requested by Reviewer #1, we noticed an error in our previous code that resulted in omitting the log2FC of replicate 1 from the average log2FCs shown in Fig 4e and Supplementary Fig 6. We thus updated Figure 4e, and Supplementary Figure 7 (Supplementary Fig. 6 in the previous submitted manuscript).

Overall, the results showed minimal differences. The most significant change was the loss of signal in sitagliptin treatment which was previously found to have the weakest association across all the connectivity scores for the predicted pre-treatment drugs. To enable assessing the effect of the updated results to the previous results, the updated and previous version of these figures are shown below.

Old Fig. 4e

New Fig. 4e

Old Supplementary Fig. 6

New Supplementary Fig. 7

We modified the text related to sitagliptin as follows:

*“We found that ~~all~~ the predicted drugs, **apart from sitagliptin**, indeed induced changes opposite to the corresponding pre-resistance signatures, with significant negative connectivity scores (Fig. 4e, Supplementary Fig. 7). For example, the top predicted pre-sensitisers for NK cells include pevonedistat, GW843682X and nocodazole, all inducing transcriptomic changes with highly negative connectivity to NK pre-resistance, while pyrrolidine-dithiocarbamate and sitagliptin perturbed profiles had the weakest **or non-significant** association with olaparib pre-sensitivity profiles.”*

...

“In contrast, the NF- κ B inhibitor pyrrolidine-dithiocarbamate was less synergistic with olaparib, whereas sitagliptin used to treat type II diabetes had no effect, in line with its failure to induce expression changes ~~that resemble those of olaparib pre-sensitive cells.~~”

Importantly, this did not change our major conclusions, as the sitagliptin treatment was not synergistic as a pre-treatment (*Supplementary Fig. 8*).

2. UMI thresholds

When plotting the UMI distribution histograms, we noticed that two samples, namely Olaparib1 pre-treatment and Olaparib1 post-treatment, had incorrect values (20 000) in the ‘Lower bound for UMI count’ column in Supplementary Data 10. The correct value for these two samples was 10 000, which has now been updated in Supplementary Data 10 file.

We have now double-checked all of our code and found no further mistakes.

REVIEWER COMMENTS

Reviewer #1 (Remarks to the Author):

My concerns regarding selection pressure that is imposed on cells, and the transcriptional signature that is dominated by the thymidine removal, were not properly addressed. Therefore, with all due respect, I have to restrain from recommending this work for publication in Nature Communications journal. Further, I now realize that the experimental strategy provided in this work, will find little use in the future and can be hardly applicable on primary cells.

Looking at Figure R2.1 there is a concern that by using threshold 0.1, the authors mis-assign the sister-discordant genes as being encoded by sister-concordant genes.

Reviewer #4 (Remarks to the Author):

This manuscript by Dai et al. addresses single-cell transcriptional priming that correlates with cancer cell therapy resistance and then it takes advantage of these transcriptional signatures to harvest from databases any compounds that can successfully synergize to increase efficacy. The ideas are good, even if not extremely novel at this point, and the execution of experiments is largely correct. I still think this would be a nice contribution to the field, provided the analyses are performed to the standards that this budding community of single-cell lineage-tracing researchers requires.

My biggest concern, which I share with reviewer 1, is the fact that to increase the number of sister cells for early state profiling, the authors decided to use thymidine block. I guess we agree that it should not have been used, especially when trying to isolate intrinsic priming signatures linked to cancer therapy, because this essentially creates a two-stressor situation for the cells. As a result of this experimental design, many confounders arise. Consequently, the authors devise a way to try to “regress” this effect out in a way that I have never seen done before. The biggest point I want to discuss is whether this actually works or not. On one hand, I think that regressing out the effects of thymidine block should not be that hard to do in a more proper way. Simply subtracting the fold change from the control or pre-treatment gene by gene is not adequate because this is affected by a large number of confounding factors (including hard-to-correct batch effects) and huge noise due to the sparsity of single-cell data (compared to bulk sequencing approaches). More importantly, it can lead to eliminating real signals (the same gene priming that helps survive after thymidine could lead to survival upon another drug). Even without regressing out the thymidine effect, just at the simplest level, authors could have focused on the pathways/correlated gene networks that are only specifically significant in each of their treatment conditions. In other words, I believe that creating a contrast effect should not be necessary and should not be used in this context (regardless of whether it is similar to a function in a well-used package like limma). Yes, authors do run cleaner experiments later, validating many of these observations, but I don't think this is an excuse, particularly because it will lead others who read this manuscript to follow similar ill-advised practices. In this regard, why didn't the authors pursue identifying “pre-resistant” clusters instead of essentially doing a pseudobulk analysis (just separating between preR and preS)? It appears very clear to me from the data in the Supp Fig 3a that different treatments enrich preR cells in different regions in the UMAP which may mean that specific clusters are enriched in preR cells (especially compared to the control preR cells). For instance, in Olaparib, there seems to be a lot of

cells on the top right part of the UMAP, compared to preR. Did they fully discard this? Again, this might be a simple and relatively easy way to get around thymidine effects.

A second minor concern: regarding the super large UMI counts I was also initially puzzled. In a response to reviewer 1 the author claims “In this case the cells had around 140 000 reads in average, ...” But then they previously suggest that they filtered cells with less than 20000 UMIs. So are they talking reads or UMIs? I suspect the confusion might be because in this cell type the reads and UMIs must be similar (i.e. RNA content is so high). Furthermore, in the plots they share, the UMI count per cell truly looks like around 50-100k per cell... So I did some simple math: if each cell has 100k UMI average, and they aimed to encapsulate about 10,000 cells per well (as one can guess by the methods section), this means they must have needed at least (theoretical) 1,000,000,000 reads per well. Since they sequenced 8 wells day 1, and 8 wells post treatment, that's *at the very theoretical minimum* 20 billion reads. It's a ginormous amount of sequencing that doesn't make any sense here.

However, looking more carefully at the actual cells used in the analysis, one can tell that there are 24000 cells in the day 1 sample, and just 7802 cells in the post-treatment sample (so, just about 32k cells in the total dataset). That's a bit bizarre considering 10x v3.1 captures about 40% of the cells loaded, which means this should have come out at about 120k cells for the total dataset (4 times more cells).

In other words, what I believe happened is the authors really targeted 40k reads per cell (which is usual in most studies). As they expected 120k cells in the dataset, they sequenced ~5 billion reads... However, there were way fewer cells than expected and the reads ended up being spread across 30k cells or so... In addition, it does look like the authors spread those reads unevenly across samples. Likely they did this on purpose, because they wanted to capture more resolution in the pre-treatment condition (to increase accuracy of state-fate analysis), so they assigned more space in the sequencing lane to these pre-treatment libraries. This all seems very plausible to me, but an actual detailed account of these details on sequencing decisions is missing in the methods section, or at least I didn't find it.

I really cannot find any major issues with the manuscript that have not been raised before by other reviewers. In fact, I think the authors did pretty excellent work following up their scLT observations. However, before I can recommend the publication of these results I really want them to get the scLT analysis part right.

–Alejo E. Rodriguez Fraticelli

Response to round 3 reviews for the manuscript “Tracing back primed resistance in cancer via sister cells”

We thank the reviewers for their additional comments and suggestions. We have revised the manuscript accordingly to address all issues that were raised.

The major changes in the re-revised manuscript are:

- To address the concerns related to regressing out the effects of our experimental set-up including the thymidine block, we have now separately analysed the association, or gene ontology enrichment, of each un-adjusted pre-resistance signature, as well as the control condition pre-survival signature, allowing readers to assess them side by side. Thus, in the revised manuscript we have updated the following figure subpanels:
 - We now display the gene ontology enrichment of the original, un-adjusted pre-resistance signatures in **Fig. 2e**, along with the enrichment patterns of the pre-survival signature from the control condition.
 - We now show the connectivity scores of unadjusted pre-resistance signatures with subclonal (**Fig. 2d**), HRD (**Fig. 3e**), and drug-induced signatures (**Fig. 4b, 4e** and **Supplementary Figs. 7 and 8**) along with the control condition as the reference.
- For the drug predictions that were used as basis for the successful validations shown in Fig. 4, we have already originally used the unadjusted signatures, and additional filtering using several variations of control condition contrasted signatures. We have now made our approach more clear also in the *Results* section of the revised manuscript.
- To clarify our chosen sequencing approach/strategy, we have added a more detailed description to the *Methods* subsection ‘*ReSisTrace experimental workflow*’, as well as a new table as ‘**Supplementary Data 11**’.

In addition to these major changes, we re-numbered some of the Supplementary Data mentioned in the *Methods* section to match the order of their appearance in the text.

Importantly, even though the revised approach changed certain individual results presented in subpanels as stated above, such as specific gene ontologies enriched by each pre-resistance signature, it did not change the major conclusions of the manuscript. These still remain to be that i) our approach can identify subtle pre-existing states that predict cancer cell resistance against upcoming treatments, and ii) it further allows targeting these pre-existing states to sensitise cancer prior specific treatments.

REVIEWER COMMENTS

Reviewer #1 (Remarks to the Author):

My concerns regarding selection pressure that is imposed on cells, and the transcriptional signature that is dominated by the thymidine removal, were not properly addressed. Therefore, with all due respect, I have to restrain from recommending this work for publication in Nature Communications journal. Further, I now realize that the experimental strategy provided in this work, will find little use in the future and can be hardly applicable on primary cells.

Looking at Figure R2.1 there is a concern that by using threshold 0.1, the authors mis-assign the sister-discordant genes as being encoded by sister-concordant genes.

Round 3 response: Related to the handling of the experimental selection pressure in the control condition, we refer to our responses below to Reviewer#4. We now further address also the applicability of our method to primary cells or other delicate material such as patient-derived organoids, in the *Discussion* section as indicated below in the Response to Reviewer#4.

Regarding Figure R2.1, there seems to be a misunderstanding; we did not use the threshold of $\text{min.pct} = 0.1$ (expressed in at least 10% of the cells) in our DGE analyses, or when assigning sister concordant and discordant genes. We rather added the line of this threshold in the plots to exemplify that when we filter the DGEs by excluding sister-discordant genes, we effectively remove the effect of genes that are expressed in below 10% of the cells (that is the *Seurat* package default threshold for removing lowly expressed genes). By excluding the sister-discordant genes, the proportion of these lowly expressed genes drops from 9% to ~1%, as we stated in the R2 response. We hope this clarifies the issue and apologise for not making ourselves sufficiently clear in the previous response.

Reviewer #4 (Remarks to the Author):

This manuscript by Dai et al. addresses single-cell transcriptional priming that correlates with cancer cell therapy resistance and then it takes advantage of these transcriptional signatures to harvest from databases any compounds that can successfully synergize to increase efficacy. The ideas are good, even if not extremely novel at this point, and the execution of experiments is largely correct. I still think this would be a nice contribution to the field, provided the analyses are performed to the standards that this budding community of single-cell lineage-tracing researchers requires.

Response: We appreciate the Reviewer for the overall positive and constructive comments, which have been very encouraging. Please find our detailed point-to-point response below. We have split the remarks and our response in several parts to be able to specifically and clearly address each remark and suggestion.

My biggest concern, which I share with reviewer 1, is the fact that to increase the number of sister cells for early state profiling, the authors decided to use thymidine block. I guess we agree that it should not have been used, especially when trying to isolate intrinsic priming signatures linked to cancer therapy, because this essentially creates a two-stressor situation for the cells. As a result of this experimental design, many confounders arise. Consequently, the authors devise a way to try to “regress” this effect out in a way that I have never seen done before. The biggest point I want to discuss is whether this actually works or not. On one hand, I think that regressing out the effects of thymidine block should not be that hard to do in a more proper way. Simply subtracting the fold

change from the control or pre-treatment gene by gene is not adequate because this is affected by a large number of confounding factors (including hard-to-correct batch effects) and huge noise due to the sparsity of single-cell data (compared to bulk sequencing approaches). More importantly, it can lead to eliminating real signals (the same gene priming that helps survive after thymidine could lead to survival upon another drug). Even without regressing out the thymidine effect, just at the simplest level, authors could have focused on the pathways/correlated gene networks that are only specifically significant in each of their treatment conditions. In other words, I believe that creating a contrast effect should not be necessary and should not be used in this context (regardless of whether it is similar to a function in a well-used package like limma). Yes, authors do run cleaner experiments later, validating many of these observations, but I don't think this is an excuse, particularly because it will lead others who read this manuscript to follow similar ill-advised practices.

Response: We agree that cell cycle synchronisation by thymidine block poses additional stress to cells, and to acknowledge this, we now further state this in the *Discussion* section as follows (new text in bold):

“Fates of cancer cells with recently shared ancestry are closely intertwined, showing aligned drug responses⁴¹ and preferential site of metastasis^{42,43}. Cells of the same progeny, at least until a couple of generations apart, also present similar gene expression profiles^{44,9}. However, transcriptional patterns of pre-treatment cells do not fully capture their future fate⁹, and cells with extreme fates, such as high-dose drug persistence, may not display evident gene expression features prior treatment¹¹. Compared to previous lineage tracing methods that allow more than one cell division after labelling, ReSisTrace has the advantage in identifying subtle and transient fate-coupled differences in gene expression patterns that are increasingly obscured by stochastic noise upon each cell division. Live cell tracking has revealed that divergence in protein expression significantly increases already from sister cells, i.e. two cell progenies, to cousins or four cell progenies⁶. However, a larger progeny size¹¹, and even clonal isolation¹⁰ of pre-resistant populations are beneficial when characterising primed states of extremely rare persisters in order to maximise the odds of detecting each pre-resistant lineage. Thus, the selection of optimal lineage tracing method depends on the context, and is a trade-off between sensitivity to detect and analyse the rare progenitors of extreme persister lineages, and sensitivity to identify more transient expression patterns. **Furthermore, as cell cycle synchronisation not only enriches sister cells but also poses additional stress to the cells being assayed, it should be carefully considered especially in more sensitive cellular contexts, such as patient-derived organoids, or primary cells.**”

To accommodate the suggestion to focus on expression patterns and pathways that are specific for each treatment condition without creating a contrasting signature, we have now fully revised our manuscript. Instead of regressing out the control effects, we now simply analyse and display the association/enrichment of both treatment-specific, un-adjusted pre-resistance signatures, and the control condition pre-survival signature, to allow readers to assess them side by side. This enables the identification of pathways and associations that are only specifically significant in the treatment conditions but not in the control condition. We have thus revised Fig. 2e to display the gene ontology enrichment of the original, un-adjusted pre-resistance signatures. We have further revised all the analyses that assess associations of pre-resistance signatures, such as their connectivity scores with subclonal (Fig. 2d), HRD (Fig. 3e), or drug-induced signatures (Fig. 4b, 4e and Supplementary Fig. 7) to use unadjusted pre-resistance signatures, and show the control condition as the reference.

Please find below the revised subpanels and related text (new text in bold). We also include below the updated *description* of drug prediction as it is related to the drug-induced signatures.

1. Subclonal signatures and gene ontologies (Fig. 2d and e)

Fig. 2, subpanel (d) connectivity scores between the pre-resistance signatures and the subclone specific signatures. *** $P < 0.001$; two-tailed t-test; (Carboplatin $n = 8,419$; Olaparib $n = 8,406$; NK $n = 8,419$; Control $n = 8,410$).

Fig. 2, subpanel (e). Representative gene ontologies enriched in the pre-resistance signatures **for each condition and** the signature of subclone A.

Results, sub-section ‘Subclonal enrichment contributes to pre-resistance signatures’

“This was true also for the control condition, further highlighting the need to **assess** the pre-resistance signals with the experimental fitness signal. “

“With the identity of pre-resistant and pre-sensitive cells, as well as the identity of subclones, we determined rank-based transcriptomic signatures for pre-resistance and subclones (Supplementary Data 2). We found that the signature of subclone A **was positively associated with the pre-resistance signatures for all treatment conditions while negatively associated with the pre-survival signature of the control condition** (Fig. 2d; associations estimated by the connectivity scores defined in²³). **Overall, the patterns were consistent with the enrichment patterns of the subclones in the pre-resistant populations** (Fig. 2c and d). Gene ontology (GO) enrichment analysis revealed that **mRNA surveillance associated terms, nonsense mediated mRNA decay and viral transcription, were enriched in the primed resistance signatures against all three treatment modalities, as well as in subclone A, but not in the pre-survival signal of the control condition.** Changes in proteostasis were associated with primed resistance against olaparib and carboplatin (Fig. 2e, Supplementary Data 3). Response to topologically incorrect protein was associated with primed sensitivity towards olaparib or carboplatin, in line with DNA damaging therapies potentiating endoplasmic reticulum (ER) stress to lethal levels²⁴. We also found that mitochondrial expression patterns were associated with primed sensitivity towards NK, consistent with recent studies²⁵. **DNA replication had a positive association only with the pre-survival signature in the control condition yet not with any of the treatment conditions. This is in line with proliferation increasing the chances of sister cell survival after replating in the control condition, but decreasing survival in the face of anti-proliferative therapies, or even NK attack²⁶.** Interestingly, the cell cycle associations of the two DNA-damaging therapies diverged for mitosis. **Mitosis was enriched in primed sensitivity against either NK cells or carboplatin** that crosslinks DNA regardless of cell cycle phase²⁷. In contrast, olaparib damages DNA by stalling replication forks, where the immediate damage itself needs to occur in the S phase²⁸. **In line with this, mitosis was not associated** with primed olaparib sensitivity. Together, our results show that subtle pre-resistant features can reflect the subclonal phenotypes that are subsequently enriched during each treatment.”

Abstract

The pre-resistant phenotypes were defined by proteostatic **and mRNA surveillance** features, reflecting traits enriched in the upcoming subclonal selection.

Discussion

We found that subclonal structures drove global transcriptional clustering, and discovered treatment-specific enrichment of subclones. These enrichment patterns explained a considerable part of the transcriptomic features of the pre-resistant populations, such as those representing changes in **transcriptional quality control**.

2. HRD (Fig. 3e)

Fig. 3, subpanel (e) Association between the pre-resistance gene signatures and the HRD signature displayed as connectivity scores. Quantiles were determined by varying the adjusted P value threshold for filtering the genes in the HRD signature ($5 \times 10^{-16} < P < 0.05$, $n = [211, 946]$). **** $P < 1 \times 10^{-4}$, two-tailed t-test.

Results, sub-section 'Context dependent association between HRD status and susceptibility to NK killing':

"We then compared the HRD signature to the ~~control-adjusted~~ pre-resistance signatures captured by ReSisTrace. We confirmed that the pre-resistance signatures in olaparib and carboplatin conditions were negatively associated to the HRD signature, as expected based on their mechanisms of action that target HRD (Fig. 3e). More interestingly, the pre-resistance signature for the NK condition showed even more pronounced negative connectivity scores to the signature (Fig. 3e).

3. Drug prediction and drug-induced signatures (Fig. 4b and e; Supplementary Figures 7 and 8)

For the drug predictions that were used as basis for the successful validations shown in Fig. 4, we have already originally used a combination of unadjusted and control condition contrasted signatures but have failed to clearly communicate this in the main text of the manuscript (see the revised text in the listing below). To gain robust shortlists for drug testing, we calculated the connectivity scores between drug-induced log2FCs and six versions of pre-resistance signatures including:

- 1) Original, unadjusted log2FC,
- 2) Original, unadjusted log2FC filtered by the Wilcoxon rank sum test P value < 0.05 ,
- 3) log2FC contrasted between treatment and non-treatment control,
- 4) log2FC contrasted between treatment and non-treatment control filtered by bootstrapping test P value < 0.05 ,
- 5) log2FC contrasted between treatment and non-treatment control filtered by permutation test P value < 0.05 ,
- 6) log2FC contrasted between treatment and non-treatment control filtered by both bootstrapping test P value < 0.05 and permutation test P value < 0.05 .

All were assessed in sister-conserved genes only. We then determined the drugs that showed significant, negative connectivity scores against *all* the signature variants listed above. Hence, we used the control contrasted signatures (3-6) here only as *additional* filtering criteria to exclude drugs that associate with pre-resistance signals that are of similar direction and magnitude in treatment and control conditions. Importantly, in 3 out of 4 control contrasted signatures (4-6), we filtered genes for statistical significance *within* the pre-resistance to pre-survival contrast with bootstrapping and permutation tests. These tests do take into account drop-outs and other noise resulting from the sparsity of single-cell data to effectively differentiate which part of the preR to preS contrast is pronounced in each treatment against the control condition. Although using multiple filtering criteria may leave the possibility of eliminating real signal due to overlap between thymidine and treatment-specific selection pressure, we chose here this rather conservative approach to minimise the chance of targeting the selection pressure purely resulting from the chosen experimental setting (including the thymidine block), to maximise chances of targeting selection pressure specific for each treatment.

To more clearly describe the approach used, as well as to update the description of results from updated Fig 4b, 4e and Supplementary Figures 7 and 8, we have now revised the text as follows (new text in bold):

Results, sub-section ‘Targeting treatment-specific primed resistance with small molecules’:

We hypothesised that small molecules that shift cancer cells toward pre-sensitive states could sensitise them to the corresponding treatments. We searched the L1000 database³⁴ for such drugs that induce gene expression changes opposite to the pre-resistance signatures (Fig. 4a, upper panel). Top drugs were ranked by the negative connectivity scores, which were further **filtered** by a consensus *P*-value threshold derived from multiple variants of the pre-resistance signatures (Fig. 4b and Supplementary Data 7). **In addition to the treatment-specific pre-resistance signatures with and without *p*-value thresholding, we used additional variants derived from the contrasts of pre-resistance and pre-survival signatures, either unfiltered or filtered by bootstrapping and/or permutation test *P*-values, and selected drugs that were predicted from all the six signature variants.** Drugs with non- or lowly expressed targets in Kuramochi cells were excluded, resulting in ten drugs for experimental validation (Supplementary Data 8).

“... We found that the predicted drugs indeed induced changes opposite to the corresponding pre-resistance signatures, with significant negative connectivity scores (Fig. 4e, Supplementary Fig. 8). For example, **clofarabine, one of the top predicted pre-sensitisers for NK cells induced transcriptomic changes with highly negative connectivity to NK pre-resistance. Pevonedistat, predicted to pre-sensitise cells against all the three treatments, induced changes with a modest yet significant negative connectivity for their pre-resistance signatures. In contrast, pevonedistat showed a positive connectivity score to the pre-survival signal in the control condition, which was not observed by any other drug.**”

“...Furthermore, we confirmed that pevonedistat - **that induced an expression profile similar to pre-sensitive cells in all the treatments yet not in the control condition** - strongly pre-sensitized Kuramochi cells to all the three treatment modalities already at a low concentration (0.33 μ M), despite having a minor inhibition on cell viability as a monotherapy (Supplementary Fig. 9a).”

Fig. 4 subpanel (b) Connectivity scores between the L1000 drug-induced signatures and acquired pre-resistance signatures filtered by $P < 0.05$.

Fig. 4 subpanel (e) Single-cell gene expression of drug perturbed Kuramochi cells projected on a UMAP, with top predicted drugs shown as separate UMAP projections along with DMSO induced profiles (d), and with induced fold changes compared to those of respective pre-resistance profiles (e).

Supplementary Fig. 7 Single-cell gene expression of drug perturbed Kuramochi cells as UMAP projections, with indicated drug-induced cells as coloured and DMSO control as grey.

Supplementary Fig. 8 Expression changes in Kuramochi cells treated with predicted pre-sensitising drugs, compared to respective pre-resistance profiles. Y axis shows mean drug induced log₂FC against DMSO from scRNA-sequenced Kuramochi cells, X axis shows mean PreR to PreS log₂FC in respective condition.

4. Methods section

In line with the changed approach, we have further revised the Methods section as follows:

Methods, sub-section '*Pre-resistant gene expression signatures*':

"After identifying the pre-resistant and pre-sensitive cells in the pre-treatment samples, fold changes of gene expressions were determined. The \log_2FC between pre-resistant and pre-sensitive cells for each gene was averaged across two replicates.

Methods, sub-section '*HRD signatures and NK signature*':

"... For the HRD signatures, we determined \log_2FC s of HRD to HRP using sister-concordant genes only, thresholded with indicated P values, We used "connectivityScore" function (method = "gwc", gwc.method='spearman', nperm=300) from PharmacoGx R package (v3.0.2)²³ to determine the connectivity scores between the HRD signatures and ~~control-corrected~~ pre-resistance signatures."

Methods, sub-section '*Predicting drugs to target primed resistance*':

"Six variants of pre-resistant gene expression signatures were generated for each treatment with thresholds of P values based on different adjustment methods, including: 1) Original \log_2FC , 2) Original \log_2FC filtered by the Wilcoxon rank sum test P value < 0.05, 3) \log_2FC **contrast between treatment and non-treatment control**, 4) \log_2FC **contrast between treatment and non-treatment control** filtered by bootstrapping test P value < 0.05, 5) \log_2FC **contrast between treatment and non-treatment control** filtered by permutation test P value < 0.05, and 6) \log_2FC **contrast between treatment and non-treatment control** filtered by both bootstrapping test P value < 0.05 and permutation test P value < 0.05.

For the contrast signatures, the \log_2FC between pre-resistant and pre-sensitive cells was calculated by for each gene was averaged across two replicates and then subtracted by the \log_2FC of pre-surviving versus pre-extinct cells from the non-treatment control samples:

$$\frac{\log_2\left(\frac{R_1}{S_1}\right) + \log_2\left(\frac{R_2}{S_2}\right)}{2} - \frac{\log_2\left(\frac{R'_1}{S'_1}\right) + \log_2\left(\frac{R'_2}{S'_2}\right)}{2}$$

where R_1 and R_2 are the gene expression values in treatment pre-R group in repeats 1 and 2, respectively; S_1 and S_2 are the gene expression values in treatment pre-S group in repeats 1 and 2, respectively; R'_1 and R'_2 are the gene expression values in control pre-R group in repeats 1 and 2, respectively; S'_1 and S'_2 are the gene expression values in control pre-S group in repeats 1 and 2, respectively. Only the sister-concordant genes were included in the rank-based signature.

The similarity of pre-resistant signatures and drug-induced consensus gene expression signatures were evaluated by the connectivity scores defined in the Connectivity Map (CMAP) project⁷⁴. The analysis was performed by using the "connectivityScore" function (*method* = "fgsea", *nperm* = 1000) from PharmacoGx R package (1.3.1). The drugs with negative connectivity scores consistently with all of the six variants of pre-resistant signatures (P value < 0.05) were selected as the top candidates (Supplementary Data 7). We filtered out drugs whose targets were not expressed or expressed at below 10% of Kuramochi cells, including pifithrin-mu, AM-404, and methenamine for carboplatin; PSI-7976 and cilengitide for olaparib; oxyphenbutazone and WAY-100635 for NK cells (Supplementary Data 8)."

In addition, we have updated **Supplementary Data 2** to accommodate these changes.

In this regard, why didn't the authors pursue identifying "pre-resistant" clusters instead of essentially doing a pseudobulk analysis (just separating between preR and preS)? It appears very clear to me from the data in the Supp Fig 3a that different treatments enrich preR cells in different regions in the UMAP which may mean that specific clusters are enriched in preR cells (especially compared to the control preR cells). For instance, in Olaparib, there seems to be a lot of cells on the top right part of the UMAP, compared to preR. Did they fully discard this? Again, this might be a simple and relatively easy way to get around thymidine effects.

Response: We agree that identifying cell subpopulations that show specific enrichment patterns for the treatments against the control condition could be a valid option. To accommodate this suggestion, we used the unbiased Leiden clustering results (shown in **Supplementary Fig. 4b**). The cells in the top right corner of the UMAP, mentioned by Reviewer#4 above, belong to transcriptional cluster 0. This transcriptional cluster is significantly enriched for olaparib as well as for carboplatin and NK pre-resistant cells, yet not for either pre-survival or pre-extinct in the control condition, suggesting that indeed this cluster presents treatment-specific signals (see **Table R3.1** below).

Table R3.1 Enrichment of pre-resistant/pre-survival cells within **Leiden cluster 0** according to chi-squared post-hoc test per condition.

	Carboplatin	Olaparib	NK	Control
Residuals	10.03	12.53	6.18	1.91
P -value	0*	0*	0*	0.78

Yet, when comparing the gene expression of this transcriptional cluster 0 against other transcriptional clusters, the DEGs correspond to the DEGs of subclone A (**Fig. R3.1**: $r = 0.97$, P -value $< 2.2e-16$ for a Pearson correlation of log₂FCs), as the transcriptional subpopulations are largely explained by the subclonal genetic structure of the cell line. Hence, it is more informative to explore and target the pre-resistance signals using the overall non-adjusted preR-preS log₂FCs as the basis, rather than using the pre-resistant cell enriched sub-populations. This is because in our cellular context, the transcriptional sub-populations are so strongly defined by subclonal structures, and the DEGs of treatment-enriched subclones can also be extracted without the need for lineage labels, resulting in a lower resolution for primed resistance than our lineage tracing approach.

We have in fact used this lower resolution sub-clonal approach earlier when analysing pre-existing resistant states in clinical tumor specimens taken from the same patients before and after neoadjuvant chemotherapy (Zhang et al 2022; DOI: 10.1126/sciadv.abm1831), as in the clinical context it is not possible to use lineage labels. Yet, in the ex vivo context, we can use lineage labels to leverage the sister cell resolution and detect more subtle changes than those driven by subclonal differences.

Fig. R3.1 Log2 fold changes of the DEGs in Leiden cluster 0 against those of subclone A.

A second minor concern: regarding the super large UMI counts I was also initially puzzled. In a response to reviewer 1 the author claims “In this case the cells had around 140 000 reads in average, ...” But then they previously suggest that they filtered cells with less than 20000 UMIs. So are they talking reads or UMIs? I suspect the confusion might be because in this cell type the reads and UMIs must be similar (i.e. RNA content is so high).

Response: Regarding the 140 000 average in R2 response, we are talking about sequencing reads, and not UMIs (that were originally introduced in the paper Kivioja*, Vähärautio*, et al, *Nature Methods* 2012). We did filter with a threshold on UMIs, and only discussed reads in the response to point out that indeed our sequencing is sufficiently deep to gain ca. 50 000 UMIs per cell, and thus use a threshold of 10 000 or 20 000 UMIs for the data. The RNA content is indeed very high in this cell line with a highly aneuploid, above diploid genome (see Fig. 1 in Tjihuis et al. 2019; doi.org/10.1186/s13039-019-0429-1) and so with the default sequencing depth of 40k or 50k reads/cell, we would gain approximately one read per unique molecule, thus further increasing the level of dropouts in our scRNA-seq data.

Furthermore, in the plots they share, the UMI count per cell truly looks like around 50-100k per cell... So I did some simple math: if each cell has 100k UMI average, and they aimed to encapsulate about 10,000 cells per well (as one can guess by the methods section), this means they must have needed at least (theoretical) 1,000,000,000 reads per well. Since they sequenced 8 wells day 1, and 8 wells post treatment, that's *at the very theoretical minimum* 20 billion reads. It's a ginormous amount of sequencing that doesn't make any sense here. However, looking more carefully at the actual cells used in the analysis, one can tell that there are 24000 cells in the day 1 sample, and just 7802 cells in the post-treatment sample (so, just about 32k cells in the total dataset). That's a bit bizarre considering 10x v3.1 captures about 40% of the cells loaded, which means this should have come out at about 120k cells for the total dataset (4 times more cells). In other words, what I believe happened is the authors really targeted 40k reads per cell (which is usual in most studies). As they expected 120k cells in the dataset, they sequenced ~5 billion reads... However, there were way fewer cells than expected and the reads ended up being spread across 30k cells or so... In addition, it does look like the authors spread those reads unevenly across samples. Likely they did this on purpose, because they wanted to capture more resolution in the pre-treatment condition (to increase accuracy of state-fate analysis), so they assigned more space in the sequencing lane to these pre-treatment libraries. This all seems very plausible to me,

but an actual detailed account of these details on sequencing decisions is missing in the methods section, or at least I didn't find it.

Response: We are sorry that we failed to communicate this more clearly. We loaded 10k to 16k cells from each of the 16 samples to the 10x Chromium controller, using the 10x Chromium v3.1 kit, which according to the manufacturer's table (page 18 in <https://www.10xgenomics.com/support/single-cell-gene-expression/documentation/steps/library-prep/chromium-single-cell-3-reagent-kits-user-guide-v-3-1-chemistry>) should result in approximately 6k to 10k recovered cells, per sample, or 128k cells in total. As the actual recovery rate is typically lower, we recovered on average 6.5k cells per sample (approx 80% of the aimed recovery), or in total around 105k cells before QC and around 78k cells left after Seurat QC. From these 78k cells, we further removed 8,640 cells without lineage label(s) and 28,640 AT-New-cell (after treatment cells with lineage labels that were not found from corresponding before treatment cells), resulting in 40,444 cells (3,102 + **29,540** from before treatment samples and 7,802 cells from after treatment samples) that were used for the analysis (**Supplementary Data 11**). Upon collecting these numbers, we realised that there was a typo in Fig. 2a legend: the number of pre-sensitive cells should be 29,540, not 19,540; this is now corrected in the revised manuscript.

We sequenced approximately 0.86 billion reads per sample, or in total approximately 14 billion reads. This is of course a huge amount of sequencing, especially when considering the numbers of cells that were eventually used for preR-preS analyses, but we initially chose to sequence very deep to achieve high-resolution data with minimal dropout of both labels and transcripts in this proof-of-concept experiment.

To clarify our approach, we revised the *Methods* subsection '**ReSisTrace experimental workflow**' as follows (new text in bold), and added a new table as Supplementary Data 11:

"After the treatment, cells were recovered for 10 (5 cell cycles), 6 (3 cell cycles) and 7 days (3-4 cell cycles) respectively, after which indicated number of cells (**Supplementary Data 11**) from each well were loaded for scRNA-seq as post-treatment samples. **The samples were sequenced very deeply, with approximately 0.86 billion reads per sample, to achieve high-resolution data with minimal dropout of both labels and transcripts in this proof-of-concept experiment. We recovered around 105k cells in total before QC and around 78k cells after Seurat QC. From these cells, we further removed 8,640 cells without lineage label(s) and 28,640 after treatment cells whose lineage labels were missing from the corresponding before treatment specimen (column AT-New-cell in Supplementary Data 11), resulting in 40,444 cells (3,102 + 29,540 from BT and + 7,802 cells from AT samples) that were used for the analysis (detailed in Supplementary Data 11).**"

I really cannot find any major issues with the manuscript that have not been raised before by other reviewers. In fact, I think the authors did pretty excellent work following up their scLT observations. However, before I can recommend the publication of these results I really want them to get the scLT analysis part right.

Response:

We thank the Reviewer for acknowledging our work on the targeting and validation of our lineage tracing observations. We hope that our revised manuscript fulfils this criteria.

–Alejo E. Rodriguez Fraticelli

REVIEWERS' COMMENTS

Reviewer #4 (Remarks to the Author):

The authors have successfully addressed my comments and corrected their methods accordingly. It is clear that the survival signatures for controls are different from the therapy-specific resistance signatures, and that in this case cell-cycle synchronization did not contribute negatively to the discoveries.

I do want to leave one last additional comment. Based on the analyses, clone A signatures are now basically sufficient to explain most of the results in the scLT part (and probably could have been found without the need to use scLT to begin with). I am left to wonder whether "gating" the analyses only on clone A would unveil interesting new observations or not. Do clone-A-specific pre-resistant states exist? Are CNVs just additional "barcodes" or are they the mechanism? Now, I am fully aware that this is not how cancer treatment works, and that treatment strategies need to be chosen to target all genetic (and non-genetic) effects. Therefore, addressing this correctly in the discussion is really important, with regards to the sources of both genetic and non-genetic differences.

Point-by-point response, round 4

We thank the editor and the reviewer for their positive response. We have addressed the remaining comment from the reviewer below, as well as all the items in the Author Checklist. In addition, we have indicated an update to correct for a minor inconsistency in the ‘*Methods*’ section.

Reviewer #4 (Remarks to the Author):

The authors have successfully addressed my comments and corrected their methods accordingly. It is clear that the survival signatures for controls are different from the therapy-specific resistance signatures, and that in this case cell-cycle synchronization did not contribute negatively to the discoveries.

I do want to leave one last additional comment. Based on the analyses, clone A signatures are now basically sufficient to explain most of the results in the scLT part (and probably could have been found without the need to use scLT to begin with). I am left to wonder whether "gating" the analyses only on clone A would unveil interesting new observations or not. Do clone-A-specific pre-resistant states exist? Are CNVs just additional "barcodes" or are they the mechanism? Now, I am fully aware that this is not how cancer treatment works, and that treatment strategies need to be chosen to target all genetic (and non-genetic) effects. Therefore, addressing this correctly in the discussion is really important, with regards to the sources of both genetic and non-genetic differences."

Response: We thank the reviewer for raising this important point. We agree and state in the *Discussion* section of our manuscript that subclonal patterns clearly contribute to pre-resistance patterns, and that we have even taken advantage of this earlier, when analysing a cohort of clinical, longitudinal specimens wherein we have, as indicated by the reviewer above, used subclones as "barcodes" (Zhang et al 2022). Copy number variations may also provide mechanisms for the resistance via affecting the expression levels of cancer drivers but as stated in the discussion, they additionally contain a large number of passenger changes that may mask the effect of driver genes. Importantly, our results show that the higher resolution provided by lineage tracing does reveal biological processes associated with primed resistance that are not represented by comparing the treatment-enriched subclone to other subclones (processes shown in Fig. 2e).

To accommodate the comment, we modified the *Discussion* section as follows (new text in red):

“Non-genetic heterogeneity should ideally be assayed in the context of genetic heterogeneity, as clinical tumours are never completely isogenic due their inherent genomic instability. We have previously shown that subclones can be traced through the treatment in clinical tumours to assess enrichment of pre-existing cellular states⁴⁵. To improve clinical relevance of ReSisTrace in HGSOc that is an extremely copy number unstable cancer⁴⁶, we analysed primed resistance within the subclonal context via scRNA-seq inferred copy number variations. We found that subclonal structures drove global transcriptional clustering, and discovered treatment-specific enrichment of subclones. These enrichment patterns explained certain transcriptomic features of the pre-resistant populations, such as those representing changes in transcriptional quality control. However, the majority of the biological processes enriched in treatment-specific pre-resistance patterns - such as those related to proteostasis or cell cycle - were not detected by subclonal analysis. This suggests that the increased

resolution of sister cell lineage tracing enables the discovery of deeper, non-genetic features preceding resistance that cannot be identified by tracing cellular progenies at the lower, subclonal resolution. It is important to consider that subclonal enrichment contributes to differential expression of not only CNV affected drivers but also the co-amplified passenger genes⁴⁷, thus providing an additional level of noise to the pre-resistance signal and supporting the analysis of pre-resistant states in their subclonal context. Thus, ReSisTrace revealed the interplay between genetic and phenotypic heterogeneity in primed resistance, an aspect that becomes increasingly relevant for patient-derived models such as organoids or xenografts.

Minor correction to the *Methods* section:

We realised we had not fully updated the sub-section '*Pathway analysis*' within '*Methods*' to accommodate the new analysis approach during the previous review round. Please find the updated text below (removed part with strikethrough):

“KEGG pathways for the expression matched sister-concordant and discordant genes were determined by Enrichr (<https://maayanlab.cloud/Enrichr/>)^{66,67}. Gene set enrichment analysis (GSEA) function from the clusterProfiler package (3.18.1)⁶⁸ was used for the ordered fold changes (logarithmic scaled) extracted from the ~~fitness-corrected~~ pre-resistance differential expression analysis to inspect the enriched gene ontology biological process terms. Default parameters were used for the analysis.”